# Structure–function analysis of oncogenic EGFR Kinase Domain Duplication reveals insights into activation and a potential approach for therapeutic targeting

Zhenfang Du [1,10], Benjamin P. Brown[2,3,4,10], Soyeon Kim[5], Donna Ferguson[6], Dean C. Pavlick[7], Gowtham Jayakumaran[6], Ryma Benayed [6], Jean-Nicolas Gallant [1], Yun-Kai Zhang[1], Yingjun Yan[1], Monica Red-Brewer[1], Siraj M. Ali[7], Alexa B. Schrock[7], Ahmet Zehir [6], Marc Ladanyi[6], Adam W. Smith[5], Jens Meiler[3,4,8✉] & Christine M. Lovly [1,9✉]

Mechanistic understanding of oncogenic variants facilitates the development and optimization of treatment strategies. We recently identified in-frame, tandem duplication of *EGFR* exons 18 - 25, which causes EGFR Kinase Domain Duplication (EGFR-KDD). Here, we characterize the prevalence of *ERBB* family KDDs across multiple human cancers and evaluate the functional biochemistry of EGFR-KDD as it relates to pathogenesis and potential therapeutic intervention. We provide computational and experimental evidence that EGFR-KDD functions by forming asymmetric EGF-independent intra-molecular and EGF-dependent inter-molecular dimers. Time-resolved fluorescence microscopy and co-immunoprecipitation reveals EGFR-KDD can form ligand-dependent inter-molecular homo- and hetero-dimers/multimers. Furthermore, we show that inhibition of EGFR-KDD activity is maximally achieved by blocking both intra- and inter-molecular dimerization. Collectively, our findings define a previously unrecognized model of EGFR dimerization, providing important insights for the understanding of EGFR activation mechanisms and informing personalized treatment of patients with tumors harboring EGFR-KDD. Finally, we establish *ERBB* KDDs as recurrent oncogenic events in multiple cancers.

---

[1] Department of Medicine, Vanderbilt University Medical Center, Nashville, TN, USA. [2] Chemical and Physical Biology Program, Vanderbilt University, Nashville, TN, USA. [3] Department of Chemistry, Vanderbilt University, Nashville, TN, USA. [4] Center for Structural Biology, Vanderbilt University, Nashville, TN, USA. [5] Department of Chemistry, University of Akron, Akron, OH, USA. [6] Department of Molecular Pathology, Memorial Sloan Kettering Cancer Center, New York City, NY, USA. [7] Foundation Medicine, Inc., Cambridge, MA, USA. [8] Institute for Drug Discovery, Leipzig University Medical School, Leipzig, Germany. [9] Vanderbilt-Ingram Cancer Center, Vanderbilt University Medical Center, Nashville, TN, USA. [10] These authors contributed equally: Zhenfang Du, Benjamin P. Brown. ✉email: jens.meiler@vanderbilt.edu; christine.lovly@vumc.org

Next generation sequencing (NGS)-based assays have demonstrated high utility as a diagnostic tool for multiple cancer types[1–4]. Interpretation of tumor genomic test results is often complicated by discovery of "variants of unknown significance" (VUS), because insufficient evidence is available to confirm whether the variant is a driver (deleterious) mutation[5,6]. Previously, we identified a VUS in *EGFR* that contains a tandem in-frame duplication of exons 18–25 in an index patient with metastatic lung adenocarcinoma. Since exons 18–25 encode the entire tyrosine kinase domain (TKD), we termed this variant "EGFR Kinase Domain Duplication" (EGFR-KDD)[7].

The ability to effectively treat patients is rooted in our mechanistic understanding of genomic variants identified via sequencing. The classic example is *BRAF* mutations, which are detected in numerous tumors[8]. There are three classes of *BRAF* mutations, stratified by mechanism and therapeutic actionability[8,9]. Generally, class I mutations, most notably V600E, are treated with a B-RAF inhibitor such as vemurafenib or dabrafenib, while class II and III mutations are insensitive to vemurafenib/dabrafenib[8,9]. Thus, a primary goal in precision medicine is to identify and mechanistically characterize mutations and translate these findings into clinically actionable therapeutic strategies.

Regarding EGFR, mutations in the kinase domain involving small deletions in exon 19 or point mutation in exon 21 (L858R) have been well described[10]. These mutations increase receptor activation by stabilizing the active conformation of the kinase domain to promote dimerization[11]. Numerous studies have now shown that patients with EGFR kinase domain mutations benefit from treatment with EGFR tyrosine kinase inhibitors (TKIs), whereas patients with tumors containing wild-type EGFR do not derive benefit[10]. Analogously, mutations in the EGFR extracellular domain (ECD) are detected in patients with glioblastoma but are significantly less sensitive to EGFR TKIs in vitro compared to the EGFR kinase domain mutations found in lung cancer[12], reinforcing the concept that not all mutations within a given gene can be therapeutically targeted in the same manner. In the case of *EGFR*-KDD, the entire gene contains wild-type sequence with an intragenic duplication of exons 18–25. The addition of a second kinase domain to the intracellular region of EGFR introduces a potentially significant structural perturbation. The functional and therapeutic implications of this variant remain uncertain. Moreover, the unique biology of this variant may make it a valuable tool in the study of *ERBB* family members and, more generally, suggests a strategy for the study of kinases.

In the present study, we evaluate the prevalence of KDD in *ERBB* family members (*EGFR*/EGFR, *ERBB2*/HER2, *ERBB3*/HER3, and *ERBB4*/HER4) across multiple types of human cancers in order to refine our understanding of KDD as an oncogenic driver. In addition, we combine structural modeling, biochemical assays, and experimental and computational biophysical analyses to understand the mechanism whereby EGFR-KDD aberrantly activates EGFR. Collectively, these complementary approaches suggest that EGFR-KDD is activated through the formation of ligand-independent intra-molecular dimers and signaling is amplified through ligand-dependent inter-molecular dimers/multimers. Furthermore, we show that inhibition of EGFR-KDD activity is maximally achieved by blocking both intra-molecular and inter-molecular dimerization. These studies have important implications for the treatment of patients whose tumor harbor EGFR-KDD.

## Results

**ERBB family KDDs are recurrent in multiple cancer types.** To investigate the prevalence of KDD in all *ERBB* family members,

we analyzed clinical NGS data from 237,701 tumor samples within the Foundation Medicine (FMI) database. In total, we identified 799 KDDs in *ERBB* family members (0.34%, 799/237,701). Of those 799 KDDs, *EGFR* accounts for 443 (55.4%), *ERBB2* 217 (27.2%), *ERBB3* 92 (11.5%), and *ERBB4* 47 (5.9%). Among the cancers present in the FMI database, *ERBB*-KDD was found most frequently in glioma (2.4%, 227/9381 total glioma cases), followed by upper gastrointestinal cancer (upper GI; 0.8%, 89/11,822) and non-small cell lung cancer (NSCLC; 0.2%, 109/48,699). For *EGFR*-KDD, glioma has the highest frequency (2.4%, 222/9381), followed by NSCLC (1.4%, 70/48,699) and GI (0.3%, 40/11,822) (Table 1a). We observed lower incidences of KDD in *ERBB2*, *ERBB3* and *ERBB4* than *EGFR*, with distributions mirroring those of other observed oncogenic mutations in brain tumors and NSCLC[13–17] (Table 1a).

We also analyzed 40,165 tumor samples from the Memorial Sloan Kettering Cancer Center (MSKCC) IMPACT database (MSK-IMPACT)[18]. These data confirm that KDD occurs most frequently in *EGFR*, followed by *ERBB2* (Table 1b). *EGFR*-KDD is most prevalent in glioma and NSCLC, while *ERBB2*-KDD is most prevalent in breast and gynecological cancers (GYN). These distributions are consistent with the observed distributions of other *EGFR* oncogenic mutations in glioblastoma[13,15] and NSCLC[14,16] and other *ERBB2* mutations in breast cancer[19], supporting the notion that specific genes may be genomically altered through a variety of mechanisms in a given tumor context.

The overall frequency of *ERBB*-KDDs from the two datasets is between 0.58–2.4% in glioma, 0.07–0.22% in NSCLC, and 0.05–0.40% in breast cancer. Differences in detection between the two datasets are likely the result of the different methodologies employed for each dataset to identify KDDs (see the "Methods" section). Nevertheless, these data suggest that *ERBB*-KDD is a recurring oncogenic driver in tumor types known to be dependent on *ERBB* signaling (lung, breast, etc.).

---

**Table 1 Incidence of ErbB family kinase domain duplications (KDDs) across tumor types.**

**(a) Incidence of different ErbB-KDDs across 237,701 solid tumor and hematological samples from the Foundation Medicine (FMI) database**

| Tumor type | *EGFR* | *ERBB2* | *ERBB3* | *ERBB4* | Sample size |
|---|---|---|---|---|---|
| Glioma | 222 | 1 | 4 | 0 | 9381 |
| NSCLC | 70 | 18 | 11 | 10 | 48,699 |
| GI | 40 | 44 | 3 | 2 | 11,822 |
| Melanoma | 2 | 0 | 5 | 0 | 6837 |
| Prostate | 4 | 1 | 1 | 4 | 8203 |
| Bladder | 6 | 3 | 2 | 2 | 4886 |
| GYN | 10 | 40 | 34 | 16 | 26,873 |
| Breast | 27 | 52 | 13 | 7 | 24,467 |
| HNC | 2 | 7 | 1 | 0 | 5380 |

**(b) Incidence of different ErbB-KDDs across 40,165 solid tumor and hematological samples from Memorial Sloan Kettering Cancer Center (MSKCC) IMPACT database**

| Tumor type | *EGFR* | *ERBB2* | Sample size |
|---|---|---|---|
| Glioma | 10 | 0 | 1735 |
| Breast | 1 | 2 | 5614 |
| NSCLC | 3 | 1 | 5986 |
| Bladder | 0 | 1 | 1266 |
| GYN | 0 | 1 | 1600 |

NSCLC: non-small cell lung cancer; GI: gastrointestinal cancer; GYN: gynecological cancer; HNC: head and neck cancer.

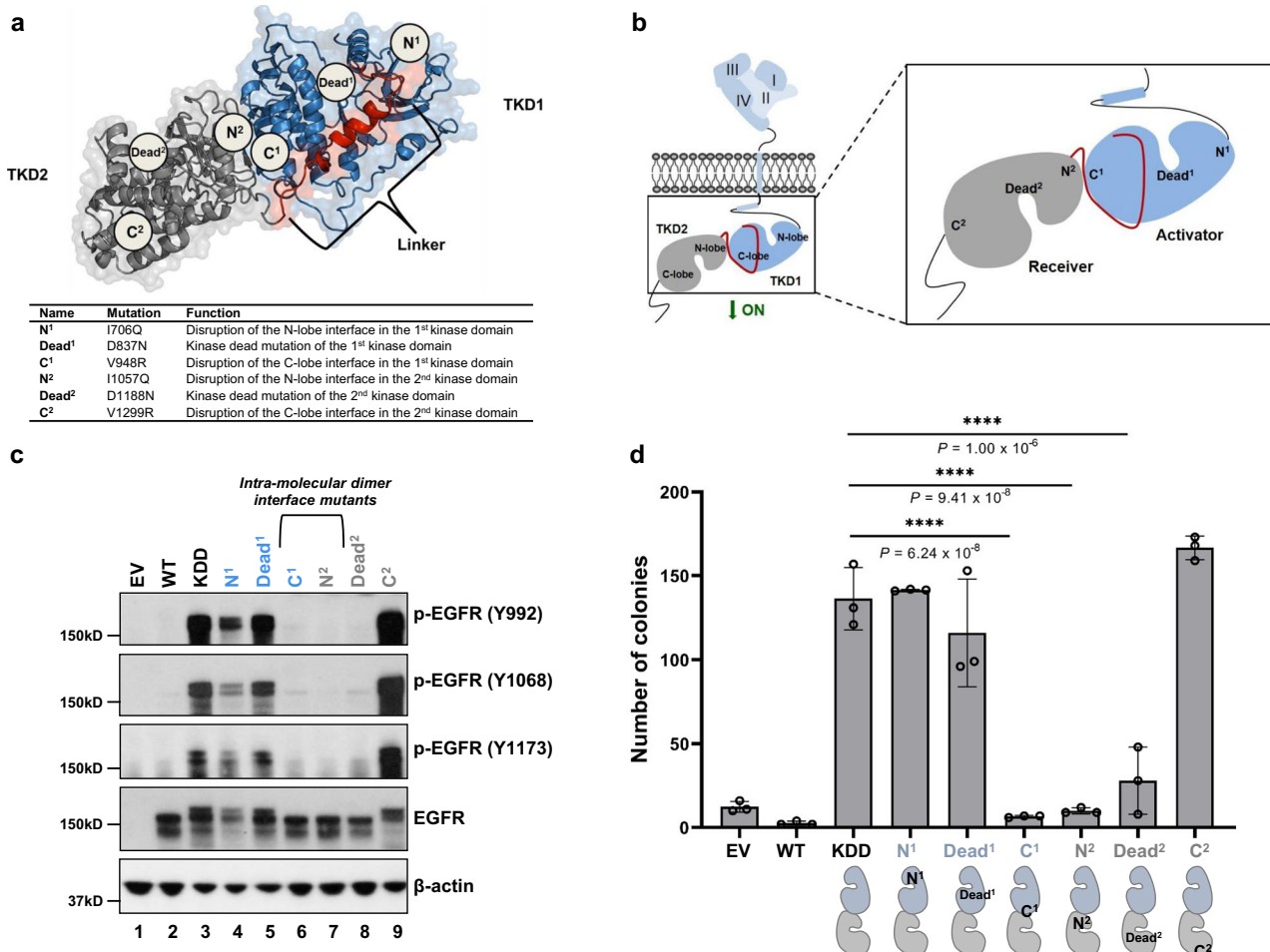

**Fig. 1 Mutations disrupting the potential intra-molecular dimer interface abrogate phosphorylation of EGFR-KDD and anchorage-independent growth.**
**a** Ribbon diagram and space-filling model of EGFR-KDD kinase domains. Mutations constructed in this study were labeled. **b** Schematic representation of mutations we constructed in this study. We generated point mutations disrupting the potential intra- ($C^1$, $N^2$) and inter-molecular ($N^1$, $C^2$) dimer interface as well as mutations inactivating kinase activity of each kinase domain (Dead[1], Dead[2]). **c** YAMC cells stably expressing EGFR-KDD and its mutants. Cells were cultured for 48 h and then harvested and lysed for analysis. Total EGFR and auto-phosphorylation at three tyrosine sites were evaluated by western blot. $n = 3$ experiments were repeated independently with similar results. EV empty vector; WT, EGFR-WT; KDD, EGFR-KDD. **d** Soft agar assays were performed in six-well plates by using YAMC cells. 5000 cells were seeded in each well and colonies were counted after 4 weeks. $n = 3$ biologically independent replicates were examined over three independent experiments with similar results. Data are presented as mean values ± SD. One-Way ANOVA test with Bonferroni post hoc test was performed to obtain the adjusted $P$ values. For **a**, the model coordinates are provided in Supplementary Data 2. For **c** and **d**, results are the representative of three independent experiments. Source data are provided as a Source Data file.

**EGFR-KDD is a constitutively active intra-molecular dimer.**
Even within a single driver gene, the type of mutation that occurs can influence prognosis and drug responsiveness. It is therefore critical to fully characterize the functional consequences of genomic variants in clinically relevant genes. To help us probe the biochemistry of the EGFR-KDD intra-molecular dimer, we leverage core principles of EGFR receptor biology.

*ERBB* family members are transmembrane tyrosine kinases that possess an extracellular ligand-binding domain, a single-pass transmembrane domain, a juxtamembrane (JM) region, an intracellular TKD, and a carboxy (C-) terminal tail with multiple tyrosine phosphorylation sites[20]. Biochemical and crystallographic studies have shown that activation of EGFR-wild type (WT) involves ligand-induced asymmetric homo- or heterodimerization of two TKDs. In the presence of ligand, the C-lobe of one TKD (activator) contacts the N-lobe of another TKD (receiver) to relieve autoinhibition and activate the receiver TKD[21]. Previous studies of EGFR-WT have identified mutations

at the inter-molecular dimer interface that can disrupt dimerization and prevent EGFR-WT enzymatic activity[21].

EGFR-KDD is composed of two intact kinase domains[7] (Fig. 1a). We hypothesized that the forced proximity of the two adjoined kinase domains could form a constitutively active intra-molecular asymmetric dimer in the absence of ligand. To test this hypothesis, we engineered EGFR-KDD constructs with putative intra-molecular dimer disruption mutants (for EGFR mutations, we utilized protein numbering of the human immature EGFR sequence that includes the 24-residue signal sequence) (Fig. 1a, b, Supplementary Table 1): V948R ($C^1$; C-lobe of TKD1) and I706Q ($N^1$; N-lobe of TKD1) in TKD1, and V1299R ($C^2$; C-lobe of TKD2) and I1057Q ($N^2$; N-lobe of TKD2) in TKD2. We also introduced catalytically inactivating mutations (kinase dead) into each TKD individually (D837N in TKD1 and D1188N in TKD2; Dead[1] and Dead[2], respectively) (Fig. 1b). We reasoned that these mutants would help us to determine: (1) if EGFR-KDD is catalytically active in the absence of ligand stimulation, (2) the

relative orientation of the two intra-molecular kinase domains (i.e. activator vs. receiver), and (3) which of the kinase domains (or both) is catalytically active.

EGFR-KDD and the mutants described above were stably expressed in NR6[22] (low endogenous EGFR expression) and YAMC (EGFR$^{-/-}$)[23] cells. We evaluated EGF ligand-independent phosphorylation at EGFR C-terminal tyrosine sites. Ligand-induced dimerization of EGFR-WT results in auto-phosphorylation of its C-terminal tyrosine residues, including Y992[24], Y1068[25], and Y1173[25] (Y1343, Y1419, and Y1524 for EGFR-KDD, respectively). For EGFR phosphorylation sites, we utilized protein numbering of mature EGFR sequence that does not include the 24-residue signal sequence (Supplementary Table 1). We observed that EGFR-KDD, but not EGFR-WT, displays phosphorylation of all three tyrosine residues in the absence of EGF ligands (Fig. 1c, lanes 2, 3), indicating that EGFR-KDD is catalytically active without ligand stimulation. We also found that the intra-molecular dimer interface mutants, C$^1$ and N$^2$ (Fig. 1c, lanes 6, 7; Supplementary Fig. 1a, lanes 6, 7), abolish phosphorylation at all three sites, while N$^1$ and C$^2$ mutants remain phosphorylated in YAMC and NR6 cells (Fig. 1c, lanes 4, 9; Supplementary Fig. 1a, lanes 4, 9), suggesting that the auto-activation of EGFR-KDD was disrupted by C$^1$ and N$^2$ mutants, rather than N$^1$ and C$^2$ mutants. These data suggest that the N-lobe-mutated TKD1 can activate the C-lobe-mutated TKD2, but not the reverse (Fig. 1a).

Our catalytically inactive EGFR-KDD TKD2 mutant (Dead$^2$) failed to autophosphorylate all three tyrosine sites. In contrast, the Dead$^1$ mutant retained phosphorylation levels comparable to EGFR-KDD in both YAMC and NR6 cells (Fig. 1c, lanes 5, 8 and Supplementary Fig. 1a, lanes 5, 8). Therefore, in this intra-molecular dimer model, TKD2 functions as the enzymatically active receiver to TKD1, while TKD1 functions as activator to TKD2 (Fig. 1a).

We further sought to evaluate EGFR-KDD in a phenotypic assay. In both YAMC and NR6 cells, we observed robust colony growth in cells stably expressing EGFR-KDD (Fig. 1d, Supplementary Fig. 1b). We observed that there were comparable numbers of colonies in N$^1$ and C$^2$ mutants compared with EGFR-KDD, while significantly fewer colonies were observed in the intra-molecular dimer-disrupted C$^1$ and N$^2$ mutants (Fig. 1d, Supplementary Fig. 1b). We also found that Dead$^1$, but not Dead$^2$, could support anchorage-independent growth of YAMC (Fig. 1d) and NR6 (Supplementary Fig. 1b) cells. Therefore, our phenotypic data provide evidence that reduced phosphorylation in the C$^1$ and N$^2$ intra-molecular dimer-disrupted mutants diminish anchorage-independent growth. Taken together, these data are evidence that EGFR-KDD forms a catalytically active asymmetric intra-molecular dimer in the absence of EGF ligand.

**Linker contributions to intra-molecular dimer stability.** The juxtamembrane B (JMB) domain is an integral component of HER-family homo- and hetero-dimerization. The receiving kinase JMB domain forms specific stabilizing enthalpic contacts in the activator kinase C-lobe (e.g. the hydrophobic residues L688, V689, and L692, and multiple polar contacts)[26,27]. Not surprisingly, the JMB residues are highly conserved in HER-family receptors (Fig. 2a). In EGFR-KDD, the TKD2 JMB is linked directly to the C-terminus of TKD1 (Fig. 2b). Thus, an important question remained as to whether constitutive EGF-independent activation of EGFR-KDD is the result of (A) sequence-specific structural perturbations to the JMB region, or (B) the sterically imposed forced proximity of TKD1 and TKD2. To address this question, we generated all-atom structural models of EGFR-KDD with Rosetta and molecular dynamics (MD)

simulations (Supplementary Fig. 2a–c). For comparison, we also modeled the EGFR-WT homodimer.

We measured the per-residue root-mean-square-fluctuations (RMSF) of the linker residues in EGFR-KDD. Our modeling suggests that the linker region corresponding to the JMB is less flexible than the activator C-terminus region, particularly near the N-terminal portion of the JMB (Fig. 2c, d). Therefore, we hypothesized that the EGFR-KDD JMB forms enthalpically stabilizing contacts at the intra-molecular dimer interface.

To test this hypothesis, we replaced pieces of the linker with unstructured glycine-glycine-serine (GGS) repeats. We substituted (GGS)$_3$ for the JMB part of the linker (KDD-(GGS)$_3$) and (GGS)$_6$ for the activator C-terminus part of the linker (KDD-(GGS)$_6$) (Fig. 2b). Substitution with (GGS)$_x$ exchanges sequence-specific contacts with a non-interacting, flexible sequence of matching length[28]. We transiently transfected the mutants into HEK293 cells and measured EGF-independent receptor phosphorylation via Western blot analysis. KDD-(GGS)$_3$ displays decreased phosphorylation relative to EGFR-KDD, while KDD-(GGS)$_6$ retained similar levels of phosphorylation as EGFR-KDD (Fig. 2e, lanes 3–5). Importantly, KDD-(GGS)$_3$ retains increased activity compared to EGFR-WT (Fig. 2e, lanes 2, 4). Taken together, these data suggest that residues in the JMB portion of the linker contribute to the stability of the EGFR-KDD intra-molecular dimer.

Interestingly, the most stable EGFR-KDD linker model packs two leucine residues (L1038 and L1039) against helices αE and αI, corresponding structurally to residue V689 in EGFR-WT (Fig. 2f, Supplementary Figs. 2b, c, and 3a–d). EGFR-WT V689 has previously been shown to be necessary for EGFR-WT dimer-dependent phosphorylation[27]. In agreement with these data, our equilibrated EGFR-WT homodimer preserves the V689 contact (Fig. 2f, Supplementary Fig. 3b). Because L1038 and L1039 were among the most stable residues in the model and correspond structurally to an EGFR-WT residue known to stabilize dimerization (V689), we hypothesized that mutation of these residues would impair EGFR-KDD EGF-independent intra-molecular dimer activity.

To test this hypothesis, we performed site-directed mutagenesis at residues L1038 and L1039. In support of this hypothesis, simultaneous introduction of L1038A/R and L1039A/R (KDD-LLAA and KDD-LLRR) resulted in a substantial reduction in phosphorylation (Fig. 2g, lanes 6, 9). Critically, however, KDD-(GGS)$_3$, KDD-LLAA, and KDD-LLRR all retain increased phosphorylation relative to EGFR-WT (Fig. 2e, lanes 2, 4; Fig. 2g, lanes 2, 6, 9). Individual point mutations L1038A/R and L1039A/R do not appreciably reduce phosphorylation; only the combined mutations reduce phosphorylation. Importantly, the sequential leucine residues in the linker are a unique feature of EGFR-KDD resulting from the domain fusion. Altogether, this suggests that despite sequence-dependent JMB contributions to stability, the forced proximity of TKD1 and TKD2 is sufficient for the formation of EGF-independent active intra-molecular dimers. Nevertheless, the linker sequence can provide additional enthalpic stabilization to increase activation.

**Ligand induces inter-molecular multimer activity.** EGFR-WT activation is achieved through ligand-induced inter-molecular dimerization[21]. Recent evidence demonstrates that EGFR-WT also forms tetramers and other small oligomers that increase phosphorylation in an EGF concentration-dependent manner[29–32]. We wanted to know if EGFR-KDD activity is similarly augmented by EGF-ligand stimulation.

To differentiate between EGFR-KDD activity caused by EGF-dependent inter-molecular dimerization and EGF-independent

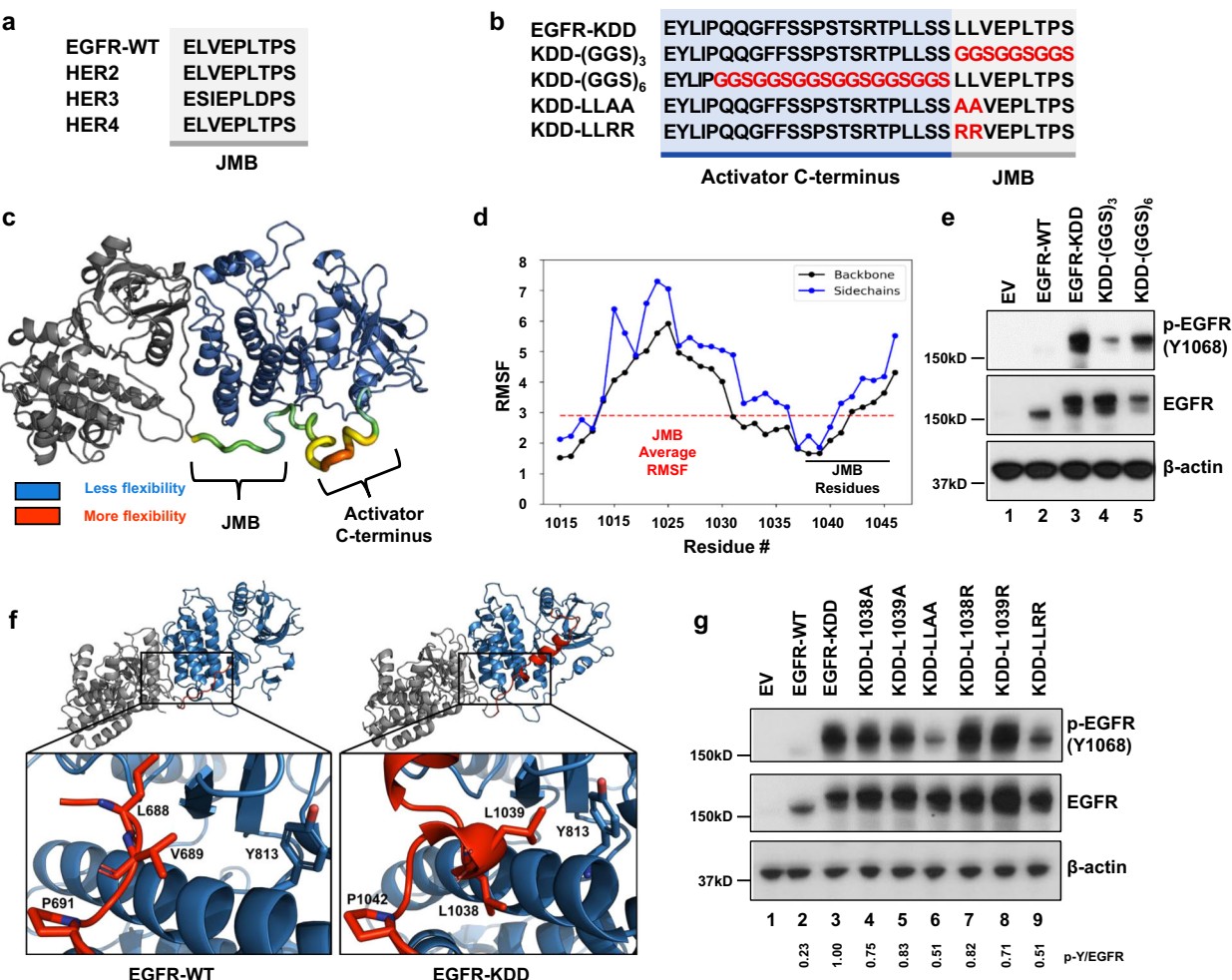

**Fig. 2 The EGFR-KDD linker has distinct enthalpic and entropic contributions to intra-molecular dimer formation. a** Amino acid sequence alignment of EGFR-WT, HER2, HER3, and HER4 JMB domain. **b** Amino acid sequence alignment of EGFR-KDD mutants to evaluate linker contributions. Residues in the activator C-terminus kinase domain (TKD1) highlighted in blue (white font). Residues in the receiver JMB domain highlighted in gray (black font). Mutations indicated by red font. **c** Per-residue root-mean-square-fluctuation (RMSF) of the EGFR-KDD linker region following an additional 1 μs of MD simulation (post-Rosetta modeling and initial 1 μs MD simulation). RMSF values are mapped onto the structure to indicate regional flexibility. Color gradient and cartoon structure width indicate flexibility. Less flexible = smaller width, colored blue; more flexible = larger with, colored red. **d** Graphical representation of per-residue RMSF displays linker residue # on x-axis and RMSF on y-axis; black horizontal line indicates JMB residues, red dashed horizontal line indicates average RMSF of JMB residues. **e** HEK293 cells transiently transfected with EGFR-KDD or (GGS)$_n$ mutants. After 48 h transfection, cells were collected for western blot analysis. EV empty vector. **f** Detailed structural models of the EGFR-WT homodimer with the JMB domain, and the EGFR-KDD intra-molecular dimer, were generated with Rosetta and refined with 1 μs MD simulations. **g** HEK293 cells transiently transfected with EGFR-KDD and different JMB interface mutants. After 48 h transfection, cells were collected for western blot analysis. p-Y/EGFR, the ratio of phosphotyrosine content at Y1068 to total EGFR expression for each construct relative to EGFR-KDD was shown. EV empty vector. For **e** and **g**, n = 3 experiments were repeated independently with similar results. Results are the representative of three independent experiments. Source data are provided as a Source Data file. For **c** and **f**, the model coordinates are provided in Supplementary Data 1–3.

intra-molecular dimerization, we utilized cetuximab, an anti-EGFR ECD antibody that blocks EGF-mediated EGFR dimerization[33]. EGF binding leads to inter-molecular dimerization of EGF receptors. Cetuximab prevents EGF binding by blocking the EGF-binding site. We stimulated cells expressing various EGFR-KDD constructs with EGF. We found that phosphorylation of EGFR-KDD is dramatically increased in the presence of EGF stimulation (Fig. 3a, lanes 5, 6; Fig. 3b, lanes 5, 7; Supplementary Fig. 4a, lanes 5, 6). Addition of cetuximab effectively mitigates EGF-induced phosphorylation of EGFR-KDD (Fig. 3b, lanes 5–8, Supplementary Fig. 4b, lanes 9–12). These data suggest that EGF stimulation may promote EGFR-KDD activity through the formation of at least inter-molecular dimers; however, cetuximab does not preclude the formation of dimers entirely.

To further test the hypothesis that EGF stimulation promotes the formation of at least inter-molecular dimers in EGFR-KDD, we administered mAb806 to YAMC EGFR-KDD cells. The mAb806 antibody inhibits EGFR dimerization by binding to ECD II (residues 287–302)[34], rather than the EGF ligand-binding site in domain III[33]. Thus, inhibition with mAb806 is highly complementary to similar experiments performed with cetuximab. As expected based on our cetuximab results, we found that mAb806 had no impact on phosphorylation level in the absence of EGF ligand (Supplementary Fig. 4c, lanes 1, 2, 5, 6) and decreased the level of phosphorylation with EGF-ligand stimulation (Supplementary Fig. 4d, lanes 3, 4, 7, 8). We also note that phosphorylation was reduced more by cetuximab than mAb806 at approximately equimolar concentrations, consistent with

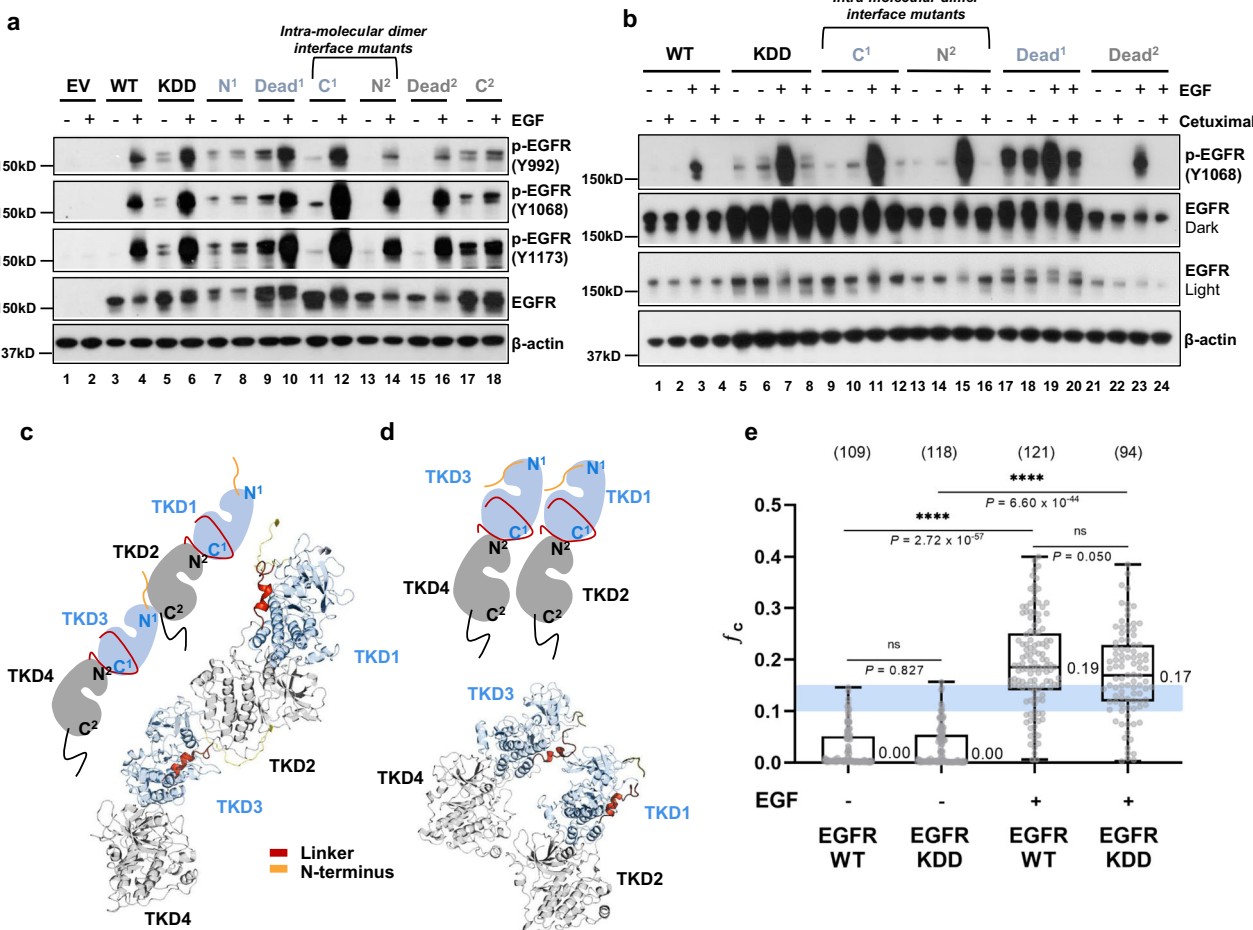

**Fig. 3 EGFR-KDD forms inter-molecular dimers and higher order oligomers after ligand stimulation. a** YAMC cells were cultured in serum-free medium for 12 h and then treated with 50 ng/mL EGF ligand for 5 min. Total EGFR and the autophosphorylation at three tyrosine sites were assessed by western blot. **b** YAMC cells were starved for 12 h and treated with cetuximab (10 μg/ml in serum-free medium) for 3 h 45 min, and EGF ligand (50 ng/mL in serum-free medium) was added for 15 min. The cells were harvested and analyzed by Western blot. WT, EGFR-WT; KDD, EGFR-KDD. **c** Template-based structural models of the intracellular portion of the EGFR-KDD inter-molecular dimer based on end-to-end EGFR-WT tetramer models. **d** Template-based structural models of EGFR-KDD inter-molecular dimer based on side-by-side EGFR-WT tetramer model. **e** Cross-correlation values of EGFR-WT and EGFR-KDD with (+) or without (−) ligand (EGF) stimulation is shown. The blue box indicates the $f_c$ value region for dimers. For the box and whiskers plot, the whiskers show the maximum and the minimum; the box shows 25th–75th percentile; and the line in the box is the median value. The median values are reported next to the boxplot. Each gray dot represents the averaged acquisition (10 s, 6 acquisitions) per area per cell. All data points are shown. Numbers in parenthesis above the boxplot are the total number of cells that data were taken on. For **a** and **b**, $n = 3$ experiments were repeated independently with similar results. Results are the representative of three independent experiments. For **e**, one-way ANOVA test with uncorrected Fisher's LSD post hoc test were performed to obtain adjusted and individual $P$ values. Data are presented as median values ± SD. For **a**, **b** and **e**, source data are provided in the Source Data file.

previous reports that the EGFR inhibitory potency of mAb806 is considerably lower than cetuximab[35].

We showed above (Fig. 1c, d) that intra-molecular dimer-disrupted mutants $C^1$ and $N^2$ are not active in the absence of ligand. Unexpectedly, we noticed that EGF-stimulation rescued these mutants, leading to a robust increase in phosphorylation (Fig. 3a, lanes 11–14; Supplementary Fig. 4a, lanes 11–14). We speculated that this could result from either (A) compensatory stabilization of the intra-molecular receiver kinase domains or (B) stabilization of the donor kinase domains during inter-molecular dimerization.

To better understand how inter-molecular dimerization increases EGFR-KDD autophosphorylation, we built template-based structural models of the intracellular portion of the EGFR-KDD inter-molecular dimer based on two proposed EGFR-WT tetramer models: (1) an extension of the inter-molecular dimer

model in which each kinase domain is successively asymmetrically docked with another (end-to-end model)[29] (Fig. 3c), and (2) two asymmetric dimers oriented such that the N-lobe and C-lobe of one dimer are in contact with the N-lobe and C-lobe of the other dimer, respectively (side-by-side model)[31] (Fig. 3d). Other models are possible (e.g. the receiver kinase of one intra-molecular dimer could act as the donor to the receiver kinase of a second intra-molecular dimer). There are currently no experimental structures (e.g. from X-ray crystallography or cryogenic electron microscopy) elucidating the organization of EGFR-WT tetramer or EGFR-KDD inter-molecular dimer. Thus, we built our template-based models of EGFR-KDD intracellular inter-molecular dimer on two published EGFR-WT tetramer models both of which have experimental and computational support.

Our models each consist of two EGFR-KDDs containing an intra-molecular donor (TKD1 or TKD3) and receiver (TKD2 or

TKD4) kinase. Both structural models suggest a mechanism for active-state stabilization of TKD3 during inter-molecular dimerization (Fig. 3c, d). In the end-to-end model, active-state stabilization of TKD3 (inter-molecular receiver, intra-molecular donor) could occur by canonical asymmetric dimerization with TKD2 (inter-molecular donor, intra-molecular receiver)[29] (Fig. 3c). In the side-by-side model, active-state stabilization of TKD3 could occur through sterically impaired inactivation by TKD1 (inter-molecular donor, intra-molecular donor) (Fig. 3d), as observed in the 40 μs MD simulation of the EGFR-WT full-length tetramer model in Needham et al. 2016[31].

We previously observed that Dead[2] (TKD2 and TKD4 are inactive), but not Dead[1] (TKD1 and TKD3 are inactive), ablates EGFR-KDD activity in the absence of EGF (Fig. 1c, lanes 3, 5, and 8). Here, we see that EGF-ligand stimulation robustly revives phosphorylation in Dead[2] (Fig. 3a, lanes 15, 16; Fig. 3b, lanes 21, 23; Supplementary Fig. 4a, lanes 15, 16), suggesting active-state stabilization of TKD3 through the formation of at least inter-molecular dimers (Fig. 3c, d). Less dramatic increases in Dead[1] from baseline intra-molecular dimer phosphorylation are consistent with changes due to ligand-induced EGFR recruitment (Fig. 3a, lanes 9, 10; Supplementary Fig. 4a, lanes 9, 10). Consistent with these results, pre-administration with cetuximab prevents EGF-dependent phosphorylation of Dead[2] and has only a minor impact on Dead[1] phosphorylation. Taken together, these data suggest that in addition to activation of TKD2 and TKD4 by TKD1 and TKD3, respectively, TKD3 becomes catalytically active in the inter-molecular dimer.

To better characterize the effect of EGF on EGFR-KDD and quantify the extent of EGFR-KDD oligomerization in live cells, we performed two-color pulsed interleaved excitation fluorescence cross-correlation spectroscopy (PIE-FCCS)[36]. PIE-FCCS has been previously applied to evaluate EGFR dimerization and multimerization[29,37]. For these experiments, the protein of interest was expressed as a mixture of eGFP and mCherry fusions and single, live-cell measurements were recorded and analyzed as described in the "Methods" section. In the absence of ligand, both EGFR-WT and EGFR-KDD have median cross-correlation ($f_c$) values of 0.00, indicating that they are predominantly monomeric (Fig. 3e, Supplementary Fig. 6b). Stimulation with EGF ligand leads to a significant level of cross-correlation for EGFR-WT ($f_c = 0.19$) and EGFR-KDD ($f_c = 0.17$) (Fig. 3e, Supplementary Fig. 6b), indicating that ligand stimulation induces dimerization and multimerization in both EGFR-WT and EGFR-KDD[36,38]. There is no statistically significant difference between EGFR-WT and EGFR-KDD, suggesting that the kinase duplication does not sterically restrict dimerization and multimerization. Taken together, these data demonstrate that EGFR-KDD forms multimers upon ligand binding.

**EGFR-KDD directly interacts with *ERBB* family members**. Our biophysical studies demonstrate that EGFR-KDD forms ligand-induced homodimers/multimers. We hypothesized that EGFR-KDD could also heterodimerize with EGFR-WT in the presence of ligand. To test this hypothesis, we performed co-immunoprecipitation in HEK293 cells with transiently co-transfected Myc-epitope tagged EGFR-KDD/EGFR-WT and V5-epitope tagged EGFR-WT/EGFR-KDD. We observed that V5-epitope-tagged EGFR-WT can interact with Myc-epitope-tagged EGFR-KDD, and vice versa (Fig. 4a, Supplementary Fig. 6a). We further evaluated potential interactions between EGFR-WT and EGFR-KDD with PIE-FCCS. With the $f_c$ values, we can distinguish homodimerization and heterodimerization, which cannot be assessed with diffusion coefficients alone. EGFR-WT-eGFP and EGFR-KDD-mCherry were simultaneously

expressed in COS7 cells. In the absence of EGF ligand, there was no interaction ($f_c = 0.00$). Upon addition of EGF-ligand, there was a significant increase in cross-correlation ($f_c = 0.22$), indicating the formation of heteromeric complexes (Fig. 4b, Supplementary Fig. 6e, Supplementary Table 2). The positive cross-correlation is rigorous evidence for heteromeric complex formation, but alone is not sufficient to define the interaction strength or stoichiometry of the complexes. For simplicity we will refer to these complexes as heterodimers as this is the minimal size consistent with positive cross-correlation. In agreement with changes to the $f_c$ values, the diffusion coefficients of both EGFR-WT and EGFR-KDD decreased after ligand addition, indicating slower diffusion due to homodimerization and hetero-dimerization/multimerization (Supplementary Fig. 6b, e, Supplementary Table 2).

Heterodimerization is especially important for the activation of HER2 and HER3. HER2 has lost the capacity to bind ligands and activates primarily as a receiver kinase domain through hetero-dimerization with other *ERBB* family members[39,40]. In contrast, the TKD of HER3 has low kinase activity, and HER3 acts as an activator in heterodimers[41]. We hypothesized that EGFR-KDD can also interact with wild-type HER2 and HER3. To test this hypothesis, we performed co-immunoprecipitation. We transiently co-transfected Myc-epitope-tagged EGFR-KDD with V5-epitope-tagged HER2-WT and HER3-WT in HEK293 cells. Independent pulldowns with V5 and Myc antibodies demonstrate that EGFR-KDD could interact with HER2 and HER3 (Fig. 4c, Supplementary Fig. 6c, d). Moreover, we observed quantitatively with PIE-FCCS that EGFR-WT and EGFR-KDD heterodimerize with HER2 to a larger extent in the presence of EGF-ligand ($f_c = 0.10$ and $f_c = 0.16$, respectively) than in its absence ($f_c = 0.00$ and $f_c = 0.06$, respectively) (Fig. 4d, Supplementary Fig. 6f, Supplementary Table 2). Interestingly, our biophysical data suggest that like EGFR-WT, EGFR-KDD also heterodimerizes with HER3 to a greater extent in the presence of NRG1 than in the presence of EGF (Fig. 4e, Supplementary Fig. 6g, Supplementary Table 2). These data demonstrate that EGFR-KDD forms direct interactions with EGFR-WT, HER2, and HER3.

**Intra- and inter-molecular dimer activity dual inhibition**. The dual nature of EGFR-KDD as an EGF-independent active intra-molecular dimer and as an EGF-dependent active inter-molecular dimer/multimer poses a unique therapeutic challenge. Our computational models and experimental data suggest that the ideal therapy would simultaneously reduce intra-molecular and inter-molecular dimer-mediated activity. One potential treatment strategy is therefore the combination of cetuximab with a TKI (here afatinib). Prior pre-clinical literature has suggested that such a combination may be effective in L858R but not Ex19Del[42].

The combination of cetuximab with various EGFR TKIs, including gefitinib[43] and afatinib[44,45], has been tested in lung cancer patients. In a phase I trial, no responses were observed with the combination of cetuximab plus gefitinib[43], and therefore has not been subsequently used in patients. The combination of cetuximab plus afatinib has advanced in the clinic, including a phase I trial (NCT01090011) that included an expansion cohort[44,45]. Results from this trial of cetuximab plus afatinib demonstrated that the combination therapy was effective in achieving tumor reduction (as assessed by CT scans using RECIST criteria) in patients with both Ex19Del and L858R *EGFR*-mutant lung cancer, in contrast to prior pre-clinical data[42]. Importantly, the combination of cetuximab plus TKI is not FDA-approved because there was no benefit (in terms of PFS, intracranial response, and OS) compared to TKI alone, and thus not standardly used in the treatment of patients with Ex19Del or

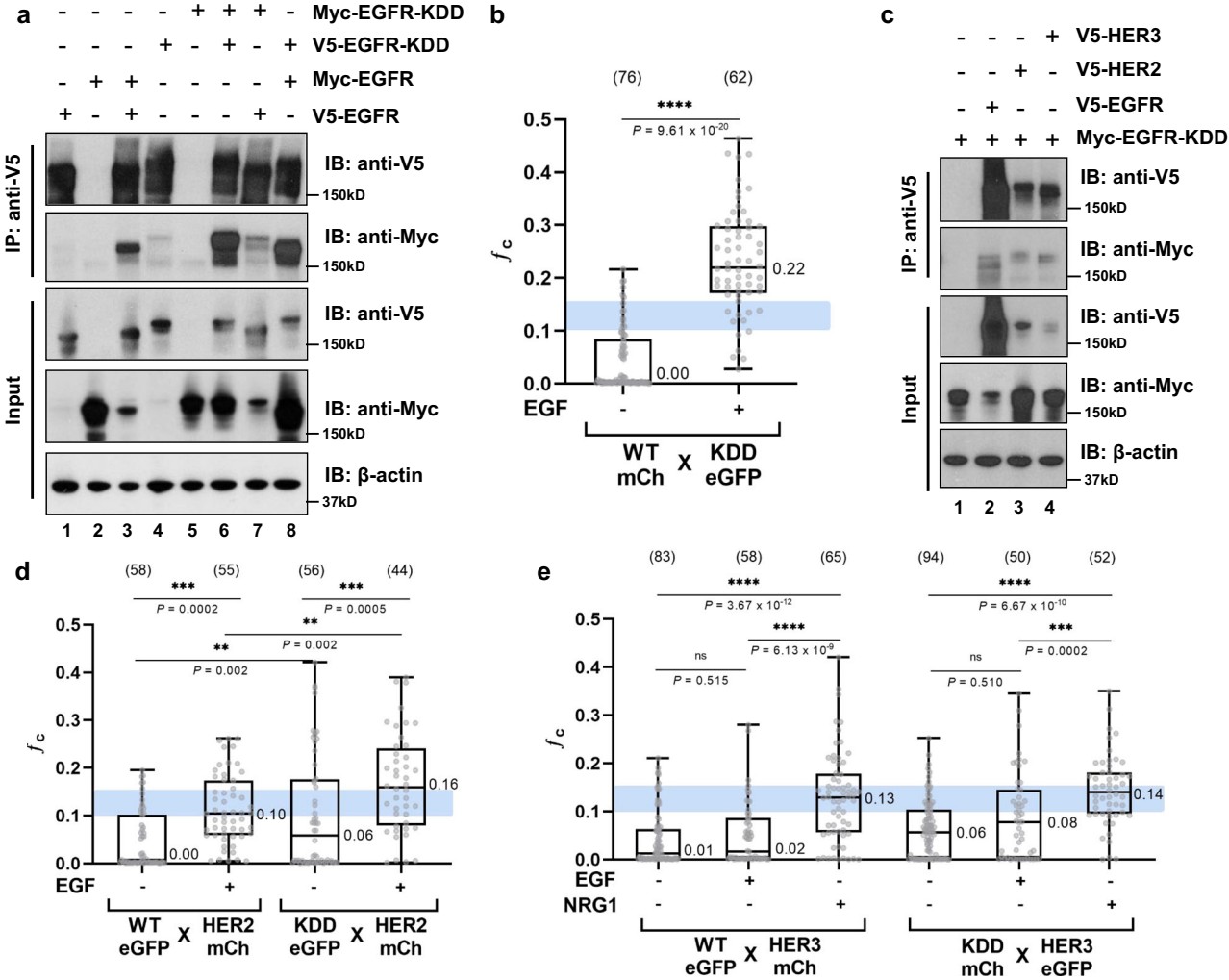

**Fig. 4 EGFR-KDD directly interacts with ERBB family members. a** V5-epitope-tagged EGFR-WT and EGFR-KDD was co-transfected with Myc-epitope-tagged EGFR-WT and EGFR-KDD in HEK293 cells. After 48 h transfection, cells were lysed by hypotonic buffer and the cell lysates were immunoprecipitated by using V5 antibody. Immunoblotting were probed by V5 and Myc antibody. **b** Cross-correlation values of co-transfected EGFR-WT (mCherry-fused) and EGFR-KDD mutant (eGFP-fused) with (+) or without (−) ligand (EGF) stimulation is shown. The light blue box indicates the $f_c$ value region for dimers. **c** Myc-epitope-tagged EGFR-KDD was co-transfected with V5-epitope-tagged EGFR-WT, HER2, and HER3 in HEK293 cells. Cell lysates were immunoprecipitated by using V5 antibody. Immunoblotting were probed by V5 and Myc antibody. **d** Cross-correlation values of co-transfected HER2 (mCherry-fused) and EGFR-KDD mutant (eGFP-fused) with (+) or without (−) ligand (EGF) stimulation is shown. **e** Cross-correlation values of co-transfected HER3 (mCherry-fused) and EGFR-KDD mutant (eGFP-fused) with (+) or without (−) ligand (EGF) stimulation is shown. For **a** and **c**, $n = 2$ experiments were repeated independently with similar results. Results are the representative of two independent experiments. For **b**, **d** and **e**, data are reported as box and whiskers plot, the whiskers show the maximum and the minimum; the box shows 25th–75th percentile; and the line in the box is the median value. The median values are reported next to the boxplot. Each gray dot represents the averaged acquisition (10 s, 6 acquisitions) per area per cell. All data points are shown. Numbers in parenthesis above the boxplot are the total number of cells where data were taken on. One-way ANOVA test with uncorrected Fisher's LSD post hoc test were performed to obtain adjusted and individual P values. Data are presented as median values ± SD. For **a–e**, source data are provided as a Source Data file.

L858R mutations. The current standard of care for these patients is the mutant-selective EGFR TKI, osimertinib, based on a seminal phase 3 clinical trial[46,47].

In contrast, no pre-clinical study or clinical trial has evaluated antibody/TKI combination vs. either alone in EGFR-KDD patients. Indeed, the index patient for EGFR-KDD described in Gallant et al. 2015 unfortunately only had a partial response to afatinib[7]. The anti-tumor response was short-lived (7 cycles of afatinib, or approximately 7 months) before the patient developed acquired resistance to afatinib driven by amplification of the EGFR-KDD allele[7]. Collectively, these observations suggested that more potent EGFR blockade is necessary to overcome the oncogenic activity of EGFR-KDD. Here, we test the hypothesis

that combined TKI and cetuximab treatment will reduce EGFR-KDD-mediated phosphorylation in vitro more than either treatment alone.

We treated YAMC cells stably expressing EGFR Ex19Del (E746_A750del), L858R, and EGFR-KDD with afatinib and cetuximab both in the absence and presence of EGF ligand (Fig. 5a, Supplementary Fig. 7a). Importantly, we observed that in both the absence and presence of EGF, afatinib resulted in a near complete ablation of p-EGFR in Ex19Del (Fig. 5a, lanes 1, 2, 5, 6) and L858R (Fig. 5a, lanes 9, 10, 13, 14), but substantial residual phosphorylation existed in EGFR-KDD (Fig. 5a, lanes 17, 18, 21, 22). As expected, cetuximab alone reduced phosphorylation in Ex19Del, L858R, and EGFR-KDD in the presence of EGF ligand

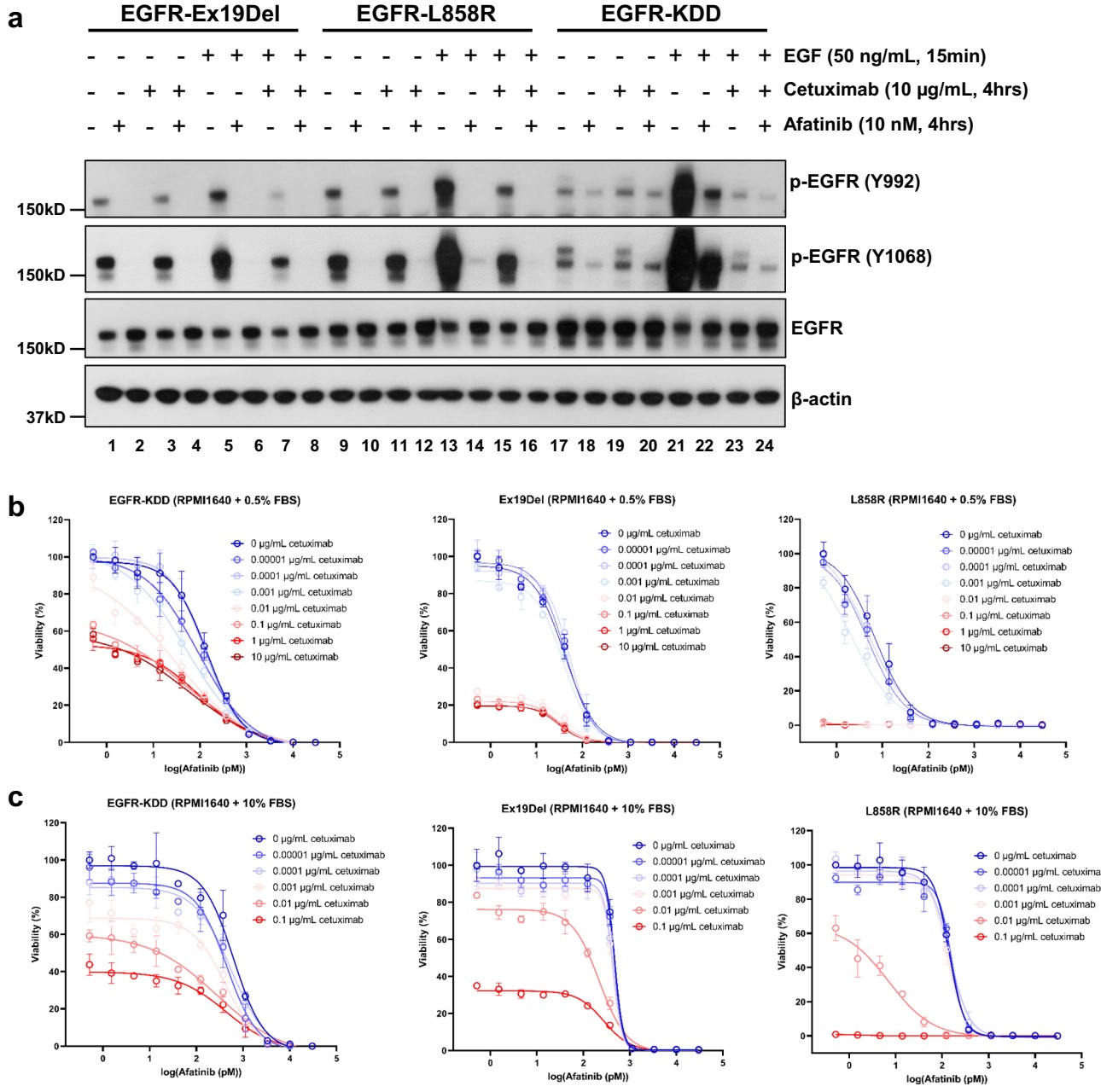

**Fig. 5 Inhibition of EGFR-KDD is maximally achieved by blocking both intra- and inter-molecular dimerization. a** YAMC cells were starved for 12 h and treated with afatinib (10 nM in serum-free medium) and cetuximab (10 μg/ml in serum-free medium) for 3 h 45 min, and then were treated with EGF (50 ng/mL in serum-free medium) for 15 min. The cells were harvested and analyzed by Western blot. **b** Cell Viability Assay was performed in mIL3-independent Ba/F3 cells stably expressing EGFR-KDD, Ex19Del and L858R supplemented with 0.5% FBS. **c** Cell Viability Assay was performed in mIL3-independent Ba/F3 cells stably expressing EGFR-KDD, Ex19Del, and L858R supplemented with 10% FBS. For **a**, $n = 3$ experiments were repeated independently with similar results. For **b** and **c**, $n = 3$ biologically independent replicates were examined over three independent experiments with similar results. Data are presented as mean values ± SD. Results in a, **b** and **c** are the representative of three independent experiments. Source data are provided as a Source Data file.

(Fig. 5a, lanes 7, 15, 23). Notably, the greatest reduction of phosphorylation for EGFR-KDD occurred with the combination of cetuximab+afatinib in the presence of EGF (Fig. 5a, lanes 21, 22, 23, 24). These data suggest that phosphorylation of EGFR Ex19Del and L858R is abolished by afatinib (TKI) or cetuximab alone, and addition of cetuximab to afatinib does not add substantially more inhibition to the decrease in autophosphorylation. Unlike EGFR Ex19Del and L858R, phosphorylation of EGFR-KDD is inhibited by both afatinib and cetuximab as single agent, but the combination treatment yielded more inhibitory effects.

We also performed viability assays with BaF3 cells stably expressing EGFR-KDD, Ex19Del (E746_A750del) or L858R. First, we evaluated Ba/F3 cell growth in serum starved (0.5% fetal bovine serine; FBS) conditions to minimize EGF activation (Supplementary Fig. 7b). At 0.5% FBS, cetuximab maximally exhibited ~40% inhibition of EGFR-KDD, ~80% inhibition of Ex19Del, and almost 100% inhibition of L858R cell viability (Fig. 5b and Supplementary Table 3). These data are consistent with a model in which EGFR-KDD retains an active intramolecular dimer in the absence of EGF stimulation (Fig. 1) and previously published models of Ex19Del and L858R in which

intrinsic αC-helix stabilization transforms them into dimer-dependent "super acceptor" kinases[11,48]. Indeed, progressively higher concentrations of FBS and the addition of exogenous EGF resulted in stable or increased viability of all mutants in the presence of cetuximab, though EGFR-KDD proved to be the least inhibited (Fig. 5c, Supplementary Fig. 7c–e, and Supplementary Table 3).

In 0.5% FBS conditions with minimal EGF-ligand present, the potency of afatinib on EGFR-KDD is approximately equivalent in the absence (0 μg/ml) and presence (10 μg/mL) of cetuximab ($EC_{50} = 0.103 \pm 0.035$ and $0.095 \pm 0.040$ nM, respectively). Similar results are observed in Ex19Del ($EC_{50} = 0.061 \pm 0.027$ and $0.060 \pm 0.017$ nM, respectively). The near complete ablation of Ba/F3 L858R viability at higher concentrations of cetuximab mask any potential similar effects. Generally, we observe that Ex19Del and L858R are more sensitive to afatinib than is EGFR-KDD (Fig. 5b and Supplementary Table 4), consistent with our phosphorylation assays (Fig. 5a and Supplementary Fig. 7a).

As the concentration of EGF-ligand in the medium is increased, we observe not only an increase in viability with cetuximab and increased $EC_{50}$ of afatinib, but also a greater potentiation of afatinib by cetuximab (Fig. 5b, c and Supplementary Fig. 7c–e). In 10% FBS+ 50 ng/ml exogenous EGF, we observe a 5.8× increase in afatinib potency transitioning from 0 to 10 μg/ml in Ba/F3 EGFR-KDD cells. We also observe potentiation of afatinib in Ex19Del (4.7×) and L858R (3.7×) (Supplementary Fig. 7e and Supplementary Table 4). Compared to Ex19Del and L858R, the larger potentiation of afatinib inhibition of Ba/F3 EGFR-KDD by cetuximab seems to be mediated by the lower inhibition of EGFR-KDD by afatinib. Together, our data suggests that a lower dose of afatinib can be administered to maximally inhibit EGFR-KDD when supplemented with cetuximab.

## Discussion

In this study, we combined methods in clinical genomics, computational structural biology, biochemistry, and biophysics to mechanistically characterize a former VUS, EGFR exon 18–25 Kinase Domain Duplication (EGFR-KDD). To investigate the prevalence of KDD in all ERBB family members across various cancers, we analyzed comprehensive genomic profiling data from two large databases. We discovered that ERBB-KDDs are recurrent at a frequency between 0.58% and 2.4% in glioma, 0.07–0.22% in NSCLC, and 0.05–0.40% in breast cancer. We identified fractions of KDDs in multiple other tumor types as well. No previous studies have reported KDD in ERBB2, ERBB3, and ERBB4. These data indicate that ERBB-KDDs account for a small but significant fraction of ERBB family-mediated cancers, and suggest utility of approved targeted therapies for patients based on standard of care clinical genomic testing. Importantly, developing targeted therapies for uncommon variants has precedent. ROS1 variants account for ~1% of lung cancers[49] and have been detected with lower prevalence in multiple other cancers[50] and NTRK fusions have been implicated in 0.31% of adult tumors and in 0.34% of pediatric tumors[51]. There are TKIs targeting both ROS1 and NTRK[52,53] that are FDA approved and additional agents in clinical development. Further, in the case of KDD, both TKIs and antibody therapies already exist for ERBB receptors, thus new trials and therapeutic strategies for this population does not depend on new therapy development.

We sought to elucidate the mechanisms of EGFR-KDD-driven oncogenicity. We demonstrate that EGFR-KDD forms a catalytically active asymmetric intra-molecular dimer in the absence of EGF-ligand stimulation. Mutations disrupting the intra-molecular dimerization interface abolish the phosphorylation of EGFR-KDD in its monomeric form, and the loss of

phosphorylation in these mutants can be recovered by the formation of inter-molecular dimerization and multimerization. These data demonstrate that ligand-independent constitutive activation of EGFR-KDD is driven by asymmetric intra-molecular dimerization.

We next characterized differences in the functionality of the JMB region of EGFR-KDD relative to EGFR-WT. The JMB is a conserved stretch of amino acids critical for inter-molecular dimerization in wild-type ERBB-family receptor kinases. In EGFR-KDD, the JMB region of TKD2 is covalently linked to the C-terminus of TKD1. All-atom computational modeling investigations coupled with in vitro mutagenesis suggests the EGFR-KDD linker region is capable of forming specific stabilizing JMB domain contacts within the intra-molecular dimer; however, the forced proximity of the two kinase domains by the linker is sufficient for elevated EGFR-KDD activity relative to EGFR-WT. In comparison, EGFR-WT depends on stable contacts in the JMB domain for dimer activity[26,27]. We focused our analysis on EGFR-KDD with duplication of exons 18–25, but other groups have recently identified EGFR-KDD with longer duplications (e.g. exons 14–26 and exons 17–25)[54] that may reduce the likelihood of forming stabilizing contacts at the linker JMB interface of the intra-molecular dimer. We speculate that there may be selective pressure for specific linker lengths/sequences in the formation of KDDs. Recent investigations have suggested similar structural constraints in the context of BRAF, HER2, and EGFR β3-αC deletion mutations[55].

EGFR-KDD further forms EGF-dependent inter-molecular dimers. Inter-molecular dimerization of EGFR-KDD increases activity in part by stabilizing the active conformation of the EGFR-KDD donor kinase domain. This has broad implications for HER-family signaling as well. We speculate that the formation of dual activator/receiver kinases in higher order oligomers of HER-family receptors may contribute to ligand-dependent increases in phosphorylation[29,31]. In the present study, we did not identify the configuration of the EGFR-KDD inter-molecular dimer/multimer. Mutations at $N^1$ and $C^2$ only partially disrupted EGF-dependent phosphorylation (Fig. 3a, lanes 7, 8, 17, 18; Supplementary Fig. 4a, lanes 7, 8, 17, 18). Moreover, in the side-by-side structural model, the N-termini of TKD1 and TKD3 are oriented in close proximity (Fig. 3c, yellow), while in the end-to-end model they are separated (Fig. 3b, yellow). Consequently, we considered the end-to-end model less likely to form interactions between the N-terminal juxtamembrane A (JMA) and TM domains of the two interacting proteins, a key feature of inter-molecular dimerization in EGFR-WT[26,27]. Nevertheless, it is clear that EGFR-KDD is forming an EGF-dependent inter-molecular dimer. We anticipate that future investigations will identify the most likely inter-molecular configuration(s).

Interestingly, EGF-stimulated EGFR-KDD displays substantially more phosphorylation than EGF-stimulated canonical-activating mutations (Fig. 5a and Supplementary Fig. 4a). We speculate that this may be because of the increased ratio of EGF-ligand to active recruited kinase domains in EGFR-KDD (i.e. EGF-mediated dimerization of two ECDs results in an effective tetramer of intracellular kinase domains with potentially 2–3 active TKDs, versus typical oncogenic activation with 1–2 active TKDs). Alternatively, it may be that the EGFR-KDD inter-molecular dimer forms a more favorable interface than other oncogenic mutants, thus resulting in increased dimerization and activity. A combination of factors likely contributes to the overall increase in phosphorylation that we observe. Additional studies are needed to characterize the EGFR-KDD inter-molecular dimer.

Through a combination of biochemical and biophysical methods, we also determined that EGF-ligand stimulation induces formation of catalytically active homo- and hetero- inter-

molecular dimers and multimers. Critically, this demonstrates that EGFR-KDD retains the ability to activate other *ERBB* family members. This has important implications for the therapeutic management of patients whose tumors harbor EGFR-KDD. Indeed, we found neither cetuximab nor afatinib alone were able to completely ablate EGFR-KDD phosphorylation. We demonstrate, however, that cetuximab can be used to potentiate afatinib inhibitory activity for greater overall inhibition. We suspect that this is because of the synergistic mechanisms of the two drugs: cetuximab disassembles dimers and removes the ability of EGFR-KDD to activate other *ERBB* kinases, and afatinib inhibits the active intra-molecular dimer EGFR-KDD. It has been well-recognized that cetuximab induces degradation of EGFR mutants in different NSCLC cells[56,57]. In this study, no degradation of EGFR-Ex19Del, L858R and EGFR-KDD levels were observed in YAMC (Fig. 5a) and NR6 cells (Supplementary Fig. 4b), probably due to the shorter treatment time than previous studies (4 h versus 24–72 h)[56,57].

Finally, our computational and biochemical insights raise important considerations for the use of EGFR-KDD as a research tool. Whereas the inactive form of EGFR can be readily studied by the introduction of inter-molecular dimer-disrupting interface mutations[21], controlling the active fraction of EGFR in vitro has typically required introduction of known oncogenic point mutations or stimulation with EGF-ligand. The former causes well-documented perturbations to enzyme kinetics[58–61], while recent literature has demonstrated that the latter can influence EGFR multimerization and phosphorylation in a concentration-dependent manner[31]. Moreover, dimerization and activation of EGFR oncogenic missense mutants is dependent on protein concentration[62] and/or EGF-ligand stimulation[48]. EGFR-KDD provides a model of a fully active EGFR dimer in an EGF-independent setting, and may provide a more native-like control than kinase domain missense mutants without the complexity of concentration-dependent signaling effects.

Kinase domain duplications (KDDs) represent a novel form of activation for oncogenic kinases via a mechanism of constitutive dimerization. In this study, we have systematically characterized the fundamental biochemical and biophysical features of a prototypical KDD, EGFR-KDD. Subsequently, we identified potential treatment strategies in pre-clinical models of EGFR-KDD-mediated disease. This represents a comprehensive mechanistic and pre-clinical evaluation of treatment strategies specifically for a KDD-mediated disease. We anticipate that our results will also be used to inform additional studies on kinase duplication domains.

## Methods

**Cell culture, reagents, and transfection.** Ba/F3 cells were purchased from DSMZ. NR6 cells were a kind gift from Dr. William Pao[63]. YAMC EGFR[−/−] cells were a kind gift from Dr. Robert H. Whitehead[23]. Plat-GP cells were purchased from CellBioLabs. HEK293 cells were purchased from ATCC. Ba/F3 cells were maintained in RPMI 1640 medium (Mediatech, Inc.) supplemented with 1 ng/mL murine IL3 (Gibco, Life Technologies). NR6 cells were maintained in DMEM (Gibco). The Plat-GP cell line was cultured in full DMEM with selection of 1 µg/mL blasticidin (Gibco). YAMC cells were cultured as previously described[23,64]. COS-7 cells were cultured in DMEM (Calsson Lab, Smithfield, UT). All media were supplemented with 10% heat-inactivated FBS (Gibco) and penicillin–streptomycin (Gibco) to final concentrations of 100 U/mL and 100 µg/mL, respectively. All cell lines were maintained in a humidified incubator with 5% CO$_2$ at 37 °C (33 °C for YAMC cells[64]) and routinely evaluated for mycoplasma contamination.

Cetuximab was purchased from Bristol-Myers Squibb (Princeton, NJ). mAb806 is produced and purified in the Biological Production Facility (Ludwig Institute for Cancer Research, Melbourne)[65,66]. Transient transfection for expression in HEK293 cells was carried out using Lipofectamine 2000 (Invitrogen) according to the manufacturer's instructions. A total of 0.45 µg of each expression plasmid was used per well in six-well plates. To assess ligand-dependent EGFR activation, cells were serum starved overnight and treated with 50 ng/mL EGF for 5 min.

For PIE-FCCS experiments, COS-7 cells were transiently transfected 24 h before the experiment using Lipofectamine 2000 (Invitrogen). A total of 5 µg DNA (1:1

ratio of mCherry-tagged and eGFP-tagged plasmids mixture) was used per 35 mm MatTek plate (MatTek Corporation, Ashland, MA) to express both fluorescent-tagged species evenly and acquire the local density of 100–2000 receptors/µm². The media was changed to Opti-MEM I Reduced Serum Medium without phenol red (Thermo Fisher Scientific) before placing the plate in the on-stage incubator (37 °C) for FCCS measurement. Measurements were taken for both ligand-free and ligand-stimulated state of each construct, with 2 µg/mL recombinant human EGF (Sigma Aldrich, St. Louis, MO) or NRG1 (R&D Systems, Inc., Minneapolis, MI) as the ligand.

**Plasmid construction.** Generation of *EGFR*-KDD, *EGFR*-WT, and *EGFR*-L858R constructs was described previously[7]. EGFR-KDD mutations were constructed by using multisite-directed mutagenesis (Agilent) on the pMa-EGFR-KDD plasmid per the manufacturer's recommendations—with the exception of extension time being set at 1.5 min/kb. To specifically introduce mutations into each TKD due to the presence of two identical TKDs at the genomic level, after bi-directional dideoxy sequencing, pMa-EGFR-KDD-mutants were digested with ClaI and recombined with other pMa-EGFR-KDD fragments to create all single mutants: ClaI digests mutated pMa-EGFR-KDD plasmid were recombined with a ClaI–ClaI segments from unmutated pMa-EGFR-KDD and/or ClaI digests of unmutated pMa-EGFR-KDD plasmid were recombined with ClaI–ClaI segments from mutated pMa-EGFR-KDD. pMa-EGFR-KDD mutants were then subcloned to the pMSCV vector by HpaI digest and then subcloned to pcDNA3.1(−) vector by XhoI/HindIII digest. All plasmids were verified in the forward and reverse directions by Sanger sequencing. To obtain V5-epitope-tagged EGFR-KDD, we used PCR to add AgeI to the 3′ end of EGFR-KDD fragment by using pMSCV-EGFR-KDD as template, then EGFR-KDD fragment was inserted into pcDNA6-V5 HisB vector by using SnaBI and XhoI. To obtain Myc-epitope-tagged EGFR-KDD, the EGFR-KDD fragment was subcloned to pEF4Myc-HisB vector by using MfeI and XhoI. pcDNA6-EGFR-WT with Myc epitope tag was purchased from Addgene (#42665). V5-epitope-tagged *HER2* and *HER3* were kind gift from Dr. Carlos L. Arteaga[67]. For PIE-FCCS experiments, EGFR-WT, HER2, and HER3 were subcloned to eGFP-N1 and mCherry-N1 vectors by XhoI and AgeI digests. EGFR-KDD was subcloned to eGFP-N2 and mCherry-N2 vectors by using SnaBI and XhoI digests. For V5-tagged epitope EGFR-WT, we replaced the eGFP fragment of pEGFR-N1-EGFR-WT by V5-tagged epitope. In this study, for EGFR mutations, we utilized codon numbering of the human immature EGFR sequence that includes the 24-residue signal sequence (Supplementary Table 1). Primers used in this study were listed in Supplementary Table 5.

**Generation of stable cell lines.** Constructs of pMSCV, *EGFR*-WT, *EGFR*-L858R, *EGFR*-KDD, and *EGFR*-KDD-I706Q, D837N, V948R, I1057Q, D1188N, and V1299R mutations were introduced into NR6 and YAMC cells separately by retroviral transduction system as described previously[7]. Construct of EGFR Ex19Del (E746_A750del) was stably introduced into YAMC cells, and constructs of EGFR Ex19Del (E746_A750del), EGFR-L858R, and EGFR-KDD were stably introduced into Ba/F3 cells as described previously[68].

**Immunoblotting and antibodies.** For immunoblotting, cells were washed in cold PBS, and lysed in RIPA buffer (150 mmol/L NaCl, 1% Triton-X-100, 0.5% Na-deoxycholate, 0.1% SDS, 50 mmol/L Tris–HCl, pH 8.0) with freshly added 40 mmol/L NaF, 1 mmol/L Na$_3$VO$_4$, and protease inhibitor (Thermo Fisher Scientific, Waltham, MA). Lysates were quantified by Bradford assay in SmartSpec Plus Spectrophotometer (Bio-Rad, Hercules, CA) following the manufacturer's instructions. Lysates were subjected to SDS–PAGE followed by blotting with the indicated antibodies and detection by Western Lightning ECL reagent (Perkin Elmer, Waltham, MA). The densitometry for both phosphotyrosine content at Y1068 and total EGFR expression was quantified by ImageJ Software. The ratio of phosphotyrosine to total EGFR expression for each construct relative to EGFR-KDD was calculated. All immunoblotting experiments were performed three independent times and one representative replicate was shown in the manuscript. Raw, uncropped, and unprocessed scans of all blots as well as quantifications and standard deviations were included in the Source Data file.

For co-immunoprecipitation experiments, cells were washed in cold PBS and lysed in hypotonic buffer (20 mM HEPES pH7.5, 10 mM KCl, 1 mM EDTA, 1 mM EGTA, 1 mM mgCl$_2$, 0.1% NP-40, EDTA-free Protease Inhibitor Cocktail (Sigma-Aldrich # 04693159001)). The lysates were supplemented with 150 mM NaCl before centrifuging. Protein G Dynabeads (# 10004D, Life Technologies, Carlsbad, CA) were incubated with the primary antibody for 30 min at room temperature. Lysates were then added and incubated for 3 h at 4 °C. Immobilized beads were washed three times with hypotonic buffer supplemented with 0.65 M NaCl. 2×SDS loading buffer was added to the beads and then used for immunoblotting analysis. All co-immunoprecipitation experiments were performed two independent times and one representative replicate was shown in the manuscript.

**Antibodies.** Antibodies used included: EGFR (1:2000, #4267), phospho-EGFR (Y992) (1:1000, #2235), phospho-EGFR (Y1068) (1:1000, #2234), phospho-EGFR (Y1173) (1:1000, #4407) (For EGFR phosphorylation sites, we utilized codon numbering of mature EGFR sequence that does not include the 24-residue signal

sequence, Supplementary Table 1), horseradish peroxidase (HRP)-conjugated anti-mouse (1:5000, #7076), and HRP-conjugated anti-rabbit (1:5000, #7074) (Cell Signaling, Beverly, MA); V5 (1:5000, MCA1360GA, AbD Serotec), Myc (1:2500, Sigma-Aldrich A5963); actin antibody (1:5000, Sigma-Aldrich A2066).

**Pulsed interleaved excitation fluorescence cross-correlation spectroscopy**. FCCS data were taken on a customized inverted microscope setup coupled with pulsed interleaved excitation and time-correlated single photon detection as described in previous works[29,37]. A supercontinuum pulsed laser (9.2 MHz repetition rate, SuperK NKT Photonics, Birkerød, Denmark) was split into two beams of 488 and 561 nm through a series of filters and mirrors for the excitation of eGFP and mCherry, respectively. The beams were directed through two different-length single mode optical fiber to introduce 50 ns time delay for pulsed interleaved excitation to eliminate possible spectral crosstalk[69]. The beams were overlapped before entering the microscope through a dichroic beam splitter (LM01-503-25, Semrock) and a customized filter block (zt488/561rpc, zet488/561m, Chroma Technology). A ×100 TIRF oil objective (Nikon, Tokyo, Japan) was used for the excitation beam focus and fluorescence emission collection. A short fluorescently tagged DNA fragment was used to verify the alignment of the system, including the confocal volume overlap. Negative and/or positive controls (Supplementary Fig. 5a, b) were tested regularly prior to the experimental samples for comparisons of the fit parameters. The excitation beams were focused to the peripheral membrane of the cell to allow the fluorescence measurements of only the membrane-bound receptors. Data were only taken on the flat, peripheral membrane area, where the distance between the basal and apical membranes were within a few hundred nanometers, to avoid inclusion of fluorescence from cytosolic organelles or vesicles. For each cell, one area of the membrane was selected for data collection. Six 10-s acquisitions were taken per area. The fluorescence signal was collected through a home-built confocal detection unit with a 50 μm confocal pinhole and dichroic beam splitter (LM01-503-25, Semrock, Rochester, NY). The two signals were filtered (91032, Chroma Technology Corp., Bellows Falls, VT; zt488/561rpc and zet488/561 m, Chroma Technology Corp., Bellows Falls, VT) and then focused independently on to single-photon avalanche diodes (Micro Photon Devices, Bolzano, Italy). The photon counts were recorded by a time-correlated single photon counting module (Picoharp 300, PicoQuant, Berlin, Germany). For analysis, the time-tagged photon data were gated to isolate photons that arrived within 40 ns after each laser pulse arrival time. Then we calculated auto-correlation and cross-correlation curves corresponding to each species using our custom MATLAB script. Curves of six consecutive acquisitions per area were averaged then fitted to a single component, 2D diffusion model as described in previous works[37,38,69].

The auto-correlation curves contain two types of decay. The first decay is due to the photophysical activity, such as triplet relaxation or blinking. The second decay indicates the average dwell time ($\tau_D$), which is used to calculate the effective diffusion coefficient using $D_{eff} = \omega_o^2/4\tau_D$. The amplitude of the correlation curves indicates local concentration of the diffusing receptors. Using the cross-correlation curve (Supplementary Fig. 5), we can calculate cross-correlation values, or fraction correlated ($f_c$) values that indicate the degree of oligomerization. For an ideal system undergoing on dimerization, the $f_c$ value varies from 0 to 1, with 0 indicating the system is monomeric and 1 indicating complete dimerization. For real systems, effects like photostability, interaction statistics, and relative expression levels drop the expected $f_c$ value for dimerization into the range of 0.10–0.15 for a monomer–dimer equilibrium. For higher order oligomerization the $f_c$ values will increase, allowing us to compare the degree of oligomerization for more complex systems[69].

**Anchorage-independent assays and cell viability assay**. Anchorage-independent assays were performed using modified protocols[70,71]. For the bottom layer of agar, 1.5 mL of a 1:1 mix of 1.0% agar (prepared in 1× PBS) and media were plated in each well of six-well plate. For the upper layer of agar, 1.5 mL of a 1:1 mix of 0.6% agar (prepared in 1× PBS) and media containing 5000 cells was plated into each well of six-well plate. Colonies were counted using GelCount (Oxford Optronix) with identical acquisition and analysis settings. Cell viability assay was performed on IL3-independent Ba/F3 cells stably expressing EGFR-KDD, Ex19Del, and L858R by using CellTiter-Blue® Cell Viability Assay (#G8080, Promega, Madison, WI) following manufacturer's instructions. Three days after incubation, CellTiter-Blue Reagent was added, and the fluorescence was detected at $560_{EX}/590_{EM}$ with a Synergy HTX microplate reader (BioTek Instruments, Winooski, VT, USA). All experiments of anchorage-independent assays and cell viability assay were performed three independent times in triplicate, and one representative replicate was shown in the manuscript.

**Molecular modeling**. Previously, we performed de novo loop modeling to determine a geometrically plausible model of the EGFR-KDD linker region[7]. Here, an all-atom structural model of the EGFR-KDD intracellular domain was generated with RosettaCM[72] with the active EGFR WT dimer PDB ID 2GS6 as the base template. Missing density in the β3-αC region was templated with PDB ID 2ITX. The N- and C- termini of the donor and receiver kinases of the EGFR-KDD intramolecular dimer, respectively, as well as the connecting linker region, are based on three templates: the previously modeled linker region from Gallant et al. 2015[7]; the

JMB domain of PDB ID 4RIW; and the JMB domain of PDB ID 3GOP. Missing residues are modeled de novo with RosettaCM fragment insertion. Three rounds of comparative modeling were performed. After rounds two and three, the best scoring models with varying RMSDs from the lowest scoring model in each round were selected as additional starting templates for the next round. After the third round, distance-based clustering of the linker region identified three low-energy clusters. The best scoring model from each cluster was refined with a 1 μs MD simulation in Amber18[73]. The final EGFR-KDD model and EGFR-WT homodimer subsequently each underwent 1 μs MD simulations.

Models were solvated in a rectangular box of SPC/E explicit solvent neutralized with monovalent anions. Protein was buffered on all sides with 12 Å solvent. Solvent and ions were minimized with 500 steps steepest gradient descent followed by 1000 steps of conjugate gradient descent while protein atoms were restrained with a force constant of 10.0 kcal/mol/A². The protein was then minimized for 200 steps steepest gradient descent followed by 800 steps of conjugate gradient descent in buffer restrained with a force constant of 5.0 kcal/mol/A². Finally, restraints were removed from the system for 100 additional steps of steepest gradient descent followed by 900 steps of conjugate gradient descent minimization.

Post-minimization, SHAKE was implemented to constrain covalent bonds to hydrogen atoms. Systems were slowly heated in NVT ensemble to 100 K over 50 ps with a 1 fs timestep. Subsequently, systems were heated in NPT ensemble at 1 bar with isotropic position scaling from 100 to 300 K over 500 ps and 1 fs timestep. Equilibration/production simulations were run in the NPT ensemble at 300 K with a Monte Carlo barostat. Temperature was controlled using Langevin dynamics with a collision frequency of 1 ps⁻¹ and a unique random seed for each simulation. Periodic boundary conditions were imposed on the system throughout heating and equilibration. Electrostatics were evaluated using the particle mesh Ewald (PME) method and a distance cutoff of 8.0 Å. A 2 fs integration timestep was employed during production simulations. All RMSD and RMSF calculations were performed with CPPTRAJ[74].

Approximations of the linker interaction energies of the top three EGFR-KDD clusters were performed with the single-trajectory molecular mechanics/ generalized Born solvent-accessible surface area (MM-GBSA) method as implemented in MMPBSA.py[75]. GBSA was calculated with the OBCII generalized born solvent model with a surface tension of 0.0072 kcal/mol/Å² and salt concentration of 0.15 M, and nonpolar contributions to the solvation free energy were computed with the LCPO method. Entropic contributions to binding were neglected. The final reported values are averaged over frames collected every 100 ps.

**Kinase domain duplication detection from foundation medicine and MSK-IMPACT datasets**. For the Foundation Medicine dataset, a minimum of 50 ng of DNA was extracted from formalin-fixed paraffin-embedded sections and comprehensive genomic profiling was performed on hybridization-captured, adaptor ligation-based libraries to a median exon coverage depth of >500× for all coding exons of 315 (FoundationOne®, $n = 152,674$), or 324 (FoundationOneCDx®, $n = 86,824$) cancer-related genes plus selected introns from genes frequently rearranged in cancer to identify base substitutions, small insertions or deletions, copy number alterations (focal amplifications and homozygous deletions), and rearrangements, as previously described[76]. Testing was performed in a Clinical Laboratory Improvement Amendments-certified, College of American Pathologists-accredited reference laboratory (Foundation Medicine, Cambridge, MA). We interrogated the Foundation Medicine dataset of $n = 239,498$ consecutive unique solid tumor specimens for KDD in *EGFR*, *ERBB2*, *ERBB3*, and *ERBB4*. These rearrangement duplications were detected by clustering chimeric and semi-mapped paired-end reads within each gene of interest and mapping breakpoints onto the hg19 reference genome assembly, as previously described[76]. A KDD was therein defined as a large genomic duplication where breakpoints both flanked and did not disrupt the region corresponding to the respective gene's kinase domain. Statistical enrichment including $P$-value and odds-ratio (OR) were calculated using Fisher's exact testing. For a detailed description of the FMI data analysis pipeline, please see the Supplementary Methods section in the Supporting Information file. Approval for this study, including a waiver of informed consent and a Health Insurance Portability and Accountability Act waiver of authorization, was obtained from the Western Institutional Review Board (protocol no. 20152817). This is retrospective research that involves no more than a minimal risk to the privacy of patients and involves no intervention or contact with the patients. FMI provides FMI Tests at the request of treating physicians and therefore has no direct relationship with any of these patients. Moreover, in many cases the patients may no longer be associated with the treating physician who ordered their FMI Test or may be deceased, and therefore it may be impossible to contact these patients.

Identification of KDDs from the MSK-IMPACT dataset is a re-analysis of published data[18]. MSK-IMPACT-sequencing data from patients whose tumor and matched normal samples were prospectively sequenced between January 2014 and September 2019 ($n = 40,165$, NCT01775072) were used in this study. Structural variant detection was performed on the paired-end reads using Delly (version 0.7.5; https://github.com/dellytools/delly). Duplication events that surrounded or overlapped known kinase domains were selected for further manual review. For copy number-based analysis, coverage data from the tumor and an unmatched normal sample were used to generate a fold change value[77] for each exon in a

kinase gene. Using *k*-mean clustering ($k = 2$), we identified samples where one of the clusters was overlapping (requiring at least 70% of the kinase domain to be involved) or encompassing the kinase domain with a median cluster fold change difference of at least 0.4. We combined the two datasets for further manual review to identify a subset of confident KDD calls.

**Statistical analysis**. Statistical significance was analyzed using one-way ANOVA with Bonferroni or uncorrected Fisher's least significant difference (LSD) post hoc test. Results were displayed as mean values or median values ± standard deviation (SD) or standard error of the mean (SEM). For all tests, the criteria for significance were nonsignificant (ns), $P < 0.05$ (*), $P < 0.01$ (**), $P < 0.001$ (***), and $P < 0.0001$ (****). Statistical analysis was carried out using Prism 9 (GraphPad Software).

**Reporting summary**. Further information on research design is available in the Nature Research Reporting Summary linked to this article.

## Data availability

The authors declare that all data supporting the findings of this study are available within the article and its supplementary information files or from the corresponding author upon request. The results underlying Table 1 are based on a combination of genomic sequencing data from Foundation Medicine Inc. and MSK-IMPACT. In accordance with the Health Insurance Portability and Accountability Act, in an effort to minimize the risk of re-identification of individuals, individual-level data are not publicly available. For the Foundation Medicine dataset, raw sequencing data are proprietary and not publicly available. However, requests from accredited researchers for access to de-identified individual-level or aggregate data relevant to this manuscript, such as tumor type and mutational status, can be made available upon request by contacting Dr. Alexa B. Schrock at aschrock@foundationmedicine.com. Accredited researchers should provide contact information, affiliation/organization, and research rationale. The analysis presented here from the MSK-IMPACT dataset is a re-analysis of data originally reported by Zehir and coworkers[18]. The MSK-IMPACT dataset is publicly available through the cBioPortal for Cancer Genomics (http://cbioportal.org/msk-impact). MSK-IMPACT KDD data can be made available upon request. Protein Data Bank (PDB) identifiers 2GS6, 2ITX, 3GOP, and 4RIW were accessed to assist with model building for this study. In addition, Dataset 1 from the Supplementary Information of Needham and colleagues[31] was accessed to assist with model building of the EGFR-KDD intermolecular dimer. Source data are provided with this paper.

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

## Acknowledgements

This work is supported through NIH NCI R01CA227833 and NIH NIGMS R01 GM080403. C.M.L. was also supported in part through P30CA086485, UG1CA233259, and 5P01CA129243-12. Z.D. is supported by the 2018 AACR-AstraZeneca Lung Cancer Research Fellowship, Grant Number 18-40-12-DU. Work in the Meiler laboratory is also supported through the NIH (R01GM080403, R01GM099842, and R01GM073151). A.W.S. and S.K. are supported through the National Science Foundation (CHE-1753060). B.P.B. is supported through the NIH by a Ruth L. Kirschstein NRSA fellowship (F30DK118774). Work at MSKCC is supported in part by the NCI Cancer Center Support Grant/Core Grant (P30CA008748). We sincerely thank John Kuriyan, Portia L. Thomas, and Wade T. Iams for their productive conversations, comments, and suggestions for this manuscript.

## Author contributions

Z.D., B.P.B., J.-N.G., J.M., M.R.-B., and C.M.L. conceived and designed the study. Z.D. constructed mutations within linker of EGFR-KDD and performed most of the in vitro cell-based assays. B.P.B and J.M. designed the computational structural modeling experiments, and B.P.B. performed the structural modeling. S.K. and A.W.S. designed the PIE-FCCS experiments, and S.K. performed PIE-FCCS. D.C.P., S.M.A., and A.B.S. acquired and analyzed the Foundation Medicine dataset. D.F., G.J., R.B., A.Z., and M.L. acquired and analyzed the MSKCC dataset. J.-N.G. constructed the dimer interface mutations and stably expressed them into NR6 and YAMC cells. Z.D. and Y.-K.Z. performed cell viability assay. Y.Y. maintained all the stocks of cells, DNA and reagents and provided technical support. Z.D., B.P.B., J.M., and C.M.L. wrote the manuscript. All authors reviewed the manuscript.

## Competing interests

D.C.P., S.M.A., and A.B.S. are employees of Foundation Medicine, Inc. C.M.L. is a consultant/advisory board member for Pfizer, Novartis, Astra Zeneca, Genoptix, Genentech, Syros, Cepheid, Roche, Sequenom, Ariad, Takeda, Foundation Medicine, Blueprints Medicine, Achilles, Amgen and reports receiving commercial research grants from Xcovery and Novartis. M.L. has received advisory board compensation from Merck, Lilly Oncology, AstraZeneca, Bristol-Myers Squibb, Takeda, and Bayer, and research support from LOXO Oncology, Helsinn Healthcare, Elevation Oncology and Merus. No potential conflicts of interest were disclosed by the other authors.
