## [Peer Review File · Nature Communications]

REVIEWER COMMENTS

Reviewer #1 (Remarks to the Author):

Following up on work that they published for EGFR in 2015, Du et al. describe the identification of kinase domain duplication (KDD) in EGFR, ERBB2, ERBB3, and ERBB4 in 443, 217, 92, and 47 cases from 237,701 tumor samples – with most common occurrence in glioma. They then proceed to introduce mutations into a KDD variant of EGFR to test hypotheses about how the kinase domain is activated (and which is active). Their data suggest that intramolecular formation of the asymmetric kinase domain dimer with TKD2 as receiver is important for constitutive activity – not unexpectedly. Moreover, a kinase dead mutation in TKD2 impairs constitutive activity – suggesting the same thing (as diagrammed in Figure 1b). To ask what happens in the JM domain, the authors use MD simulations. They also replace JMB with a (GGG)_n sequence, and find (as Hristova had in the wild-type EGF-activated receptor) that this reduces activity. In the model (Figure 2f), L1039 and P1042 seem to be positioned similarly with respect to Y813 as seen in the EGFR TK dimer including this region (PDB 3GOP), and mutating the leucines reduces activity (as for normal dimers).

The authors next analyze the response of KDD to EGF – which does activate it. For EGF-induced activation, mutations suggest that TKD2 is now activator, and TKD1 receiver in intermolecular kinase dimers – activating TKD1 more robustly than is seen for TKD2 intramolecularly. As might be expected, the authors also show that KDD also heterodimerizes with WT EGFR following EGF binding, and maintains HER2/3/4 heterodimerization capacity. Given the role of dimerization as well (of course) as kinase activity, the authors finally surmise that combining cetuximab and TKIs will be beneficial – as it is in many cases involving mutationally activated EGFR. The final data slide supports this – but the authors do not provide any evidence to suggest that this combination is more advantageous here than it is for L858R for example (see PMID 24063894). The absence of this quantitative comparison makes the punchline fall a little flat, since similar antibody/TKI combinations are being tried (including in clinical trials) for a variety of mutated EGFR settings.

Specific Comments:

1. The authors should be careful with their protein sequence numbering in the manuscript – which they have confused. They refer to the EGFR precursor numbers when discussing mutations on page 4. They then switch to mature protein numbers (which are 24 less – accounting for the signal sequence) when discussing C-terminal tyrosines (Y992, Y1068 and Y1173 are actually Y1016, Y1092, and Y1197 in the scheme used elsewhere in the paper). I'm not sure which they have used in the second kinase domain. The authors should pick one numbering scheme, state which they have chosen, and stick with it – else things get very confusing.

2. How do the JMB residues compare (in the model) in location to those seen in the Red-Brewer crystal structure (PDB 3GOP) – where they do form “enthalpically stabilizing contacts” between kinase domains? It seems odd not to have any comparison of this. How does the linker length affect the structure here? This was not clear from the analysis. And the only comparison is with a model – when there is a crystal structure of WT including this region.

3. It would be nice to see some experimental structural analysis of the KDD.

4. The authors argue that the cetuximab results in Figure 3 show that EGF stimulation promotes KDD activity through formation of intermolecular dimers – but they really do not. It’s likely, but the main function of cetuximab is to block EGF binding (see ref 34) – which is a different facet.

5. It is not clear that the model in Figure 3d has any support. Is it needed? It seems clear that 3c can happen, and there is really no evidence in the literature for activation (of TKD3) in a side-by-side manner, so 3d seems very speculative (and probably should be deleted).

6. Why are JMB residues of TKD3 not interacting with the C-lobe of TKD2 in the model in Figure 3c. They surely will. Indeed, such a model can be generated just by looking at the (experimentally seen) crystal packing in PDB entry 3GOP.

7. The authors do not discuss why EGF-activated KDD is so much more robust than constitutive. This presumably reflects the fact that the asymmetric dimer forms more ideally intermolecularly – with TKD2 as activator and TKD1’ (or TKD3) as receiver – than intramolecularly (with TKD2 as receiver and TKD1 as activator). This is not reflected in Figure 3c, but is strongly supported by looking at crystal packing as in point 6.

8. It would be nice to see some experiments studying KDD heterodimerization with other ERBB/HERs when all expressed at cancer-relevant levels (rather than in the transient expressions used here, which rather force the system).

9. In Figure 4e, did the authors try adding EGF and NRG1? They comment that heterodimerization of KDD with HER3 was seen for NRG1 but not EGF. This seems also to be true for WT EGFR here – so this sentence might be rephrased?

10. Did the authors undertake any studies to determine whether KDD versions of HER2,3,4 are constitutively active?

11. One important point is that the individual L1038 and L1039 mutations do not cause any reduction in phosphorylation – only mutating both does this in Figure 2g. It might be worth pointing out that the LL here is a consequence of the fusion (the real JMB sequence is ELVEPLTPS. This might warrant some discussion.

12. Why no phospho-receptor blots in Figure 4a and c?

Reviewer #2 (Remarks to the Author):

This study of kinase domain duplicate EGFR (EGFR-KDD) is comprehensive and convincing in demonstrating the molecular mechanism underlying the mutant's aberrant activity. It showed that the activity of EGFR-KDD, like the WT, is dependent on the formation of the asymmetric kinase domain dimer and the juxtamembrane segments remain important. The simulation study of the EGFR-KDD kinase domain dimer is well executed and clearly represented. The authors' model is consistent with the model reported by Arkhipov et al., Cell, 2012, in terms of how the juxtamembrane segment of the receiver interacts with the donor in activation. (This should be most obvious from the structural coordinates of EGFR active dimer model as SI material of the paper.) The other very interesting part of this work concerns the oligomerization of the kinase domains. The authors showed that likely upon EGF binding EGFR-KDD forms dimers, within which the kinase domains form two asymmetric dimers in contact with one another. Here this reviewer would like to suggest one arrangement that the authors may have not considered but fits their experimental data well. In this EGFR-KDD dimer model, the KD1 of one EGFR-KDD forms an asymmetric dimer with the KD2 of the other EGFR-KDD as receiver. This would explain why KD2-dead EGFR KDD can be activated by EGF-induced dimerization. The authors' findings suggest the EGFR-KDD may be a valuable system for investigation of the intracellular arrangement of EGF-induced EGFR oligomers, the details of which remain unclear. The authors should discuss this possibility in the Discussion section.

Reviewer #3 (Remarks to the Author):

Regarding the FCCS studies, the quantitative interpretation of the data should be improved.

1) Homo-interactions are quantified using FCCS and labeling the same protein with either mCherry or GFP. This can be deduced but it is not clearly explained. I had also trouble deducing (for hetero-interactions) which construct was labeled with GFP and which with mCherry.

2) The authors use the words dimerization and multimerization. That implies a quite precise quantification of the multimeric state of the receptor (and stoichiometries) which, in the current state of the manuscript, appears doubtful. I understand that similar measurements were performed already few years ago and the authors refer to them when, for example, mentioning controls (Ref. 48). Nevertheless, I believe that a positive control (definitely) and a negative control (this is maybe less needed) should be included in these experiments. In my experience, the maximal detectable CC varies during time (due to e.g. misalignment, glass slides with slightly different thickness, dirt on the objective, wearing of optics) and that is why we perform positive controls every day. Finally, the use of a positive control helps surely in the case of a 1-to-1 interaction, but this case the analysis is also more complicated because other stoichiometries cannot be excluded (see below).

3) The lack of positive controls for hetero-dimers makes quantitative interpretation difficult. For example (lines 307-317), WT X HER2 and KDD x HER2. EGF increases f_c to 0.1 or 0.16. What does that imply? That the amount of hetero-dimers increases? Does 0.16 indicate more hetero-dimers than 0.1? Or maybe the presence of more abundant high-order hetero-multimers? The authors say that these values of f_c indicate that the proteins "heterodimerize", but it might be more complex than that. Once more, in my opinion, this conclusion would need positive controls for GFP-mCherry and the absence of complexes with different stoichiometries.

Also, f_c of 0.06 appears to mean absence of heterodimerization. But how quantitative is that? If 0.16 is e.g. 100% heterodimers, is 0.06 33% heterodimers and the rest are monomers?

Another example: KDD heterodimerizes with HER3 in the presence of NRG1 but not EGF. A quite strong assessment. But f_c is 0.14 in the presence of NRG1 and more than half of that (0.08) in the presence of EGF. Does that mean half so many heterodimers as with NRG1? If so, can we say that EGF does not cause heterodimerization?

Once more, as a reminder, the possible presence larger (hetero-) multimers with more complex stoichiometry would make the calculations much more complex, but this is not discussed so far.

3) The authors measure and report brightness values. This is actually valuable because more information regarding the average stoichiometry could be extracted (if needed! If this is not needed, that would be also fine, but then precise expressions such as "dimerization" should be avoided). Brightness values are reported in supplementary table 1. I would recommend clarifying which values refer to GFP or mCherry.

Of course, in this case, monomer and homo-dimer (!) controls are additionally needed to normalize the brightness values. Both for GFP and mCherry.

Especially in the case of mCherry, it is known that the protein has a low fluorescence/maturation probability and this affects the CC values that can be measured. That is why homo-dimer controls would also be required.

Looking at the brightness values, some interesting questions arise. For example, the brightness of WT (alone) increases with EGF (in accordance to the increase in CC) due probably to the presence of a small amount of dimers with the same fluorescent proteins, in addition to dimers with different fluorescent proteins. But this does not happen with KDD. Does this indicate different complex stoichiometries between the two proteins?

In the case of WT X HER2, the brightness of the WT increases in the presence of EGF (while that of HER2 does not). Might this indicate the presence of WT multimers bound to HER2 monomers? Also, the brightness of KDD increases (w/o EGF), compared to WT. And it increases even more in the presence of EGF (while HER2 brightness remains again constant). Is this data worth discussing maybe?

Finally, as mentioned above, the authors mention the presence of "heterodimers² between KDD and HER3 in the presence of NRG1 but not EGF. This is motivated by the strong increase in fc. Nevertheless, the brightness of WT increases significantly in the presence of EGF, but not NRG1. The same happens with WT and HER3. Might this be relevant and indicate, e.g., the presence of hetero-multimers with different stoichiometries, rather than simply dimers?

4) "Part numbers" should be corrected

Response to Reviewer Queries

Reviewer #1 (Reviewer Comments to the Author)

Following up on work that they published for EGFR in 2015, Du et al. describe the identification of kinase domain duplication (KDD) in EGFR, ERBB2, ERBB3, and ERBB4 in 443, 217, 92, and 47 cases from 237,701 tumor samples – with most common occurrence in glioma. They then proceed to introduce mutations into a KDD variant of EGFR to test hypotheses about how the kinase domain is activated (and which is active). Their data suggest that intramolecular formation of the asymmetric kinase domain dimer with TKD2 as receiver is important for constitutive activity – not unexpectedly. Moreover, a kinase dead mutation in TKD2 impairs constitutive activity – suggesting the same thing (as diagrammed in Fig. 1b). To ask what happens in the JM domain, the authors use MD simulations. They also replace JMB with a (GGG)_n sequence, and find (as Hristova had in the wild-type EGF-activated receptor) that this reduces activity. In the model (Fig. 2f), L1039 and P1042 seem to be positioned similarly with respect to Y813 as seen in the EGFR TK dimer including this region (PDB 3GOP), and mutating the leucines reduces activity (as for normal dimers).

The authors next analyze the response of KDD to EGF – which does activate it. For EGF-induced activation, mutations suggest that TKD2 is now activator, and TKD1 receiver in intermolecular kinase dimers – activating TKD1 more robustly than is seen for TKD2 intramolecularly. As might be expected, the authors also show that KDD also heterodimerizes with WT EGFR following EGF binding, and maintains HER2/3/4 heterodimerization capacity. Given the role of dimerization as well (of course) as kinase activity, the authors finally surmise that combining cetuximab and TKIs will be beneficial – as it is in many cases involving mutationally activated EGFR. The final data slide supports this – but the authors do not provide any evidence to suggest that this combination is more advantageous here than it is for L858R for example (see PMID 24063894). The absence of this quantitative comparison makes the punchline fall a little flat, since similar antibody/TKI combinations are being tried (including in clinical trials) for a variety of mutated EGFR settings.

Authors' Response: We appreciate the Reviewer's acknowledgement that our data shown in Fig. 5 support the efficacy of a combination therapeutic strategy consisting of EGFR TKI (afatinib) and EGFR monoclonal antibody (cetuximab) against the EGFR-KDD variant. The foundational premise of our work is to understand the structural and biochemical mechanism whereby EGFR-KDD (or any EGFR oncogenic variant) activates the EGF receptor, in order to provide sound rationale for optimal therapeutic utilization of the many EGFR inhibitors (TKIs and monoclonal antibodies) that we currently have in clinical practice.

The Reviewer expressed concern that our evidence in favor of co-administering afatinib and cetuximab for EGFR-KDD “falls flat” since (1) antibody / TKI combinations are already being evaluated in clinic and (2) suggests comparing the efficacy of such a treatment on EGFR-KDD vs. L858R. We thank the Reviewer for these comments, and we address both of the comments below.

First, to address the co-administration of EGFR antibody / EGFR TKI combination approaches in the clinic. The Reviewer is correct to note that the combination of cetuximab with various EGFR TKIs, including gefitinib¹ and afatinib^{2,3}, has been tested in lung cancer patients. In a phase I trial, no responses were observed with the combination of cetuximab plus gefitinib¹, therefore, this combination has not subsequently been used in patients. By contrast, the combination of cetuximab plus afatinib has advanced in the clinic, including a phase I trial (NCT01090011) that included an expansion cohort^{2,3}. Results from this trial of cetuximab plus afatinib demonstrated that the combination therapy was effective in achieving tumor reduction (as assessed by CT scans using RECIST criteria) in patients with *EGFR*-mutant lung cancer. Notably, the combination was effective against both tumors harboring Ex19Del variants and tumors harboring L858R variants. In fact, the majority of patients in the trial (62%) had EGFR Ex19Del. These clinical data are in contrast to the pre-clinical data cited by the Reviewer⁴, where the pre-clinical studies suggested that the combination of cetuximab plus afatinib would not be efficacious against Ex19Del variants. It is also worth noting that cetuximab as a single agent has no efficacy in patients with *EGFR*-mutant lung cancer who have previously been treated with an EGFR TKI⁵. Finally, we note that the combination of cetuximab plus afatinib is not FDA-approved and not standardly used in the treatment of patients with EGFR Ex19Del or EGFR L858R mutation. The current standard of care for these patients is the mutant-selective EGFR TKI, osimertinib, based on a seminal phase 3

clinical trial^{6, 7}. Importantly, the development of osimertinib as an EGFR mutant-selective inhibitor – both in the lab and in the clinic – *was based on the unique biology of EGFR kinase domain mutations*, which lower ATP affinity⁸⁻¹¹, thereby sensitizing to TKIs, which are ATP mimetics. We postulate that the success of osimertinib was based on the same principles that we apply to our studies – that understanding the structural and biochemical mechanisms whereby EGFR variants (Ex19Del, L858R, KDD, etc.) activate the EGF receptor provides the necessary sound rationale for optimal therapeutic development. Therefore, we respectfully disagree with the reviewer that our evidence in favor of co-administering afatinib and cetuximab for EGFR-KDD “falls flat”, as this combination, though tested in clinical trials, is not the standard of care for patients with EGFR Ex19Del or EGFR L858R mutation containing tumors, and the biology of EGFR-KDD compared to the biology of EGFR Ex19Del and EGFR L858R is different, resulting in the need for different – yet rational - therapeutic strategies. Furthermore, we provide new experimental data (below and in our revised manuscript) that demonstrates that TKI alone is not as efficacious against EGFR-KDD vs. L858R and Ex19Del, and that the combination therapy (cetuximab plus afatinib) is advantageous in the context of the EGFR-KDD variant.

Second, to address the Reviewer’s comment regarding the efficacy of cetuximab plus afatinib on EGFR-KDD vs. L858R. We start by again emphasizing that our goal is to advance precision medicine by providing mechanistic insight into EGFR-KDD so that therapeutic decisions have a rational basis. The reviewer is correct that antibody/TKI (as well as chemo/TKI) combinations have been tried pre-clinically and clinically, as described above. Our goal is not to come up with a new treatment combination, but to help identify *when* such treatment strategies could be implemented to have the most positive impact on our patients based on science and detailed mechanistic understanding of drug targets.

This need for precision is highlighted by the manuscript the reviewer identified⁴. Cho and colleagues suggested that L858R is responsive to cetuximab treatment because it activates in a dimer-dependent manner, while Ex19Del (variant E746_A750>P) is not responsive to cetuximab because it activates in a dimer-independent manner⁴. In patients, however, we have not observed such an effect. Indeed, patients responded to neither cetuximab alone⁵ nor in combination with gefitinib¹. Combination treatment of cetuximab and afatinib did, however, result in a 29% response rate across both Ex19Del (variants unspecified) and L858R together², in opposition to the suggestion by Cho et al. that only an effect in L858R is expected. These inconsistencies suggest that our current expectations from clinical trials are built on incomplete understandings of mutant EGFR biology. It is therefore imperative that we perform detailed pre-clinical studies, such as we have done here for EGFR-KDD, to better inform trial design.

That being said, we agree with the reviewer that it would be beneficial to have a comparison of cetuximab/afatinib treatment in the context of EGFR-KDD vs. L858R. Therefore, we treated YAMC cells stably expressing EGFR Ex19Del (most common variant, E746_A750del), L858R, and EGFR-KDD with afatinib and cetuximab both in the absence and presence of EGF ligand (revised **Fig. 5a**, shown below to facilitate review and also included in the revised manuscript). Importantly, we see that in both the absence and presence of EGF, afatinib results in a near complete ablation of p-EGFR in Ex19Del (**lanes 1 vs. 2 and 5 vs. 6, Fig. 5a**) and L858R (**lanes 9 vs. 10 and 13 vs. 14, Fig. 5a**). In contrast, substantial residual p-EGFR exists in EGFR-KDD expressing cells treated with afatinib alone, particularly in the presence of EGF (**lanes 17 vs. 18 and 21 vs. 22, Fig. 5a**). Cetuximab alone inhibited Ex19Del, L858R and EGFR-KDD only in the presence of EGF ligand (**lane 7, 15, 23, Fig. 5a**). In the absence of EGF, cetuximab had minimal effect on Ex19Del and L858R activity (**lane 3, 11, Fig. 5a**), consistent with previous studies suggesting that some α C-helix-stabilizing EGFR mutants may form an “inside-out” (i.e. kinase domain-mediated) EGF-independent dimer with a reduced response to cetuximab^{4, 12, 13}. We speculate that such effects may also be in part due to concentration-dependent factors.

Notably, the greatest reduction in pEGFR in the presence of EGF for EGFR-KDD occurred with the combination of cetuximab plus afatinib (**lanes 21, 22, 23, 24, Fig. 5a**). While these results do not provide a quantitative (e.g. IC50) difference between Ex19Del, L858R, and EGFR-KDD relative sensitivities to the different treatments, these data do clearly show that EGFR-KDD is less sensitive to afatinib alone (compared to Ex19Del and L858R), and that combination treatment (cetuximab plus afatinib) is advantageous in EGFR-KDD. In contrast, the potential advantage of afatinib over cetuximab plus afatinib is less clear in Ex19Del and L858R. These results are directly supported by our other computational, biophysical, and biochemical

experiments, which all suggest that EGFR-KDD forms catalytically active intra-molecular dimers in the absence of EGF and increases activity upon stimulation with EGF to form inter-molecular dimers/multimers.

Revised Fig. 5a:

Fig. 5 | Inhibition of EGFR-KDD is maximally achieved by blocking both intra- and inter-molecular dimerization.

a, YAMC cells were serum starved for 12 hours and treated with afatinib (10 nM in serum-free medium) and cetuximab (10 µg/ml in serum-free medium) for 3 hours 45 minutes, and then were treated with EGF (50 ng/ml in serum-free medium) for 15 minutes. The cells were harvested and analyzed by Western blot.

- Revised text, page 8: “We treated YAMC cells stably expressing EGFR Ex19Del (E746_A750del), L858R, and EGFR-KDD with afatinib and cetuximab both in the absence and presence of EGF ligand (Fig. 5a). Importantly, we observed that in both the absence and presence of EGF, afatinib resulted in a near complete ablation of p-EGFR in Ex19Del (Fig. 5a, lanes 1, 2, 5, 6) and L858R (Fig. 5a, lanes 9, 10, 13, 14), but substantial residual phosphorylation existed in EGFR-KDD (Fig. 5a, lanes 17, 18, 21, 22). As expected, cetuximab alone inhibited Ex19Del, L858R and EGFR-KDD only in the presence of EGF ligand (Fig. 5a, lane 7, 15, 23). Notably, the greatest reduction of phosphorylation for EGFR-KDD occurred with the combination of cetuximab+afatinib in the presence of EGF (Fig. 5a, lanes 21, 22, 23, 24). These data suggested that phosphorylation of EGFR Ex19Del and L858R is abolished by afatinib (TKI) alone, and addition of cetuximab does not add anything to the decrease in auto-phosphorylation. Unlike EGFR Ex19Del and L858R, phosphorylation of EGFR-KDD is inhibited by both afatinib and cetuximab as single agent, but the combination treatment yielded more inhibitory effects.”

Specific comments:

1) The authors should be careful with their protein sequence numbering in the manuscript – which they have confused. They refer to the EGFR precursor numbers when discussing mutations on page 4. They then switch to mature protein numbers (which are 24 less – accounting for the signal sequence) when discussing C-terminal tyrosines (Y992, Y1068 and Y1173 are actually Y1016, Y1092, and Y1197 in the scheme used elsewhere in the paper). I’m not sure which they have used in the second kinase domain. The authors should pick one numbering scheme, state which they have chosen, and stick with it – else things get very confusing.

Authors’ Response: We apologize for the confusion and lack of clarity on the protein sequence numbering we are using throughout the original manuscript. For EGFR mutations, immature numbering of EGFR protein has

been well and deeply embedded in clinical literatures^{14, 15}. To attract broader clinical audiences, we used immature protein numbering (e.g. L858R *etc.*) in our manuscript. For EGFR phosphorylation sites, most biochemical studies used mature protein numbering^{16, 17}. To keep the consistency, we used mature protein numbering for all EGFR phosphorylation sites in this study (e.g. Y992, Y1068 and Y1173 for EGFR-WT, and Y1343, Y1419 and Y1524 for EGFR-KDD, respectively). To clarify it, all immature and mature protein numbers were included in Supplementary Table 1. We also added the clarification in our revised text.

Revised Supplementary Table 1:

Supplementary Table 1 | Codon numbering of human EGFR protein in this study

Protein	Immature protein numbering ^a	Mature protein numbering ^b
EGFR-WT/EGFR-KDD	I706Q	I682Q
EGFR-WT/EGFR-KDD	D837N	D813N
EGFR-WT	L858R	L834R
EGFR-WT/EGFR-KDD	V948R	V924R
EGFR-WT	Y1016	Y992
EGFR-WT	Y1092	Y1068
EGFR-WT	Y1197	Y1173
EGFR-KDD	I1057Q	I1033Q
EGFR-KDD	D1188N	D1164N
EGFR-KDD	V1299R	V1275R
EGFR-KDD	Y1367	Y1343
EGFR-KDD	Y1443	Y1419
EGFR-KDD	Y1548	Y1524

^a Immature, codon numbering of the human immature EGFR sequence that includes the 24-residue signal sequence;

^b Mature, codon numbering of the human mature EGFR sequence that does not include the 24-residue signal sequence.

- Revised text, page 4 and 12: “For EGFR mutations, we utilized codon numbering of the human immature EGFR sequence that includes the 24-residue signal sequence.”
- Revised text, page 5 and 13: “For EGFR phosphorylation sites, we utilized codon numbering of mature EGFR sequence that does not include the 24-residue signal sequence.”

2) How do the JMB residues compare (in the model) in location to those seen in the Red-Brewer crystal structure (PDB 3GOP) – where they do form “enthalpically stabilizing contacts” between kinase domains? It seems odd not to have any comparison of this. How does the linker length affect the structure here? This was not clear from the analysis. And the only comparison is with a model – when there is a crystal structure of WT including this region.

Authors’ Response: This is an excellent suggestion by the reviewer. We have now included an additional supplementary figure, Supplementary Fig. 3, which in addition to showing the MD-equilibrated structural models in Fig. 2, also includes a comparison to the X-ray structure of the EGFR-WT homodimer with JMB (PDB ID 3GOP). We reference the new supplementary figure in the Results subsection “Linker-mediated proximity of the intra-molecular dimer is sufficient, but not solely responsible, for EGFR-KDD intra-molecular dimer catalytic activity”, paragraphs 1 and 3. Note that the X-ray structure and MD-equilibrated EGFR-WT homodimer do not appreciably differ in their contacts at the JMB latch site.

Revised Supplementary Fig. 3:

Revised figure legend of Supplementary Fig 3:

Supplementary Fig. 3 | Comparison of EGFR-KDD computational models with X-ray structure of EGFR-WT juxtamembrane latch

- a, X-ray structure of the EGFR-WT homodimer with juxtamembrane latch;
- b, Rosetta model of EGFR-WT homodimer with juxtamembrane latch post-equilibration for 1.0 μ s MD simulation;
- c, Rosetta model of EGFR-KDD intra-molecular dimer post-equilibration for 1.0 μ s MD simulation;
- d, Rosetta model of EGFR-KDD intra-molecular dimer post-equilibration for 2.0 μ s MD simulation; the receiver kinase domain N-terminal JMB domain is colored green; residues within 6.0 \AA of JMB are colored blue.

With respect to the reviewer's great question regarding how linker length effects the interactions at the JMB interface, we have no specific data; however, we have added text in the Discussion to speculate (page 10 of the revised main text):

- Revised text, page 10: "We next characterized differences in the functionality of the JMB region of EGFR-KDD relative to EGFR-WT. The JMB is a conserved stretch of amino acids critical for inter-molecular dimerization in wild-type HER-family receptor kinases. In EGFR-KDD, the JMB region of

TKD2 is covalently linked to the C-terminus of TKD1. All-atom computational modeling investigations coupled with in vitro mutagenesis suggests the EGFR-KDD linker region is capable of forming specific stabilizing JMB domain contacts within the intra-molecular dimer; however, the forced proximity of the two kinase domains by the linker is sufficient for elevated EGFR-KDD activity relative to EGFR-WT. In comparison, EGFR-WT depends on stable contacts in the JMB domain for dimer activity. We focused our analysis on EGFR-KDD with duplication of exons 18-25, but other groups have recently identified EGFR-KDD with longer duplications (e.g. exons 14-26 and exon 17-25) that may reduce the likelihood of forming stabilizing contacts at the linker JMB interface of the intra-molecular dimer. We speculate that there may be selective pressure for specific linker lengths/sequences in the formation of KDDs. Recent investigations have suggested similar structural constraints in the context of EGFR exon 19 deletion mutations.”

3) It would be nice to see some experimental structural analysis of the KDD.

Authors' Response: We thank the reviewer for this comment, and completely agree with his/her assessment that it would be great to have experimental structural data on EGFR-KDD. Indeed, we pursued this strategy with a collaborator, but were unfortunately unsuccessful. We hope future studies will be able to provide an experimental structure. Instead, we have provided a combination of biochemical, biophysical, and computational studies to probe the EGFR-KDD structure-function relationship.

4) The authors argue that the cetuximab results in Fig. 3 show that EGF stimulation promotes KDD activity through formation of intermolecular dimers – but they really do not. It's likely, but the main function of cetuximab is to block EGF binding (see ref 34) – which is a different facet.

Authors' Response: We thank the reviewer for this comment. The reviewer is concerned that cetuximab does not necessarily prevent EGF-dependent dimerization of EGFR, citing a seminal work by Li et al., 2005, *Cancer Cell*, where Li and colleagues crystallized the antigen-binding region (FabC225) in complex with the EGFR ectodomain (sEGFR)¹⁸. We acknowledge and agree that there may be spontaneous EGF-independent intermolecular dimerization in the presence of cetuximab that is primarily mediated by intracellular kinase domain interactions (e.g. as has been supported elsewhere with α C-helix stabilizing mutants Ex19Del (E746_A750del) and L858R)^{12, 17} and which may be exacerbated with elevated intracellular concentrations of EGFR. Nevertheless, our focus here is on EGF-dependent dimerization, the literature is clear that the dominant equilibrium state of EGFR in complex with cetuximab is that of a monomer precisely because cetuximab prevents dimeric association mediated by EGF-ligand^{4, 18-20}.

To quote the Results & Discussion subsection “Conformation of sEGFR bound to cetuximab” from Li et al., 2005, *Cancer Cell*¹⁸, “Most importantly, the extended conformation that is required for dimerization is inaccessible to FabC225-bound sEGFR”. Moreover, multiple additional independent investigations have demonstrated that Cetuximab disrupts EGFR dimer formation, shifting the equilibrium toward monomers^{4, 19, 20}.

Our own data in the present manuscript show that YAMC cells stably expressing EGFR-KDD demonstrate an increase of phosphorylated EGFR in the presence of EGF (**Fig. 3b, lanes 5, 7**). Based on our Co-IP (**Fig. 4a**) and PIE-FCCS (**Fig. 4d**) data, EGFR-KDD stimulated by EGF causes increased EGFR-KDD homo-dimerization/multimerization). If we pre-incubate EGFR-KDD-expressing YAMC cells with cetuximab prior to EGF-ligand treatment, we do not observe an increase of phosphorylated EGFR (**Fig. 3b, lanes 5-8**). All of these data are consistent with the previously reported literatures^{4, 18-20}, and it all suggests that the most likely scenario is one in which cetuximab disrupts EGF-ligand-mediated dimer formation.

5) It is not clear that the model in Fig. 3d has any support. Is it needed? It seems clear that 3c can happen, and there is really no evidence in the literature for activation (of TKD3) in a side-by-side manner, so 3d seems very speculative (and probably should be deleted).

Authors' Response: We thank the reviewer for this comment. The reviewer is concerned that there is not sufficient evidence to warrant including the template-based model presented in Fig. 3d. We acknowledge that

this model is not as intuitive as that presented in Fig. 3c (which is a template-based model based on Huang et al. 2016 Elife²¹), but we maintain that there is strong literature evidence to support it.

Needham et al., 2016 *Nature Communications* combined fluorophore localization imaging with photobleaching (FLIMP) and over 50 μ s of MD simulations to build an EGFR-WT tetramer model in the side-by-side (or “dimer of dimers” as it is referred in Needham et al. 2016) configuration²². In that manuscript, they also demonstrated that predictions from this tetramer model are consistent with EGF-ligand dose-dependent phosphorylation effects observed with EGFR-WT. The C-terminal tails of the receiver kinase domains are accessible to the neighbor for phosphorylation, and inactivation of TKD3 is sterically hindered over the course of a single long 40 μ s MD simulation (well beyond the inactivation timescale of EGFR-WT^{23, 24}; we have our own replicates of EGFR-WT inactivation during MD simulation on the timescales of 2-8 μ s, sans DFG-flip, unpublished, available if needed) of the proposed full-length EGFR-WT tetramer despite full inactivation of the other donor kinase, TKD1. PDB coordinates for their full-length EGFR-WT model are available as supplemental material in Needham et al. 2016²².

Nevertheless, the reviewer’s point is well-received, and we have softened the language surrounding Fig. 3d. We have changed the sentence in the main text to emphasize that this event should be considered a steric effect and not a full dimerization event.

- Original text, page 7: “In the side-by-side model, active-state stabilization of TKD3 occurs through a non-canonical asymmetric dimerization event with TKD1 (inter-molecular donor, intra-molecular donor) (Fig. 3d).”
- Revised text, page 7: “In the side-by-side model, active-state stabilization of TKD3 could occur through sterically impaired inactivation by TKD1 (inter-molecular donor, intra-molecular donor) (Fig. 3d), as observed in the 40 μ s MD simulation of the EGFR-WT full-length tetramer model in Needham et al. 2016.”

6) Why are JMB residues of TKD3 not interacting with the C-lobe of TKD2 in the model in Fig. 3c. They surely will. Indeed, such a model can be generated just by looking at the (experimentally seen) crystal packing in PDB entry 3G0P.

Authors’ Response: We thank the reviewer for this astute observation. He/she is correct – the JMB domain from TKD3 would interact with the C-lobe of TKD2 in the Fig. 3c model. To address the reviewer’s concern and add clarity to our manuscript, we have amended Fig. 3c to include the JMB interaction between TKD3 and TKD2.

Revised Fig. 3c:

We also emphasize that the template-based models presented in Fig. 3 are hypothesized models meant to illustrate two possibilities supported by low-resolution biophysical data. They are fundamentally just extensions of two posited EGFR-WT tetramer models in the literature^{21,22}.

7) The authors do not discuss why EGF-activated KDD is so much more robust than constitutive. This presumably reflects the fact that the asymmetric dimer forms more ideally intermolecularly – with TKD2 as activator and TKD1' (or TKD3) as receiver – than intramolecularly (with TKD2 as receiver and TKD1 as activator). This is not reflected in Fig. 3c, but is strongly supported by looking at crystal packing as in point 6.

Authors' Response: This is an excellent point by the reviewer. There are several reasons why EGFR-KDD may have robust EGF-dependent activation than other activating EGFR mutants. We have added the following text to the Discussion to address this point:

- **Revised text, page 11:** “Interestingly, EGF-stimulated EGFR-KDD displays substantially more phosphorylation than EGF-stimulated canonical activating mutations (**Fig. 5a** and **Supplementary Fig. 4a**). We speculate that this may be because of the increased ratio of EGF-ligand to active recruited kinase domains in EGFR-KDD (i.e. EGF-mediated dimerization of two extracellular domains results in an effective tetramer of intracellular kinase domains with potentially 2 – 3 active TKDs, versus typical oncogenic activation with 1 – 2 active TKDs). Alternatively, it may be that the EGFR-KDD intermolecular dimer forms a more favorable interface than other oncogenic mutants, thus resulting in increased dimerization and activity. A combination of factors likely contributes to the overall increase in phosphorylation that we observe. Additional studies are needed to characterize the EGFR-KDD intermolecular dimer.”

8) It would be nice to see some experiments studying KDD heterodimerization with other ERBB/HERs when all expressed at cancer-relevant levels (rather than in the transient expressions used here, which rather force the system).

Authors' Response: We thank the reviewer for this insight and agree that these experiments would be ideal. We do note that while the PIE-FCCS experiments were done with transient transfection, the expression level of each cell was fully quantified. Monitoring expression level and dimerization simultaneously is a key advantage of PIE-FCCS. The average protein density for these experiments was reported in the last row of Supplementary Table 2. *These expression levels are below those seen in cancer cells²⁵*, and thus high expression levels are not forcing the system toward dimerization.

9) In Fig. 4e, did the authors try adding EGF and NRG1? They comment that heterodimerization of KDD with HER3 was seen for NRG1 but not EGF. This seems also to be true for WT EGFR here – so this sentence might be rephrased?

Authors' Response: For the PIE-FCCS experiments we did not co-stimulate with EGF and NRG1. We did add a statement that WT-EGFR also heterodimerizes with HER3 in the presence of NRG1. Revised text:

- **Revised text, page 8:** “Interestingly, our biophysical data suggest that like EGFR-WT, EGFR-KDD also heterodimerizes with HER3 in the presence of NRG1 but not EGF (Fig. 4e, Supplementary Fig. 4g, Supplementary Table 1).”

10) Did the authors undertake any studies to determine whether KDD versions of HER2,3,4 are constitutively active?

Authors' Response: This is a great question by the reviewer. Unfortunately, the time required to generate the necessary constructs for these studies precluded us from performing KDD investigations into HER2, 3, 4 proteins. This is an important area for future investigations.

11) One important point is that the individual L1038 and L1039 mutations do not cause any reduction in phosphorylation – only mutating both does this in Fig. 2g. It might be worth pointing out that the LL here is a consequence of the fusion (the real JMB sequence is ELVEPLTPS. This might warrant some discussion.

Authors' Response: We thank the reviewer for this excellent point. We have added the following sentences to paragraph 4 of the Results subsection “Linker-mediated proximity of the intra-molecular dimer is sufficient, but not solely responsible, for EGFR-KDD intra-molecular dimer catalytic activity” to acknowledge that neither single mutation substantially reduces phosphorylation and that the domain fusion uniquely results in the dual leucine residues at the interface:

- Revised text, page 8: “Individual point mutations L1038A/R and L1039A/R do not appreciably reduce phosphorylation; only the combined mutations reduce phosphorylation. Importantly, the sequential leucine residues in the linker are a unique feature of EGFR-KDD resulting from the domain fusion”

12) Why no phospho-receptor blots in Fig. 4a and c?

Authors' Response: We thank reviewer for this question. Our primary goal with Fig. 4 was to probe for hetero-interactions between EGFR-KDD and EGFR-WT/HER2/HER3, not to evaluate the effects these potential hetero-interactions may have on phosphorylation. Our constructs do not preclude the formation of homo-interactions, and thus any observed phosphorylation would not be strictly attributable to hetero-interactions. Nevertheless, in response to the reviewer's query, we did probe the phosphorylation of each receptor.

We observed phosphorylation of EGFR-WT and EGFR-KDD with co-transfection of EGFR-WT/EGFR-KDD (**Fig. 4a, lane 7, 8**). We also observed phosphorylation of HER2 with co-transfection of EGFR-KDD/HER2 (Y1222/1221) (**lane 3, Fig. 4c**), and HER3 with co-transfection of EGFR-KDD/HER3 (Y1289) (**lane 3, Supplementary Fig. 5d**).

For reviewer only modified Fig. 4a and 4c:

While our phospho-receptor blots do show phosphorylation, we cannot attribute it specifically to the formation of hetero-dimers/multimers, and therefore we do not think it adds value to the manuscript.

Reviewer #2 (Reviewer Comments to the Author)

This study of kinase domain duplicate EGFR (EGFR-KDD) is comprehensive and convincing in demonstrating the molecular mechanism underlying the mutant's aberrant activity. It showed that the activity of EGFR-KDD, like the WT, is dependent on the formation of the asymmetric kinase domain dimer and the juxtamembrane segments remain important. The simulation study of the EGFR-KDD kinase domain dimer is well executed and clearly represented. The authors' model is consistent with the model reported by Arkhipov et al., Cell, 2012, in terms of how the juxtamembrane segment of the receiver interacts with the donor in activation. (This should be most obvious from the structural coordinates of EGFR active dimer model as SI material of the paper.) The other very interesting part of this work concerns the oligomerization of the kinase domains. The authors showed that likely upon EGF binding EGFR-KDD forms dimers, within which the kinase domains form two asymmetric dimers in contact with one another. Here this reviewer would like to suggest one arrangement that the authors may have not considered but fits their experimental data well. In this EGFR-KDD dimer model, the KD1 of one EGFR-KDD forms an asymmetric dimer with the KD2 of the other EGFR-KDD as receiver. This would explain why KD2-dead EGFR KDD can be activated by EGF-induced dimerization.

Authors' Response: We thank the reviewer for his/her favorable impression of our manuscript. We will provide PDB coordinates for our EGFR-KDD structural models in the Source Data upon resubmission. Specifically, PDB structures for the MD-equilibrated EGFR-WT dimer at 1.0 μ s, EGFR-KDD at 1.0 μ s, and EGFR-KDD at 2.0 μ s have been included in "Source Data". We have added the additional suggested inter-molecular dimer arrangement as a thoughtful and important speculation for readers in the Results subsection "EGF ligand stimulation induces inter-molecular multimer activity":

- Revised text, page 7: "Other models are possible (e.g. the receiver kinase of one intra-molecular dimer could act as the donor to the receiver kinase of a second intra-molecular dimer). The models presented here are simply based on two supported models of EGFR-WT tetramer in the literature."

To address the relevance of EGFR-KDD as a research tool, we have added the following paragraph to the Discussion:

- Revised text, page 7: "Finally, our computational and biochemical insights raise important considerations for the use of EGFR-KDD as a research tool. Whereas the inactive form of EGFR can be readily studied by the introduction of inter-molecular dimer-disrupting interface mutations¹⁶, controlling the active fraction of EGFR in vitro has typically required introduction of known oncogenic point mutations or stimulation with EGF-ligand. The former causes well-documented perturbations to enzyme kinetics^{9, 11, 26, 27}, while recent literature has demonstrated that the latter can influence EGFR multimerization and phosphorylation in a concentration-dependent manner²². Moreover, dimerization and activation of EGFR oncogenic missense mutants is dependent on protein concentration²⁸ and/or EGF- ligand stimulation¹⁷. EGFR-KDD provides a model of a fully active EGFR dimer in an EGF-independent setting, and may provide a more native-like control than kinase domain missense mutants without the complexity of concentration-dependent signaling effects."

Reviewer #3 (Reviewer Comments to the Author)

Regarding the FCCS studies, the quantitative interpretation of the data should be improved.

1) Homo-interactions are quantified using FCCS and labeling the same protein with either mCherry or GFP. This can be deduced but it is not clearly explained. I had also trouble deducing (for hetero-interactions) which construct was labeled with GFP and which with mCherry.

Authors' Response: We thank the reviewer for this question. These details were originally omitted from the main text due to space concerns, but we agree with the reviewer suggestion that the labeling details should be clarified. Therefore, we have added a new sentence to the manuscript explaining the labels for homodimerization studies:

- Revised text, page 7 “For these experiments, the protein of interest was expressed as a mixture of eGFP and mCherry fusions, and single, live-cell measurements were recorded and analyzed as described in the Methods.”

In addition, we have also revised Fig. 4 (panels b, d, and e) and Supplementary Fig. 5 (panels b, e, f, and g) to explicitly indicate the fusion tags applied to each construct for the heteromeric interaction studies:

- Revised Fig. 4b, 4d, and 4e:

Revised figure legend of Fig. 4:

“b, Cross correlation values of co-transfected EGFR-WT (mCherry-fused) and EGFR-KDD mutant (eGFP-fused) with (+) or without (-) ligand (EGF) stimulation is shown. The light orange box indicates the f_c value region for dimers.

d, Cross correlation values of co-transfected HER2 (mCherry-fused) and EGFR-KDD mutant (eGFP-fused) with (+) or without (-) ligand (EGF) stimulation is shown.

e, Cross correlation values of co-transfected HER3 (mCherry-fused) and EGFR-KDD mutant (eGFP-fused) with (+) or without (-) ligand (EGF) stimulation is shown. For Fig. b, c and d, the median values are reported next to the boxplot. Each grey dot represents the averaged acquisition (10s, 6 acquisitions) per area per cell. Numbers in parenthesis above the boxplot are the total number of cells where data were taken on. (+: outliers)”

- Revised Supplementary Fig. 5b, 5e, 5f, and 5g:

Revised figure legend of Supplementary Fig. 5:

“b, Diffusion coefficient of EGFR WT homodimers with (+) or without (-) ligand (EGF) stimulation is shown (upper panel).

e, Diffusion coefficient of EGFR WT and EGFR KDD mutant with (+) or without (-) ligand (EGF) stimulation is shown (bottom panel).

f, Diffusion coefficient of HER2 and EGFR-KDD mutant with (+) or without (-) ligand (EGF) stimulation is shown.

g, Diffusion coefficient of HER3 and EGFR-KDD mutant with (+) or without (-) ligand (EGF or NRG1) stimulation is shown.”

2) The authors use the words dimerization and multimerization. That implies a quite precise quantification of the multimeric state of the receptor (and stoichiometries) which, in the current state of the manuscript, appears doubtful. I understand that similar measurements were performed already few years ago and the authors refer to them when, for example, mentioning controls (Ref. 48). Nevertheless, I believe that a positive control (definitely) and a negative control (this is maybe less needed) should be included in these experiments. In my experience, the maximal detectable CC varies during time (due to e.g. misalignment, glass slides with slightly different thickness, dirt on the objective, wearing of optics) and that is why we perform positive controls every day. Finally, the use of a positive control helps surely in the case of a 1-to-1 interaction, but this case the analysis is also more complicated because other stoichiometries cannot be excluded (see below).

Authors' Response: We thank the reviewer for his/her expertise in pointing out these potential complications, and we agree that we should clarify our interpretation of the data. For the homodimerization experiments, we have developed a quantitative model that enables us to distinguish monomers, dimers, and small multimers from the fc distributions²⁹. We have tested it against two control systems, and the data have been published previously³⁰.

That said, we agree that the control data need to be continually collected (for the reasons pointed out by the reviewer), and we did in fact perform those control experiments alongside those presented in the paper. We have added a new supplementary figure, Supplementary Fig. 4b, showing the control experiments that were performed concurrently with the main experiments.

- Revised Supplementary Fig. 4b:

Revised figure legend of Supplementary Fig. 4b:

b, Cross correlation values of PIE-FCCS control constructs. The monomer control (Myr-FP: myristoylated fluorescent protein [mCh or eGFP; coexpressed together]) had an f_c value of 0.01 indicating no interaction. Upon cross-linking by a synthetic dimerizer (AP: AP20187) the dimer control (1xFKBP-FP) had an average f_c value of 0.11, consistent with dimerization. The multimer control (3xFKBP-FP) had an f_c value of 0.26 consistent with the formation of a mixture trimer and tetramer species.

In addition, we also added a few sentences to the Methods section to explain the quality control measures taken during these experiments.

Revised text, page 13: "A short fluorescently tagged DNA fragment was used to verify the alignment of the system, including the confocal volume overlap. Negative and/or positive controls (**Supplementary Fig. 4b**) were tested regularly prior to the experimental samples for comparisons of the fit parameters."

3) The lack of positive controls for hetero-dimers makes quantitative interpretation difficult. For example (lines 307-317), WT X HER2 and KDD x HER2. EGF increases f_c to 0.1 or 0.16. What does that imply? That the amount of hetero-dimers increases? Does 0.16 indicate more hetero-dimers than 0.1? Or maybe the presence of more abundant high-order heteromultimers? The authors say that these values of f_c indicate that the proteins "heterodimerize", but it might be more complex than that. Once more, in my opinion, this conclusion would need positive controls for GFP-mCherry and the absence of complexes with different stoichiometries. Also, f_c of 0.06 appears to mean absence of heterodimerization. But how quantitative is that? If 0.16 is e.g. 100% heterodimers, is 0.06 33% heterodimers and the rest are monomers?

Another example: KDD heterodimerizes with HER3 in the presence of NRG1 but not EGF. A quite strong assessment. But f_c is 0.14 in the presence of NRG1 and more than half of that (0.08) in the presence of EGF. Does that mean half so many heterodimers as with NRG1? If so, can we say that EGF does not cause heterodimerization? Once more, as a reminder, the possible presence larger (hetero-) multimers with more complex stoichiometry would make the calculations much more complex, but this is not discussed so far.

Authors' Response: We thank the reviewer for his/her insightful questions here. Quantifying heteromeric complexes in live cells is indeed very challenging. This work is still at the cutting edge of bioanalytical methods development, and only a handful of FCCS studies have presented such data - none of which developed the necessary controls to precisely determine the dimerization affinity or oligomer size distribution. Developing

heterodimer/multimer controls is especially challenging because of potential competition with homodimerization. These competing factors are clearly at play for EGFR and the HER family, so even with ideal controls it would be difficult to establish precise interaction affinities without extensive knock out and mutagenesis studies. Since this level of characterization is not necessary for the key conclusions of our current study, we instead amended the text to make it clear we are not claiming to have quantified the heterodimerization affinity or stoichiometry. Assessment of the heterodimer percentages as brought up by the reviewer is not possible without substantial further work. *Importantly, positive f_c values are rigorous evidence for at least the formation of heteromeric species, the minimal size of which is a heterodimer.* For this reason, we will use that word for simplicity, but clearly indicate the caveats in the text.

In discussing our PIE-FCCS data in the Results subsection “EGFR-KDD directly interacts with EGFR-WT and other ERBB family members”, we have added the following sentence for clarification:

- Revised text, page 7: “With f_c value, we can distinguish homo- and heterodimerization, which cannot be assessed with diffusion coefficient alone.”

We further revised the subsequent text to clearly express the limitations of our approach:

- Revised text, page 8: “Upon addition of EGF-ligand, there was a significant increase in cross-correlation ($f_c = 0.22$) indicating the formation of heteromeric complexes (**Fig. 4b, Supplementary Fig. 4b - c, Supplementary Table 2**). The positive cross-correlation is rigorous evidence for heteromeric complex formation, but alone is not sufficient to define the interaction strength or stoichiometry of the complexes. For simplicity we will refer to these complexes as heterodimers as this is the minimal size consistent with positive cross-correlation. In agreement with changes to the f_c values, the diffusion coefficients of both EGFR-WT and EGFR-KDD decreased after ligand addition, indicating slower diffusion due to homo- and hetero-dimerization/multimerization (**Supplementary Fig. 4b - c, Supplementary Table 2**).”

We have also clarified text related to EGFR-KDD/HER3 interactions:

- Revised text, page 8: “Interestingly, our biophysical data suggest that like EGFR-WT, EGFR-KDD also heterodimerizes with HER3 in the presence of NRG1 but not EGF (**Fig. 4e, Supplementary Fig. 5g, Supplementary Table 2**).”

4) The authors measure and report brightness values. This is actually valuable because more information regarding the average stoichiometry could be extracted (if needed! If this is not needed, that would be also fine, but then precise expressions such as “dimerization” should be avoided). Brightness values are reported in supplementary table 1. I would recommend clarifying which values refer to GFP or mCherry. Of course, in this case, monomer and homo-dimer (!) controls are additionally needed to normalize the brightness values. Both for GFP and mCherry. Especially in the case of mCherry, it is known that the protein has a low fluorescence / maturation probability, and this affects the CC values that can be measured. That is why homo-dimer controls would also be required. We agree that controls should be included for brightness. A new figure was added to the supporting information file with brightness data for the controls discussed in the previous comment.

Authors’ Response: We agree that controls should be included for brightness. To address this concern, we have reported our brightness data for the controls for both GFP and mCherry fusion tags as Supplementary Fig. 4c (please see page 16).

Revised Supplementary Fig. 4c:

Revised figure legend of Supplementary Fig. 4c:

c, Molecular brightness of PIE-FCCS negative and positive controls in Supp Fig 3b is shown (Left: constructs with eGFP tag; right: constructs with mCh tag). The oligomer control (3xFKBP+AP) has much higher molecular brightness as expected due to clustering. mCh-tagged constructs show subtle changes in the molecular brightness due to the photophysical properties of mCherry. However, the molecular brightness changes are still statistically significant between all constructs.

Furthermore, we also modified Supplementary Table 2 to indicate the fluorescent protein labels.

Revised Supplementary Table 2:

Construct	EGFR-WT		EGFR-KDD		EGFR-WT × EGFR-KDD			
	EGF		EGF		EGFR-WT mCh	EGFR-KDD eGFP	EGFR-WT mCh	EGFR-KDD eGFP
Number of cells	-	109	+	121	-	76	+	62
f_e^a		0.027 ± 0.004		0.191 ± 0.008		0.045 ± 0.007		0.232 ± 0.013
f_e^b		0.004		0.185		0.005		0.220
Brightness ^a (cpsm)		275 ± 9		335 ± 9		395 ± 10		333 ± 16
Diffusion Coefficient ^a (μm ² /s)		0.50 ± 0.02		0.29 ± 0.02		0.44 ± 0.02		0.24 ± 0.02
Density ^a (mol/μm ²)		733 ± 55		868 ± 56		605 ± 41		574 ± 38
								894 ± 50

^amean value; ^bmedian value (reported on the figures)

Construct	EGFR-WT × HER2				EGFR-KDD × HER2			
	EGFR-WT eGFP	HER2 mCh	EGFR-WT eGFP	HER2 mCh	EGFR-KDD mCh	HER2 eGFP	EGFR-KDD mCh	HER2 eGFP
Number of cells	-	58	+	55	-	56	+	44
f_e^a		0.048 ± 0.008		0.113 ± 0.010		0.102 ± 0.016		0.168 ± 0.016
f_e^b		0.007		0.105		0.058		0.159
Brightness ^a (cpsm)		269 ± 7		312 ± 13		338 ± 8		233 ± 6
Diffusion Coefficient ^a (μm ² /s)		0.53 ± 0.03		0.30 ± 0.02		0.32 ± 0.02		0.29 ± 0.03
Density ^a (mol/μm ²)		749 ± 77		489 ± 48		252 ± 24		900 ± 107
				907 ± 94				359 ± 42
				838 ± 63				891 ± 92

Construct	EGFR-WT × HER3				EGFR-KDD × HER3					
	EGFR-WT eGFP	HER3 mCh	EGFR-WT eGFP	HER3 mCh	EGFR-WT eGFP	HER3 mCh	EGFR-KDD mCh	HER3 eGFP	EGFR-KDD mCh	HER3 eGFP
Number of cells	-	84	+	58	-	65	+	94	-	50
f_e^a		0.047 ± 0.007		0.051 ± 0.009		0.129 ± 0.012		0.084 ± 0.006		0.089 ± 0.013
f_e^b		0.014		0.017		0.130		0.057		0.078
Brightness ^a (cpsm)		260 ± 7		383 ± 10		377 ± 13		330 ± 8		271 ± 6
Diffusion Coefficient ^a (μm ² /s)		0.51 ± 0.02		0.33 ± 0.01		0.28 ± 0.02		0.28 ± 0.01		0.36 ± 0.03
Density ^a (mol/μm ²)		433 ± 27		258 ± 19		438 ± 42		379 ± 36		430 ± 39
				393 ± 46				169 ± 18		232 ± 19
				302 ± 7				186 ± 17		313 ± 30
				297 ± 10				212 ± 25		257 ± 28

4-1) Looking at the brightness values, some interesting questions arise. For example, the brightness of WT (alone) increases with EGF (in accordance to the increase in CC) due probably to the presence of a small amount of dimers with the same fluorescent proteins, in addition to dimers with different fluorescent proteins. But this does not happen with KDD. Does this indicate different complex stoichiometries between the two proteins?

Authors' Response: In principle, the brightness values could be used to assess the stoichiometry of the complexes. However, in two color experiments, the changes to the molecular brightness are reduced because of competition with the opposite color. Complications also arise because brightness is much more susceptible to the minor changes in the instrument alignment from day to day. Therefore, we have chosen not to try to use molecular brightness for interpretation of the complex sizes for this manuscript. The values in Supplementary Table 1 are provided mainly to show that the alignment across all the experiments was consistent.

4-2) In the case of WT X HER2, the brightness of the WT increases in the presence of EGF (while that of HER2 does not). Might this indicate the presence of WT multimers bound to HER2 monomers? Also, the brightness of KDD increases (w/o EGF), compared to WT. And it increases even more in the presence of EGF (while HER2 brightness remains again constant). Is this data worth discussing maybe? Finally, as mentioned above, the authors mention the presence of "heterodimers" between KDD and HER3 in the presence of NRG1 but not EGF. This is motivated by the strong increase in *fc*. Nevertheless, the brightness of WT increases significantly in the presence of EGF, but not NRG1. The same happens with WT and HER3. Might this be relevant and indicate, e.g., the presence of hetero-multimers with different stoichiometries, rather than simply dimers?

Authors' Response: These are very perceptive comments, and we generally agree with the reviewer's interpretations. There is certainly going to be competition between homodimer and heterodimer formation upon stimulation and the brightness changes could in principle be used to tease out the details of these changes. However, because brightness comparisons between must be done in single color experiments to calculate the precise FRET efficiency (because of the difference in molecular brightness as shown in Supplementary Fig. 4c), and because the lack of hetero-interaction controls makes it challenging to identify the interaction stoichiometry, we chose not to include discussion of these ideas in the manuscript. In future work we plan to knock down the receptors and perform *in situ* competition assays to quantify the affinity and stoichiometry of homomeric and heteromeric interactions between HER family members.

5) "Part numbers" should be corrected

Authors' Response: This has been corrected in the revised manuscript.

- Revised text, page 13: "The fluorescence signal was collected through a home-built confocal detection unit with a 50 μm confocal pinhole and dichroic beam splitter (LM01-503-25, Semrock, Rochester, NY). The two signals were filtered (91032, Chroma Technology Corp., Bellows Falls, VT; zt488/561rpc and zet488/561m, Chroma Technology Corp., Bellows Falls, VT) and then focused independently on to single-photon avalanche diodes (Micro Photon Devices, Bolzano, Italy)."

References

1. Ramalingam, S. et al. Dual inhibition of the epidermal growth factor receptor with cetuximab, an IgG1 monoclonal antibody, and gefitinib, a tyrosine kinase inhibitor, in patients with refractory non-small cell lung cancer (NSCLC): a phase I study. *J Thorac Oncol* **3**, 258-264 (2008).
2. Janjigian, Y.Y. et al. Dual inhibition of EGFR with afatinib and cetuximab in kinase inhibitor-resistant EGFR-mutant lung cancer with and without T790M mutations. *Cancer discovery* **4**, 1036-1045 (2014).
3. Horn, L. et al. Continued use of afatinib with the addition of cetuximab after progression on afatinib in patients with EGFR mutation-positive non-small-cell lung cancer and acquired resistance to gefitinib or erlotinib. *Lung Cancer* **113**, 51-58 (2017).
4. Cho, J. et al. Cetuximab response of lung cancer-derived EGF receptor mutants is associated with asymmetric dimerization. *Cancer Res* **73**, 6770-6779 (2013).
5. Neal, J.W. et al. Cetuximab monotherapy in patients with advanced non-small cell lung cancer after prior epidermal growth factor receptor tyrosine kinase inhibitor therapy. *J Thorac Oncol* **5**, 1855-1858 (2010).
6. Soria, J.C. et al. Osimertinib in Untreated EGFR-Mutated Advanced Non-Small-Cell Lung Cancer. *N Engl J Med* **378**, 113-125 (2018).
7. Ramalingam, S.S. et al. Overall Survival with Osimertinib in Untreated, EGFR-Mutated Advanced NSCLC. *N Engl J Med* **382**, 41-50 (2020).
8. Yoshikawa, S. et al. Structural basis for the altered drug sensitivities of non-small cell lung cancer-associated mutants of human epidermal growth factor receptor. *Oncogene* **32**, 27-38 (2013).
9. Yun, C.H. et al. The T790M mutation in EGFR kinase causes drug resistance by increasing the affinity for ATP. *Proceedings of the National Academy of Sciences of the United States of America* **105**, 2070-2075 (2008).
10. Yasuda, H. et al. Structural, biochemical, and clinical characterization of epidermal growth factor receptor (EGFR) exon 20 insertion mutations in lung cancer. *Science translational medicine* **5**, 216ra177 (2013).
11. Carey, K.D. et al. Kinetic analysis of epidermal growth factor receptor somatic mutant proteins shows increased sensitivity to the epidermal growth factor receptor tyrosine kinase inhibitor, erlotinib. *Cancer Res* **66**, 8163-8171 (2006).
12. Wang, Z. et al. Mechanistic insights into the activation of oncogenic forms of EGF receptor. *Nat Struct Mol Biol* **18**, 1388-1393 (2011).
13. Doody, J.F. et al. Inhibitory activity of cetuximab on epidermal growth factor receptor mutations in non small cell lung cancers. *Mol Cancer Ther* **6**, 2642-2651 (2007).
14. Sharma, S.V., Bell, D.W., Settleman, J. & Haber, D.A. Epidermal growth factor receptor mutations in lung cancer. *Nature reviews. Cancer* **7**, 169-181 (2007).
15. Yasuda, H., Kobayashi, S. & Costa, D.B. EGFR exon 20 insertion mutations in non-small-cell lung cancer: preclinical data and clinical implications. *The Lancet. Oncology* **13**, e23-31 (2012).
16. Zhang, X., Gureasko, J., Shen, K., Cole, P.A. & Kuriyan, J. An allosteric mechanism for activation of the kinase domain of epidermal growth factor receptor. *Cell* **125**, 1137-1149 (2006).
17. Red Brewer, M. et al. Mechanism for activation of mutated epidermal growth factor receptors in lung cancer. *Proceedings of the National Academy of Sciences of the United States of America* **110**, E3595-3604 (2013).
18. Li, S. et al. Structural basis for inhibition of the epidermal growth factor receptor by cetuximab. *Cancer cell* **7**, 301-311 (2005).
19. Oashi, A. et al. Monomer Preference of EGFR Tyrosine Kinase Inhibitors Influences the Synergistic Efficacy of Combination Therapy with Cetuximab. *Mol Cancer Ther* **18**, 1593-1601 (2019).
20. Jia, Y. et al. Overcoming EGFR(T790M) and EGFR(C797S) resistance with mutant-selective allosteric inhibitors. *Nature* **534**, 129-132 (2016).
21. Huang, Y. et al. Molecular basis for multimerization in the activation of the epidermal growth factor receptor. *Elife* **5** (2016).
22. Needham, S.R. et al. EGFR oligomerization organizes kinase-active dimers into competent signalling platforms. *Nature communications* **7**, 13307 (2016).
23. Shan, Y., Arkhipov, A., Kim, E.T., Pan, A.C. & Shaw, D.E. Transitions to catalytically inactive conformations in EGFR kinase. *Proceedings of the National Academy of Sciences of the United States of America* **110**, 7270-7275 (2013).

24. Shan, Y. et al. Oncogenic mutations counteract intrinsic disorder in the EGFR kinase and promote receptor dimerization. *Cell* **149**, 860-870 (2012).
25. Endres, N.F. et al. Conformational coupling across the plasma membrane in activation of the EGF receptor. *Cell* **152**, 543-556 (2013).
26. Yun, C.H. et al. Structures of lung cancer-derived EGFR mutants and inhibitor complexes: mechanism of activation and insights into differential inhibitor sensitivity. *Cancer cell* **11**, 217-227 (2007).
27. Gilmer, T.M. et al. Impact of common epidermal growth factor receptor and HER2 variants on receptor activity and inhibition by lapatinib. *Cancer Res* **68**, 571-579 (2008).
28. Sholl, L.M. et al. Lung adenocarcinoma with EGFR amplification has distinct clinicopathologic and molecular features in never-smokers. *Cancer Res* **69**, 8341-8348 (2009).
29. Kaliszewski, M.J. et al. Quantifying membrane protein oligomerization with fluorescence cross-correlation spectroscopy. *Methods* **140-141**, 40-51 (2018).
30. Christie, S., Shi, X. & Smith, A.W. Resolving Membrane Protein-Protein Interactions in Live Cells with Pulsed Interleaved Excitation Fluorescence Cross-Correlation Spectroscopy. *Acc Chem Res* **53**, 792-799 (2020).

Reviewers' comments:

Reviewer #1 (Remarks to the Author):

Du et al. have changed relatively little in their revised manuscript, primarily editing only the FCC section and adding to the Discussion. In their rebuttal, they mostly try to dismiss the criticisms concerning interpretations that were raised in the previous round of review. Overall, although the work is reasonable for the most part (except for issues of reproducibility mentioned below and a few specific questions listed below), it is not clear that the revised paper extends the work in the same lab's 2015 Cancer Discovery paper sufficiently to warrant publication in Nature Communications.

When boiled down, Figure 1 presents data that just confirms the model proposed in the lab's earlier paper for constitutive activation of KDD. Their new modeling seems to reaffirm JMB interactions that were described before in structural and computational studies of asymmetric dimers, essentially duplicating interactions seen experimentally in PDB ID 3GOP according to Figure S3. Like other EGF mutants, KDD is activated by EGF, and studies of mutants suggest that TKD2 of one monomer can activate TKD1 of the other and vice versa (which could resemble the wild-type receptor). FCC studies further suggest that KDD can interact with wild-type EGFR when EGF is added – as expected – and with other HER family members. Finally, the authors argue for using afatinib plus cetuximab for patients with KDD, but the effects they describe seem only to be relevant for EGF-activated mutants. They do not compare quantitatively in the absence of EGF (where KDD or other mutants may drive oncogenesis), and the case for the antibody/TKI combination being different for KDD compared with other mutations is just not convincing.

Comment on Reproducibility

As a general comment, were the blots in Figure 1c, 2e and g, 3a and b, 4a and c, and 5a all repeated? If so, how many times? It is typical in such studies to present quantitation and report standard deviations. Repeats and reproducibility are not currently mentioned anywhere, which is an important concern. Were biological repeats performed.

Other comments

1. The authors might want to reword the sentence “These data suggest that the N-terminal lobe TKD1 activates the C-terminal lobe TKD2, but not the reverse.” It is not clear what it means. It is the C-lobe that ‘does the activation’. Perhaps they mean: ‘These data suggest that the N-lobe-mutated TKD1 in Fig. 1a can activate the C-lobe-mutated TKD2, but not the reverse.’?

2. It is misleading, and perhaps slightly dishonest, to say (as the authors do on page 5) that “Addition of cetuximab, an anti-EGFR extracellular domain antibody that blocks EGFR dimerization, effectively mitigates EGF-induced phosphorylation...”. This antibody binds squarely to the EGF binding site, and blocks EGF binding (that’s how it was first selected), so of course it will – regardless of its influence on dimerization. If EGF cannot bind, EGF cannot activate the receptor. If cetuximab is bound, EGF cannot bind. Yes, cetuximab will inhibit dimerization, but there can be no doubt that cetuximab blocks EGF-induced activation of EGFR by blocking EGF binding.

Given all of this, the ability of cetuximab to inhibit EGF-induced activation of KDD just means that EGF binding is required for EGF to activate KDD. It says nothing about dimerization.

The authors need to comment that cetuximab blocks EGF binding, as pointed out in the last round of review. In their rebuttal, the authors puzzlingly justify their selective comment about cetuximab’s mechanism by stating that “...our focus is on EGF-dependent dimerization...”. This simply does not make sense as a justification, because there will be none if cetuximab is bound, because it binds and blocks the EGF-binding site (and with much higher affinity than EGF does).

3. Related to point 2 above, the data in Figure 3b do not demonstrate that EGF stimulation promotes EGFR-KDD activity through the formation of intermolecular dimers. This is almost certainly what happens, although it could be tetramers or other oligomers. But, the experiment shown here simply does not show that, and cannot be interpreted in this way for the reasons mentioned in 2 above.

4. Do the authors have an explanation for the finding in Figure 3b as to why (if the result is reproducible) cetuximab appears to activate KDD, apparently quite substantially?

5. Despite the authors’ comments, this reviewer does not consider there to be sufficient evidence for the ‘side-by-side; model in Figure 3d. Needham et al. presented it as a model. If the authors are to re-present this model, they should include some data to support it over the more reasonable TKD1’/TKD2 and TKD2’/TKD1 asymmetric dimers suggested by all the experimental evidence.

6. The authors state that ‘Taken together, these data suggest that TKD3 is catalytically active in the inter-molecular dimer’. It seems that TKD3 can be active in activity of Dead2 and C1 (with EGF), but surely not in Dead1 or C2 (with EGF). The results surely argue that either kinase can be active, depending on which mutant is studied.

7. The authors need to state clearly in the text that Figure 5b was performed in the presence of a high EGF concentration, and the inhibitory effect is almost entirely from cetuximab (which blocks EGF binding). Since 50 ng/ml EGF was added, how is this relevant for KDD in cancer? The authors should also note the long-known effect of cetuximab on EGFR downregulation. Crucially, for the (no-EGF) data in Figure S6, the absence of experiments with L858R or Ex19 leave the reader unconvinced that they would not look just the same. The absence of these additional/comparative experiments (in S6b or 5b), and the use of EGF in Fig 5b, result in there really being no data here to suggest that cetuximab plus TKI will be any more useful for KDD than any other mutants. Moreover, in Fig. 5b there is still clearly detectable KDD phosphorylation (with or without EGF) when cetuximab plus afatinib are added. In fact, in both cases, one could argue (based on the gel shown – was this repeated or quantitated?) that cetuximab actually elevates phosphorylation of KDD in the presence of afatinib (compare lanes 18 and 20 or 18 and 24), which seems to contradict the authors' assertions and would not suggest combination.

8. References to Figures S5 and S6 are confused by the addition of the new S5 without corresponding corrections in the text.

Reviewer #3 (Remarks to the Author):

<We have also clarified text related to EGFR-KDD/HER3 interactions:

<● Revised text, page 8: “Interestingly, our biophysical data suggest that like EGFR-WT, EGFR-KDD
<also

<heterodimerizes with HER3 in the presence of NRG1 but not EGF (Fig. 4e, Supplementary Fig. 5g,

<Supplementary Table 2).”

I am thankful to the authors for their clear explanations. It is clear (also from the previous work) that they have a clear grasp of the methodologies used here. I also understand that we are working here at the limit of these techniques. I hope they will also be patient regarding any doubt I might have left (on the wording, not on the experiment themselves). In fact, I still am not sure that I fully agree with some dichotomic views expressed in the manuscript (I am referring to the above KDD+HER3 w EGF for example). Looking at figure 4e, it is clear that f_c for KDD-HER with NRG1 is higher than with EGF (although many measurements for this sample have a f_c way above 0.1, also quite interesting). But is this enough to say that KDD does not heterodimerize (I am assuming "at all")? Is it maybe somewhere assumed that the K_d for this complexes is either infinitely large or infinitely small? I do

not know if this might be relevant in the biological context, but what about the possibility of a moderate dimerization for these samples (giving rise to a mixture of monomers and heterodimers, with intermediate f_c values?). The same could be said of course for other samples, e.g. KDDxHER2 w/o EGF. Might e.g. "heterodimerize to a smaller/larger extent" be a preferable expression?

Regarding the considerations on brightness analysis, I fully agree with the authors. Apart from those suggestions regarding the modification in the wording of the discussion, I believe that the manuscript should be accepted for publication.

Response to Reviewer #1

Reviewer #1 (Remarks to the Author)

- I. “Du et al. have changed relatively little in their revised manuscript, primarily editing only the FCC section and adding to the Discussion.

We thank the reviewer for the opportunity to outline changes made in the manuscript. In total since the initial submission, we have now added 8 new / revised data panels in the main figures and 17 new / revised data panels in the supplementary figures, 3 new supplementary tables, and one revised supplementary table. In addition, we generated 3 revised data panels for reviewer evaluation. We have also added / revised substantial portions of the text (see the tracked changes version of the revised document where cumulative text changes are indicated by red font).

In their rebuttal, they mostly try to dismiss the criticisms concerning interpretations that were raised in the previous round of review. Overall, although the work is reasonable for the most part (except for issues of reproducibility mentioned below and a few specific questions listed below), it is not clear that the revised paper extends the work in the same lab’s 2015 *Cancer Discovery* paper sufficiently to warrant publication in *Nature Communications*.”

▪ **Regarding the final statement in this paragraph, we note substantial extensions of the present work from Gallant et al. in *Cancer Discovery*:**

- i. Genomics analysis in Gallant et al. was limited to samples from ~38,000 clinical cases. Only *EGFR-KDD* was evaluated. Here, we have analyzed samples from >277,000 cases from two of the world’s most comprehensive genomic databases (Foundation Medicine dataset and MSK-IMPACT dataset), and we have extended our analysis to include *EGFR-KDD*, *ERBB2-KDD*, *ERBB3-KDD*, and *ERBB4-KDD*. For the first time, we can now say that KDD is a recurrent oncogenic event in HER-family receptors across multiple different types of human malignancies. We respectfully submit that these genomic data alone documenting the frequency of *EGFR-KDD*, *ERBB2-KDD*, *ERBB3-KDD*, and *ERBB4-KDD* alterations across human tumors is a significant addition, and coupled with the new computational, biophysical, and biochemical experiments (discussed below) represent a significant and substantial addition to the literature over our previous work, which was just a Brief Report format in *Cancer Discovery*.
- ii. The computational model of EGFR-KDD tandem kinase domains produced for Gallant et al. in *Cancer Discovery* was a preliminary model generated purely to see if the asymmetric dimer was geometrically plausible with the given linker length. The Rosetta loop modeling protocol employed in that paper failed to converge. The new modeling strategy combined extensive Rosetta comparative modeling (which combines homology modeling with *de novo* structure prediction of undefined sequences using fragment insertion) to converge on several low energy states. These states were then systematically evaluated with molecular dynamics (MD) simulations, and the most stable structure with the best interface energy was taken as the most likely model and carried forward for additional MD simulations (see **Supplementary Fig. 2**).
- iii. Subsequent MD simulations coupled with mutagenesis completed for **Fig. 2** extend well beyond what was accomplished in Gallant et al. 2015, which did not probe the functional biochemistry of the linker region at all.
- iv. In addition to the EGF-independent intra-molecular dimer studies presented in **Fig. 1** that supported a **hypothesis** from Gallant et al. (i.e. was not determined previously), we also characterized the differences in EGFR-KDD in the presence and absence of EGF ligand. These new data demonstrate that EGFR-KDD forms a constitutive dimer in the absence of any EGF-ligand

stimulation and gets a substantial boost in activity in the presence of EGF-ligand. We further demonstrated biochemically and biophysically that this is due to the formation of at least inter-molecular dimers, and likely also EGFR-KDD homo-multimers. This is a substantially more detailed understanding of EGFR-KDD than was achieved in Gallant et al.

- v. On the basis of the above findings, we demonstrate that EGFR-KDD is inhibited more effectively by Cetuximab + Afatinib than it is by either alone (discussed in more detail below in response to the Reviewer's other queries). Importantly, the index patient for EGFR-KDD in Gallant et al. 2015 only had a partial response to afatinib, consistent with our new pre-clinical studies showing residual EGFR-KDD phosphorylation in YAMC and NR6 cells when treated with afatinib only. In addition, the anti-tumor response was short lived (7 cycles of afatinib, which is approximately 7 months) before the patient developed acquired resistance to the drug. We show in the Gallant et al. paper that acquired resistance was driven by amplification of the EGFR-KDD allele. Collectively, these observations suggested that more potent EGFR blockade was necessary to overcome the oncogenic activity of EGFR-KDD.
- vi. We demonstrate biochemically and biophysically that EGFR-KDD directly interacts with other HER-family receptors. None of these data were presented in the Gallant et al. paper. Yet, these data are important because of known interactions between HER receptors.

▪ **Regarding the statement “Overall, although the work is reasonable for the most part (except for issues of reproducibility mentioned below and a few specific questions listed below),”**

- i. We thank the Reviewer for acknowledging the merit of our work.
- ii. Despite concerns regarding reproducibility expressed in this second set of Reviewer comments, we note that we had provided all of the source data underlying our previous Figs. 1c, 1d, 2c - g, 3a, 3b, 3e, 4a - e, 5a and 5b and Supplementary Figs. 1a, 1b, 3a - d, 4a-c, 5a-g, 6a and 6b as a Source Data file in the previous version of the manuscript. We discuss the issue of reproducibility in additional detail on page 7 below.

II. **“Finally, the authors argue for using afatinib plus cetuximab for patients with KDD, but the effects they describe seem only to be relevant for EGF-activated mutants. They do not compare quantitatively in the absence of EGF (where KDD or other mutants may drive oncogenesis)...”**

- We appreciate the Reviewer's suggestion. To address this comment in the most rigorous way, we attempted to grow the cells in the absence of fetal bovine serum (FBS), which contains a manufacturer-unspecified concentration of EGF ligand. We observed that cells were not viable in these conditions. Therefore, we performed serial FBS titration experiments (in triplicate) in which we measured the growth of cells in various amounts of FBS. Of the concentrations test, consistent with EGFR studies on other mutants^{1,2}, we found that 0.5% FBS sufficiently supports cell growth for further experimentation in each of our tested mutants (**New Supplementary Fig. 7b**).
- Each of the subsequent experiments quantitatively evaluating cetuximab/afatinib inhibitory efficacy were performed in three independent replicates each of which contained three technical replicates (see **Summary of Repeat Experiments** table at the end of this document).

- We have added the following text to our manuscript to report drug efficacy results in serum starved conditions:

New Supplementary Fig. 7b

Fig. 5b

New text, page 9: At 0.5% FBS, cetuximab maximally exhibited ~40% inhibition of EGFR-KDD, ~80% inhibition of Ex19Del, and almost 100% inhibition of L858R cell viability (Fig. 5b and Supplementary Table 3). These data are consistent with a model in which EGFR-KDD retains an active intra-molecular dimer in the absence of EGF stimulation (Fig. 1) and previously published models of Ex19Del and L858R in which intrinsic α C-helix stabilization transforms them into dimer-dependent “super acceptor” kinases^{3,4}.

In 0.5% FBS conditions with minimal EGF-ligand present, the potency of afatinib on EGFR-KDD is approximately equivalent in the absence (0 µg/ml) and presence (10 µg/mL) of cetuximab ($EC_{50} = 0.103 \pm 0.035$ nM and 0.095 ± 0.040 nM, respectively). Similar results are observed in Ex19Del ($EC_{50} = 0.061 \pm 0.027$ nM and 0.060 ± 0.017 nM, respectively). The near complete ablation of Ba/F3 L858R viability at higher concentrations of cetuximab mask any potential similar effects. Generally, we observe that Ex19Del and L858R are more sensitive to afatinib than is EGFR-KDD (Fig. 5b and Supplementary Table 4), consistent with our phosphorylation assays (Fig. 5a and Supplementary Fig. 7a).

“, and the case for the antibody/TKI combination being different for KDD compared with other mutations is just not convincing.”

- We thank the reviewer for this comment. In the initial set of comments from the first version of the manuscript, Reviewer #1 states: “The final data slide supports this – but the authors do not provide any evidence to suggest that this combination is more advantageous here than it is for L858R for example (see PMID 24063894). We note that the reviewer did agree with and acknowledge in his/her statement that our data (shown in the original manuscript) supported our conclusion regarding the efficacy of combined antibody + TKI therapy against EGFR-KDD.

The absence of this quantitative comparison makes the punchline fall a little flat, since similar antibody/TKI combinations are being tried (including in clinical trials) for a variety of mutated EGFR settings.”

- We have now provided quantitative comparisons of Ba/F3 cell viability when stably expressing EGFR-KDD, Ex19Del (not explicitly requested but included for completeness), and L858R in the presence and absence of cetuximab and/or afatinib in various combinations of serum starvation and exogenous EGF. These can be found in revised Fig. 5 and Supplementary Fig. 7.

▪ **Revised Figure 5:**

Revised Figure 5 Legend:

Fig. 5 | Inhibition of EGFR-KDD is maximally achieved by blocking both intra- and inter-molecular dimerization

a, YAMC cells were starved for 12 hours and treated with afatinib (10 nM in serum-free medium) and cetuximab (10 µg/ml in serum-free medium) for 3 hours 45 minutes, and then were treated with EGF (50 ng/mL in serum-free medium) for 15 minutes. The cells were harvested and analyzed by Western blot.

b, Cell Viability Assay was performed in mL3-independent Ba/F3 cells stably expressing EGFR-KDD, Ex19Del and L858R supplemented with 0.5% FBS. 5,000 cells were seeded in 96-well plate with the treatment of afatinib and cetuximab. Three days after incubation, CellTiter-Blue Reagent was added, and the fluorescence was detected at 560_{EX}/590_{EM} with a Synergy HTX microplate reader (BioTek Instruments, Winooski, VT, USA).

c, Cell Viability Assay was performed in mL3-independent Ba/F3 cells stably expressing EGFR-KDD, Ex19Del and L858R supplemented with 10% FBS.

• **Revised Supplementary Figure 7:**

• **Revised Supplementary Figure 7 Legend:**

Supplementary Fig. 7 | Inhibition of EGFR-KDD is maximally achieved by blocking both intra- and inter-molecular dimerization.

a, Quantification of YAMC antibody/TKI treatment Western blots; blue columns indicate mean pEGFR/EGFR and error bars indicate standard deviations (n=3).

b, Ba/F3 cell growth at different concentration of fetal bovine serum (FBS). 5,000 cells were seeded in 96-well plate with the treatment of afatinib and cetuximab. Three days after incubation, CellTiter-Blue Reagent was added, and the fluorescence was detected at 560EX/590EM with a Synergy HTX microplate reader (BioTek Instruments, Winooski, VT, USA).

c, Cell Viability Assay was performed in mL3-independent Ba/F3 cells stably expressing EGFR-KDD, Ex19Del and L858R in RPMI1640 supplemented with 10% FBS.

d, Cell Viability Assay was performed in mL3-independent Ba/F3 cells stably expressing EGFR-KDD, Ex19Del and L858R in RPMI1640 supplemented with 10% FBS and 5ng/mL EGF.

e, Cell Viability Assay was performed in mL3-independent Ba/F3 cells stably expressing EGFR-KDD, Ex19Del and L858R in RPMI1640 supplemented with 10% FBS and 50ng/mL EGF.

- To satisfy the Reviewer's comments on quantification and standard deviation, we have also added quantification of the **Fig. 5a** YAMC Western blot replicates (mean and standard deviation) as **Supplementary Fig. 7a**.
- We explained with detailed references to both pre-clinical and clinical literature that antibody/TKI combinations are *not* the standard of care treatment for either Ex19Del or L858R. *Results from the clinical trial of afatinib + cetuximab even disagree with the preclinical study suggesting otherwise (PMID 24063894) mentioned by the reviewer.* In other words, the best available treatment for patients with L858R or Ex19Del-mediated cancer is TKI alone. Thus, the need to demonstrate an improvement of antibody/TKI relative to Ex19Del and/or L858R is not clinically relevant. What we did was demonstrate that EGFR-KDD is more effectively treated with antibody/TKI than with either alone *and* identify a mechanistic basis for this effect, suggesting that this may be a worthwhile combination to evaluate in further pre-clinical and clinical trials. **We have addressed a related comment with substantial revision to the manuscript text as discussed in detail below (Pages 14 – 18) in response to query #7.**

Comment on Reproducibility

- III. As a general comment, were the blots in Figure 1c, 2e and g, 3a and b, 4a and c, and 5a all repeated? If so, how many times? It is typical in such studies to present quantitation and report standard deviations. Repeats and reproducibility are not currently mentioned anywhere, which is an important concern. Were biological repeats performed.
- These replicates do exist. Please see the Summary of Experimental Replicates table at the end of this document. We also have compiled all the raw replicates in the Source Data file.
 - Notably, our experiments were not only repeated for biological replicates, but we also show that our experimental results are recapitulated in different model systems. For example, we performed *in vitro* experiments in two different cell lines (YAMC and NR6), and the biological behavior of the various EGFR-KDD mutants we expressed was consistent across both cell lines.
 - We completed the required forms related to rigor and reproducibility on submission to *Nature Communications*.
 - With respect to quantification of Western blots, based on our experience and review of the literature, Western blot quantification and/or statistical analysis is not typically the expected standard. For example, in high-profile academic journals including *Nature Communications* (PMID: 28695896, 24476626) and *Cancer Cell* (PMID: 30449325, 26996308), Western blot quantification was not presented. Indeed, in the manuscript referenced by the Reviewer (PMID 24063894), Western blot quantification was not included in the published manuscript. However, to satisfy the reviewer's comments, we now include such quantification in the Source Data file as well as in Figures and Supplementary Figures where necessary.

Other comments

#1. The authors might want to reword the sentence "These data suggest that the N-terminal lobe TKD1 activates the C-terminal lobe TKD2, but not the reverse." It is not clear what it means. It is the C-lobe that 'does the activation'. Perhaps they mean: 'These data suggest that the N-lobe-mutated TKD1 in Fig. 1a can activate the C-lobe-mutated TKD2, but not the reverse.'?

- We thank the reviewer for catching this. It has been adjusted accordingly.
 - **Revised text, page 5:** These data suggest that the N-lobe-mutated TKD1 can activate the C-lobe-mutated TKD2, but not the reverse (Fig. 1a).

#2. It is misleading, and perhaps slightly dishonest, to say (as the authors do on page 5) that "Addition of cetuximab, an anti-EGFR extracellular domain antibody that blocks EGFR dimerization, effectively mitigates

EGF-induced phosphorylation...”. This antibody binds squarely to the EGF binding site, and blocks EGF binding (that’s how it was first selected), so of course it will – regardless of its influence on dimerization. If EGF cannot bind, EGF cannot activate the receptor. If cetuximab is bound, EGF cannot bind. Yes, cetuximab will inhibit dimerization, but there can be no doubt that cetuximab blocks EGF-induced activation of EGFR by blocking EGF binding.

Given all of this, the ability of cetuximab to inhibit EGF-induced activation of KDD just means that EGF binding is required for EGF to activate KDD. It says nothing about dimerization.

The authors need to comment that cetuximab blocks EGF binding, as pointed out in the last round of review. In their rebuttal, the authors puzzlingly justify their selective comment about cetuximab’s mechanism by stating that “...our focus is on EGF-dependent dimerization...”. This simply does not make sense as a justification, because there will be none if cetuximab is bound, because it binds and blocks the EGF-binding site (and with much higher affinity than EGF does).

- We appreciate the Reviewer’s concerns and have revised the manuscript text accordingly:
Revised text, page 6: To differentiate between EGFR-KDD activity caused by EGF-dependent inter-molecular dimerization and EGF-independent intra-molecular dimerization, we utilized cetuximab, an anti-EGFR extracellular domain antibody that blocks EGF-mediated EGFR dimerization⁵. EGF binding leads to inter-molecular dimerization of EGF receptors. Cetuximab prevents EGF binding by blocking the EGF binding site. These data suggest that EGF stimulation may promote EGFR-KDD activity through the formation of at least inter-molecular dimers; however, cetuximab does not preclude the formation of dimers entirely.
- In addition, to alleviate the Reviewer’s primary concern regarding conflating EGF binding and dimerization, we have also performed new experiments with the mAb806 antibody, which inhibits EGFR dimerization through binding to the EGFR extracellular domain II without interfering the ligand-binding site in extracellular domain III⁶. Our data showed that mAb806 was able to reduce the phosphorylation of EGFR-KDD in the presence of EGF ligand (**Supplementary Fig. 4c and 4d**), indicating that the dimerization was induced by EGF binding. To satisfy the Reviewer’s concern about quantification and standard deviations, we also provided quantification here and in the Source Data file.
- **Revised Supplementary Fig. 4:**
 - See page 9 for the actual figure; the image was placed on the subsequent page for clarity of presentation of the data

• **Revised Supplementary Fig. 4 Legend:**

Supplementary Fig. 4 | Disruption of EGF-induced inter-molecular activation of EGFR-KDD with cetuximab and mAb806

a, NR6 cells were cultured in serum-free medium for 36 hrs and then treated with 50ng/mL EGF ligand for 5min. Total EGFR and the autophosphorylation at three tyrosine sites were assessed by Western blot.

b, NR6 cells were starved overnight and treated with cetuximab (10 µg/ml in serum-free medium) for 3hrs 45min, and then were treated with EGF (50 ng/mL in serum-free medium) and cetuximab (10 µg/ml in serum-free medium) for 15min, then cells were harvested for western blot.

c, YAMC EGFR-WT and EGFR-KDD cells were starved for 12 hrs and pre-treated with mAb806 antibody (10 µg/ml in serum-free medium) for 3hrs 45min, respectively, and EGF ligand (50 ng/mL in serum-free medium) was added for 15min. The cells were harvested and analyzed by Western blot (left panel). The ratio of phospho-EGFR (Y1068) to total EGFR expression was also shown (right panel). Results represent the means of three independent experiments ± SD.

d, YAMC EGFR-KDD cells were starved for 12 hrs and pre-treated with cetuximab (10 µg/ml in serum-free medium) and mAb806 antibody (10 µg/ml in serum-free medium) for 3hrs 45min, respectively, and EGF ligand (50 ng/mL in serum-free medium) was added for 15min. The cells were harvested and analyzed by Western blot (left panel). The ratio of phospho-EGFR (Y1068) to total EGFR expression was also shown (right panel). Results represent the means of three independent experiments ± SD.

- We have added new text to the manuscript to describe our new data:
 - **New text, page 7:** “To further test the hypothesis that EGF stimulation promotes the formation of at least inter-molecular dimers in EGFR-KDD, we administered mAb806 to YAMC EGFR-KDD cells. The mAb806 antibody inhibits EGFR dimerization by binding to extracellular domain II (residues 287–302)⁶, rather than the EGF ligand binding site in domain III⁵. Thus, inhibition with mAb806 is highly complementary to similar experiments performed with cetuximab. As expected based on our cetuximab results, we found that mAb806 had no impact on phosphorylation level in the absence of EGF ligand (Supplementary Fig. 4b, lane 1, 3) and decreased the level of phosphorylation with EGF-ligand stimulation (Supplementary Fig. 4b, lane 4, 6). We also note that phosphorylation was reduced more by cetuximab than mAb806 at approximately equimolar concentrations, consistent with previous reports that the EGFR inhibitory potency of mAb806 is considerably lower than cetuximab⁷.”

#3. Related to point 2 above, the data in Figure 3b do not demonstrate that EGF stimulation promotes EGFR-KDD activity through the formation of intermolecular dimers. This is almost certainly what happens, although it could be tetramers or other oligomers.

- We agree that this could be the result of the formation of higher order oligomers. The above revised text (in response to point #2) now includes “at least inter-molecular dimers” instead of just “dimers”.

But, the experiment shown here simply does not show that, and cannot be interpreted in this way for the reasons mentioned in 2 above.

- This has been addressed in our response to #2 above in which we provide additional experiments with mAb806 (Page 9, Revised Supplementary Figure 4).

#4. Do the authors have an explanation for the finding in Figure 3b as to why (if the result is reproducible) cetuximab appears to activate KDD, apparently quite substantially?

- We thank the reviewer for this comment. We performed three independent experiments (page 11), to test the effects of cetuximab on EGFR-KDD. In our other independent experiments (see **Summary of Repeat Experiments** on pages 21 – 23), we clearly observe that cetuximab treatment does not increase EGFR-KDD phosphorylation in the absence of EGF. In the current Fig. 3b (lane 6 vs lane 5), the difference in phosphorylation is likely due to the higher total EGFR level, as we can see actin and total EGFR in lane 6 is stronger than lane 5. Therefore, we interpreted this as normal technical variation.
- While we generally do not believe that Western blot quantification and statistical analysis are best practice, to satisfy the Reviewer’s comments about quantification and reproducibility, we quantified all 3 biological replicates. We performed an analysis of variance (ANOVA) test on pEGFR(Y1068)/EGFR and observed significant differences between groups. We followed the ANOVA up with independent t-tests assuming unequal variances between groups. The alpha value was Bonferroni-corrected for multiple comparisons ($\alpha=0.0083$ to compare all groups to the untreated/no-EGF EGFR-KDD lane). The comparison between EGFR-KDD without treatment in the absence of EGF and with cetuximab-only treatment in the absence of EGF has a P-value of 0.26, indicating that there is no statistically significant difference between EGFR-KDD with no treatment and with cetuximab as the reviewer was wondering.
- To avoid further confusion for future audiences, we replaced the current Fig. 3b with Independent experiment #3 in the main text.

• **Revised Fig. 3b:**

#5. Despite the authors' comments, this reviewer does not consider there to be sufficient evidence for the 'side-by-side; model in Figure 3d. Needham et al. presented it as a model. If the authors are to re-present this model, they should include some data to support it over the more reasonable TKD1'/TKD2 and TKD2'/TKD1 asymmetric dimers suggested by all the experimental evidence.

- We thank the reviewer for the discussion on potential inter-molecular dimer models of EGFR-KDD. In the reviewer's original comment, he/she said, "It is not clear that the model in Fig. 3d has any support. Is it needed? It seems clear that 3c can happen, and there is really no evidence in the literature for activation (of TKD3) in a side-by-side manner, so 3d seems very speculative (and probably should be deleted)." In response we provided literature evidence combining long-timescale molecular dynamics (MD) simulations, fluorescence spectroscopy, and experimental biochemistry published by Needham et al. in *Nature Communications* that supported this model. We also acknowledged that our language was too strong previously. We therefore modified the text to soften our language as follows:
 - i. **Original text, page 7:** "In the side-by-side model, active-state stabilization of TKD3 occurs through a non-canonical asymmetric dimerization event with TKD1 (inter-molecular donor, intra-molecular donor) (Fig. 3d)."
 - ii. **Revised text, page 7:** "In the side-by-side model, active-state stabilization of TKD3 could occur through sterically impaired inactivation by TKD1 (inter-molecular donor, intra-molecular donor) (Fig. 3d), as observed in the 40 μs MD simulation of the EGFR-WT full-length tetramer model in Needham et al. 2016."

"Despite the authors' comments, this reviewer does not consider there to be sufficient evidence for the 'side-by-side; model in Figure 3d."

- We respectfully disagree with the reviewer on this point. The EGFR-WT tetramer model, on which the EGFR-KDD inter-molecular dimer model from Fig. 3d is based, was sufficiently validated to be published in *Nature Communications* (Needham et al., 2016, PMID 27796308). Indeed, *Nature Communications* published a 2018 follow-up study to Needham et al. that again utilized MD simulations, fluorescence spectroscopy, and experimental biochemistry to study EGFR oligomerization (see Zanetti-Domingues et al., 2018, *Nature Communications*, PMID: 30337523). Both of these papers are well-cited (combined > 100 citations total per Google Scholar) and represent the cutting edge of EGFR oligomerization biology.

"Needham et al. presented it as a model."

- We agree with the reviewer that Needham et al. did indeed present their EGFR-WT tetramer structure as a model. We have now revised our text to stress that there are no experimental structures of EGFR-WT

tetramer, and as such all published representations are just models, including the model that the reviewer prefers:

- iii. **Revised text, page 7:** “There are currently no experimental structures (e.g. from X-ray crystallography or cryogenic electron microscopy) elucidating the organization of EGFR-WT tetramer or EGFR-KDD inter-molecular dimer. Thus, we built our template-based models of EGFR-KDD intracellular inter-molecular dimer on two published EGFR-WT tetramer models both of which have experimental and computational support.”

#6. The authors state that ‘Taken together, these data suggest that TKD3 is catalytically active in the inter-molecular dimer’. It seems that TKD3 can be active in activity of Dead2 and C1 (with EGF), but surely not in Dead1 or C2 (with EGF). The results surely argue that either kinase can be active, depending on which mutant is studied.

- We agree with the Reviewer on this point, and we think that the contention is a result of lack of clarity in our text. We have rewritten this paragraph to provide clarity:
 - **Revised text, page 7:** We previously observed that Dead² (TKD2 and TKD4 are inactive), but not Dead¹ (TKD1 and TKD3 are inactive), ablates EGFR-KDD activity in the absence of EGF (**Fig. 1c, lane 3, 5, and 8**). Here, we see that EGF-ligand stimulation robustly revives phosphorylation in Dead²; **Fig. 3a, lane 15, 16; Fig. 3b, lane 21, 23; Supplementary Fig. 3a, lane 15, 16**), suggesting active-state stabilization of TKD3 through the formation of at least inter-molecular dimers (**Fig. 3c – d**). Less dramatic increases in Dead¹ from baseline intra-molecular dimer phosphorylation are consistent with changes due to ligand-induced EGFR recruitment (**Fig. 3a, lane 9, 10; Supplementary Fig. 3a, lane 9, 10**). Consistent with these results, pre-administration with cetuximab prevents EGF-dependent phosphorylation of Dead² and has only a minor impact on Dead¹ phosphorylation. Taken together, these data suggest that in addition to activation of TKD2 and TKD4 by TKD1 and TKD3, respectively, TKD3 becomes catalytically active in the inter-molecular dimer.

#7. The authors need to state clearly in the text that Figure 5b was performed in the presence of a high EGF concentration, and the inhibitory effect is almost entirely from cetuximab (which blocks EGF binding). Since 50 ng/ml EGF was added, how is this relevant for KDD in cancer?

- We thank the Reviewer for these comments. In our revised manuscript, we have performed all quantitative viability studies at multiple concentrations of EGF: serum starved (0.5% FBS), 10% FBS, 10% FBS + 5 ng/ml EGF, and 10% FBS + 50 ng/ml EGF. Each of these experiments was performed with three independent replicates. Each independent replicate contained three technical replicates. **See revised Fig. 5 and Supplementary Fig. 7** and corresponding figure legends (provided above in response to query II of the “Reviewer 1 (Remarks to Author)” section (Pages 3 – 5).
- We also note that an EGF concentration of 50 ng/ml has been utilized in other peer reviewed manuscripts (PMID: 29748209, PMID: 19560417). Indeed, concentrations as high as 100 ng/ml have been used in EGFR studies by prominent experts, including Dr. John Kuriyan (for example, please see Huang et al. eLife, PMID: 27017828).
- To the best of our knowledge, the concentration of EGF in patient tumor samples is not known and likely varies substantially based on patient specific factors (age, gender, co-morbid conditions, medications, underlying organ function), tumor intrinsic factors (type of tumor, site of the tumor – primary vs. metastasis, tumor genomics, tumor RNA processing, trafficking within the tumor), and tumor microenvironment (organ site of the primary tumor vs. metastasis, proximity to blood vessels, etc.)
- As a result, we used the published, peer reviewed literature from experts in the EGFR field to guide selection of the EGF concentrations used in our experiments. We welcome suggestions from the Reviewer for future studies regarding rational, tumor-specific concentrations of EGF to utilize in our experiments.

The authors should also note the long-known effect of cetuximab on EGFR downregulation.

- We thank the Reviewer for this comment. In the revised manuscript, we have included a statement and citation on the downregulation of EGFR in the presence of cetuximab.
 - **Revised text, page 12:** It has been well-recognized that cetuximab induces degradation of EGFR mutants in different NSCLC cells^{8,9}. In this study, no degradation of EGFR-Ex19Del, L858R and EGFR-KDD levels were observed in YAMC (**Fig. 5a**) and NR6 cells (**Supplementary Fig. 4b**), probably due to the shorter treatment time than previous studies (4hrs versus 24 – 72hrs)^{8,9}.

Crucially, for the (no-EGF) data in Figure S6, the absence of experiments with L858R or Ex19 leave the reader unconvinced that they would not look just the same. The absence of these additional/comparative experiments (in S6b or 5b), and the use of EGF in Fig 5b, result in there really being no data here to suggest that cetuximab plus TKI will be any more useful for KDD than any other mutants.

- We thank the reviewer for this comment. First, as discussed above in response to query II of “Reviewer 1 (Remarks to Author)” section (Pages 6 – 7) and at length in our initial response, the comparison of EGFR-KDD to L858R or Ex19Del is not clinically valuable. The clinical trials to which the Reviewer previously alluded to in order to justify these experiments found no benefit (in terms of PFS, intracranial response, and OS) of antibody/TKI combination compared to TKI alone. TKI, and more specifically osimertinib, remains the standard-of-care for patients with tumors harboring EGFR L858R and EGFR Ex19Del mutations. We have added text to the manuscript to emphasize this critical piece of information:
 - **New text, page 9:** “The dual nature of EGFR-KDD as an EGF-independent active intra-molecular dimer and as an EGF-dependent active inter-molecular dimer/multimer poses a unique therapeutic challenge. Our computational models and experimental data suggest that the ideal therapy would simultaneously reduce intra- and inter-molecular dimer-mediated activity. One potential treatment strategy is therefore the combination of cetuximab with a TKI (here afatinib). Prior pre-clinical literature has suggested that such a combination may be effective in L858R but not Ex19Del¹⁰. The combination of cetuximab with various EGFR TKIs, including gefitinib¹¹ and afatinib^{12,13}, has been tested in lung cancer patients. In a phase I trial, no responses were observed with the combination of cetuximab plus gefitinib¹¹, and therefore has not been subsequently used in patients. The combination of cetuximab plus afatinib has advanced in the clinic, including a phase I trial (NCT01090011) that included an expansion cohort^{12,13}. Results from this trial of cetuximab plus afatinib demonstrated that the combination therapy was effective in achieving tumor reduction (as assessed by CT scans using RECIST criteria) in patients with both Ex19Del and L858R EGFR-mutant lung cancer, in contrast to prior pre-clinical data¹⁰. Importantly, the combination of cetuximab plus TKI is not FDA-approved because there was no benefit (in terms of PFS, intracranial response, and OS) compared to TKI alone, and thus not standardly used in the treatment of patients with Ex19Del or L858R mutations. The current standard of care for these patients is the mutant-selective EGFR TKI, osimertinib, based on a seminal phase 3 clinical trial^{14,15}. In contrast, no pre-clinical study or clinical trial has evaluated antibody/TKI combination vs. either alone in EGFR-KDD patients. Indeed, the index patient for EGFR-KDD described in Gallant et al. 2015 unfortunately only had a partial response to afatinib. The anti-tumor response was short-lived (7 cycles of afatinib, or approximately 7 months) before the patient developed acquired resistance to afatinib driven by amplification of the EGFR-KDD allele (Gallant et al. 2015). Collectively, these observations suggested that more potent EGFR blockade is necessary to overcome the oncogenic activity of EGFR-KDD. Here, we test the hypothesis that combined TKI and cetuximab treatment will reduce EGFR-KDD-mediated phosphorylation *in vitro* more than either treatment alone.”
- Second, in response to the Reviewer’s request for a comparison to L858R, we previously performed additional experiments in the presence and absence of EGF stimulation for both Ex19Del and L858R in our last round of revision. Here, we also add quantitation of our replicates and revise the accompanying text:

- **Revised text, page 9:** “We treated YAMC cells stably expressing EGFR Ex19Del (E746_A750del), L858R, and EGFR-KDD with afatinib and cetuximab both in the absence and presence of EGF ligand (Fig. 5a, Supplementary Fig. 7a). Importantly, we observed that in both the absence and presence of EGF, afatinib resulted in a near complete ablation of p-EGFR in Ex19Del (Fig. 5a, lanes 1, 2, 5, 6) and L858R (Fig. 5a, lanes 9, 10, 13, 14), but substantial residual phosphorylation existed in EGFR-KDD (Fig. 5a, lanes 17, 18, 21, 22). As expected, cetuximab alone reduced phosphorylation in Ex19Del, L858R and EGFR-KDD in the presence of EGF ligand (Fig. 5a, lane 7, 15, 23). Notably, the greatest reduction of phosphorylation for EGFR-KDD occurred with the combination of cetuximab + afatinib in the presence of EGF (Fig. 5a, lanes 21, 22, 23, 24). These data suggest that phosphorylation of EGFR Ex19Del and L858R is abolished by afatinib (TKI) or cetuximab alone, and addition of cetuximab to afatinib does not add substantially more inhibition to the decrease in auto-phosphorylation. Unlike EGFR Ex19Del and L858R, phosphorylation of EGFR-KDD is inhibited by both afatinib and cetuximab as single agent, but the combination treatment yielded more inhibitory effects.”
- Finally, we have now also added full quantitative viability comparisons of EGFR-KDD, Ex19Del and L858R in Ba/F3 cells, and we have added new text to accommodate the additional data. Please see the corresponding revised **Fig 5** and **Supplementary Fig 7** in response to query II in the “Reviewer 1 (Remarks to Author)” section above (Pages 4 – 7) to see the new data.

New text, page 9: We also performed viability assays with BaF3 cells stably expressing EGFR-KDD, Ex19Del (E746_A750del) or L858R. First, we evaluated Ba/F3 cell growth in serum starved (0.5% fetal bovine serum; FBS) conditions to minimize EGF activation (Supplementary Fig. 7c). At 0.5% FBS, cetuximab maximally exhibited ~40% inhibition of EGFR-KDD, ~80% inhibition of Ex19Del, and almost 100% inhibition of L858R cell viability (Fig. 5b and Supplementary Table 3). These data are consistent with a model in which EGFR-KDD retains an active intra-molecular dimer in the absence of EGF stimulation (Fig. 1) and previously published models of Ex19Del and L858R in which intrinsic α C-helix stabilization transforms them into dimer-dependent “super acceptor” kinases^{3,4}. Indeed, progressively higher concentrations of FBS and the addition of exogenous EGF resulted in stable or increased viability of all mutants in the presence of cetuximab, though EGFR-KDD proved to be the least inhibited (Fig. 5c, Supplementary Fig. 7d – f, and Supplementary Table 3).

In 0.5% FBS conditions with minimal EGF-ligand present, the potency of afatinib on EGFR-KDD is approximately equivalent in the absence (0 μ g/ml) and presence (10 μ g/ml) of cetuximab ($EC_{50} = 0.103 \pm 0.035$ nM and 0.095 ± 0.040 nM, respectively). Similar results are observed in Ex19Del ($EC_{50} = 0.061 \pm 0.027$ nM and 0.060 ± 0.017 nM, respectively). The near complete ablation of Ba/F3 L858R viability at higher concentrations of cetuximab mask any potential similar effects. Generally, we observe that Ex19Del and L858R are more sensitive to afatinib than is EGFR-KDD (Fig. 5b and Supplementary Table 4), consistent with our phosphorylation assays (Fig. 5a and Supplementary Fig. 7a).

As the concentration of EGF-ligand in the medium is increased, we observe not only an increase in viability with cetuximab and increased EC_{50} of afatinib, but also a greater potentiation of afatinib by cetuximab (Fig. 5b – c and Supplementary Fig. 7d – f). In 10% FBS + 50 ng/ml exogenous EGF, we observe a 5.8x increase in afatinib potency transitioning from 0 μ g/ml to 10 μ g/ml in Ba/F3 EGFR-KDD. We also observe potentiation of afatinib in Ex19Del (4.7x) and L858R (3.7x) (Supplementary Fig. 7f and Supplementary Table 4). Compared to Ex19Del and L858R, the larger potentiation of afatinib inhibition of Ba/F3 EGFR-KDD by cetuximab seems to be mediated by the lower inhibition of EGFR-KDD by afatinib. Together, our data suggests that a lower dose of afatinib can be administered to maximally inhibit EGFR-KDD when supplemented with cetuximab.

- We have also added supplementary tables describing the viability (**New Supplementary Table 3**) and afatinib EC_{50} (**New Supplementary Table 4**) at multiple concentrations of cetuximab in multiple serum/EGF combinations.

New Supplementary Table 3 | Viability of BaF3 EGFR-KDD, Ex19Del and L858R cells at different concentration of cetuximab (N=3^a, mean ± sd)

Cetuximab (µg/mL)	RPMI1640 + 0.5% FBS			RPMI1640 + 10% FBS			RPMI1640 + 10% FBS + 5ng/mL EGF			RPMI1640 + 10% FBS + 50ng/mL EGF		
	EGFR-KDD	Ex19Del	L858R	EGFR-KDD	Ex19Del	L858R	EGFR-KDD	Ex19Del	L858R	EGFR-KDD	Ex19Del	L858R
0	100	100	100	100	100	100	100	100	100	100	100	100
0.00001	102.23 ± 1.65	104.87 ± 6.02	100.19 ± 4.45	97.54 ± 1.30	101.72 ± 9.61	96.86 ± 7.91	100.20 ± 2.08	102.09 ± 1.04	103.57 ± 4.26	NT	NT	NT
0.0001	104.60 ± 5.34	102.62 ± 7.75	96.96 ± 4.29	92.74 ± 5.78	95.38 ± 3.52	101.25 ± 3.11	95.21 ± 2.99	99.41 ± 4.60	102.04 ± 4.98	NT	NT	NT
0.001	110.95 ± 7.60	100.63 ± 10.90	91.00 ± 7.40	92.66 ± 15.99	95.90 ± 9.68	101.46 ± 5.11	95.75 ± 9.24	99.52 ± 3.51	104.71 ± 1.50	NT	NT	NT
0.01	89.24 ± 4.75	42.65 ± 13.74	15.15 ± 13.26	66.71 ± 11.73	68.01 ± 14.09	50.62 ± 18.57	88.87 ± 6.71	100.71 ± 2.31	100.54 ± 7.81	NT	NT	NT
0.1	60.95 ± 5.35	39.66 ± 18.78	1.75 ± 0.37	49.12 ± 9.82 ^a	19.72 ± 10.04 ^a	0.48 ± 0.32 ^a	70.98 ± 3.78	84.88 ± 3.29	69.87 ± 20.42	104.06 ± 13.59	101.55 ± 0.99	102.33 ± 2.41
1.0	57.05 ± 1.66	34.14 ± 14.27	2.83 ± 2.33	45.29 ± 7.25 ^a	12.34 ± 3.44	0.46 ± 0.25	67.53 ± 2.15	44.96 ± 6.80	17.24 ± 5.30	104.62 ± 7.79	78.66 ± 3.06	90.93 ± 8.60
10	61.01 ± 2.54	35.19 ± 14.37	2.83 ± 2.19	45.84 ± 6.49 ^a	12.67 ± 5.07	1.45 ± 1.85	68.66 ± 1.23	46.31 ± 6.60	5.11 ± 2.60	104.81 ± 9.74	46.59 ± 4.58	66.04 ± 4.68

NT, not tested; a, N=6

New Supplementary Table 4 | EC50 of afatinib (nM) in different concentration of cetuximab in BaF3 cells (N=3^a, mean ± sd)

Cetuximab (µg/mL)	RPMI1640 + 0.5% FBS			RPMI1640 + 10% FBS			RPMI1640 + 10% FBS + 5ng/mL EGF			RPMI1640 + 10% FBS + 50ng/mL EGF		
	EGFR-KDD	Ex19Del	L858R	EGFR-KDD	Ex19Del	L858R	EGFR-KDD	Ex19Del	L858R	EGFR-KDD	Ex19Del	L858R
0	0.103 ± 0.035	0.060 ± 0.017	0.020 ± 0.013	0.456 ± 0.168 ^a	0.340 ± 0.236 ^a	0.088 ± 0.043 ^a	0.357 ± 0.119	0.410 ± 0.062	0.466 ± 0.242	1.762 ± 1.715	0.951 ± 1.033	0.626 ± 0.551
0.00001	0.101 ± 0.009	0.069 ± 0.033	0.024 ± 0.019	0.344 ± 0.132	0.243 ± 0.201	0.103 ± 0.049	0.358 ± 0.107	0.408 ± 0.082	0.525 ± 0.212	NT	NT	NT
0.0001	0.104 ± 0.021	0.061 ± 0.008	0.020 ± 0.015	0.463 ± 0.135	0.240 ± 0.178	0.101 ± 0.039	0.349 ± 0.114	0.401 ± 0.046	0.448 ± 0.164	NT	NT	NT
0.001	0.077 ± 0.021	0.060 ± 0.019	0.021 ± 0.019	0.269 ± 0.135	0.229 ± 0.175	0.085 ± 0.038	0.333 ± 0.067	0.420 ± 0.076	0.483 ± 0.115	NT	NT	NT
0.01	0.027 ± 0.009	0.052 ± 0.025	0.002 ± 0.002	0.208 ± 0.063	0.128 ± 0.083	0.013 ± 0.012	0.235 ± 0.081	0.331 ± 0.075	0.353 ± 0.118	NT	NT	NT
0.1	0.104 ± 0.050	0.062 ± 0.027	0.002 ± 0.003	0.267 ± 0.110 ^a	0.110 ± 0.087 ^a	0.003 ± 0.002	0.220 ± 0.044	0.143 ± 0.059	0.118 ± 0.024	0.832 ± 0.803	0.453 ± 0.451	0.362 ± 0.310
1.0	0.098 ± 0.013	0.050 ± 0.017	0.001 ± 0.002	0.361 ± 0.104 ^a	0.207 ± 0.157 ^a	NA	0.183 ± 0.086	0.150 ± 0.055	0.025 ± 0.020	0.501 ± 0.476	0.630 ± 0.674	0.404 ± 0.366
10	0.095 ± 0.040	0.061 ± 0.027	0.003 ± 0.004	0.273 ± 0.102 ^a	0.095 ± 0.045 ^a	NA	0.189 ± 0.071	0.140 ± 0.039	0.011 ± 0.010	0.303 ± 0.293	0.204 ± 0.203	0.168 ± 0.134

NA, EC50 is not available due to the invalid drug dose-response curves.

NT, not tested.

a, N=6

Moreover, in Fig. 5a there is still clearly detectable KDD phosphorylation (with or without EGF) when cetuximab plus afatinib are added.

- We thank the reviewer for this comment. The presence of some remainder of phosphorylation is not at odds with the observed effect of a large (87%) reduction in phosphorylation when adding cetuximab plus afatinib (**Fig. 5a, lane 24 vs. 21**).

In fact, in both cases, one could argue (based on the gel shown – was this repeated or quantitated?)...

- We appreciate the reviewer's emphasis on reproducibility. The Western blot shown was completed in triplicate. We have also added a quantitation, per reviewer request. See above response query #7. Also see a breakdown of all replicates at the end of this document, as well as the Source Data file.

...that cetuximab actually elevates phosphorylation of KDD in the presence of afatinib (compare lanes 18 and 20 or 18 and 24), which seems to contradict the authors' assertions and would not suggest combination.

- We thank the reviewer for this astute observation. It is not surprising that cetuximab did not further reduce phosphorylation going from lane 18 to 20 since both of these are performed in the absence of EGF. As discussed above, cetuximab blocks EGF binding to reduce EGF-mediated activation of EGFR. Moreover, this effect was not observed in independent replicates (lane 18 and 20, page 15).
- Comparing lanes 18 (afatinib alone) to 24 (afatinib plus cetuximab) is not appropriate since lane 18 was performed in the absence of EGF and lane 24 had 50 ng/ml EGF stimulation. Thus, the net reduction in phosphorylation achieved in lane 24 is far greater than that achieved in 18, despite the fact that they look comparable. An appropriate comparison would be between lanes 22 (afatinib alone in the presence of EGF) and 24 (afatinib plus cetuximab in the presence of EGF). In the latter comparison, it is clear that cetuximab plus afatinib is more effective than afatinib alone, an effect that is not observed for Ex19Del (lanes 6 and 8) or L8585R (lanes 14 and 16).
- Also note that in response to query #4 that we provided 3 independent biological samples and a quantification that all demonstrate that cetuximab does *not* increase EGFR-KDD phosphorylation.

#8. References to Figures S5 and S6 are confused by the addition of the new S5 without corresponding corrections in the text.

- We thank the Reviewer for this comment. We have revised figures and corrected the figure citations accordingly.

Response to Reviewer #2

- No new comments from Reviewer 2 requiring response

Response to Reviewer #3

Reviewer #3 (Remarks to the Author):

I am thankful to the authors for their clear explanations. It is clear (also from the previous work) that they have a clear grasp of the methodologies used here. I also understand that we are working here at the limit of these techniques. I hope they will also be patient regarding any doubt I might have left (on the wording, not on the experiment themselves).

- We have appreciated the Reviewer's feedback and believe it has strengthened the manuscript considerably.

In fact, I still am not sure that I fully agree with some dichotomic views expressed in the manuscript (I am referring to the above KDD+HER3 w EGF for example). Looking at figure 4e, it is clear that f_c for KDD-HER with NRG1 is higher than with EGF (although many measurements for this sample have a f_c way above 0.1, also quite interesting). But is this enough to say that KDD does not heterodimerize (I am assuming "at all")? Is it maybe somewhere assumed that the K_d for this complexes is either infinitely large or infinitely small? I do not know if this might be relevant in the biological context, but what about the possibility of a moderate dimerization for these samples (giving rise to a mixture of monomers and heterodimers, with intermediate f_c values?). The same could be said of course for other samples, e.g. KDDxHER2 w/o EGF. Might e.g. "heterodimerize to a smaller/larger extent" be a preferable expression?

- We understand the Reviewer's concerns and have revised our text to soften our language:

Original text, page 8: "Moreover, we observed quantitatively with PIE-FCCS that EGFR-WT and EGFR-KDD heterodimerize with HER2 in the presence of EGF-ligand ($f_c = 0.10$ and $f_c = 0.16$, respectively) but not in its absence ($f_c = 0.00$ and $f_c = 0.06$, respectively) (Fig. 4d, Supplementary Fig. 5, Supplementary Table 2)."

Revised text, page 8: "Moreover, we observed quantitatively with PIE-FCCS that EGFR-WT and EGFR-KDD heterodimerize with HER2 to a larger extent in the presence of EGF-ligand ($f_c = 0.10$ and $f_c = 0.16$, respectively) than in its absence ($f_c = 0.00$ and $f_c = 0.06$, respectively) (Fig. 4d, Supplementary Fig. 6f, Supplementary Table 2)."

Original text, page 8: "Interestingly, our biophysical data suggest that like EGFR-WT, EGFR-KDD also heterodimerizes with HER3 in the presence of NRG1 but not EGF (Fig. 4e, Supplementary Fig. 5, Supplementary Table 2)."

Revised text, page 8: "Interestingly, our biophysical data suggest that like EGFR-WT, EGFR-KDD also heterodimerizes with HER3 in the presence of NRG1 to a greater extent than in the presence of EGF (Fig. 4e, Supplementary Fig. 6g, Supplementary Table 2)."

Regarding the considerations on brightness analysis, I fully agree with the authors. Apart from those suggestions regarding the modification in the wording of the discussion, I believe that the manuscript should be accepted for publication.

- We thank the Reviewer for his/her feedback and consideration of our manuscript.

References:

- 1 Jia, Y. *et al.* Overcoming EGFR(T790M) and EGFR(C797S) resistance with mutant-selective allosteric inhibitors. *Nature* **534**, 129-132, doi:10.1038/nature17960 (2016).
- 2 Zhang, L. *et al.* ERBB3/HER2 signaling promotes resistance to EGFR blockade in head and neck and colorectal cancer models. *Mol Cancer Ther* **13**, 1345-1355, doi:10.1158/1535-7163.MCT-13-1033 (2014).
- 3 Shan, Y. *et al.* Oncogenic mutations counteract intrinsic disorder in the EGFR kinase and promote receptor dimerization. *Cell* **149**, 860-870, doi:10.1016/j.cell.2012.02.063 (2012).
- 4 Red Brewer, M. *et al.* Mechanism for activation of mutated epidermal growth factor receptors in lung cancer. *Proceedings of the National Academy of Sciences of the United States of America* **110**, E3595-3604, doi:10.1073/pnas.1220050110 (2013).
- 5 Li, S. *et al.* Structural basis for inhibition of the epidermal growth factor receptor by cetuximab. *Cancer cell* **7**, 301-311, doi:10.1016/j.ccr.2005.03.003 (2005).
- 6 Sivasubramanian, A., Chao, G., Pressler, H. M., Wittrup, K. D. & Gray, J. J. Structural model of the mAb 806-EGFR complex using computational docking followed by computational and experimental mutagenesis. *Structure* **14**, 401-414, doi:10.1016/j.str.2005.11.022 (2006).
- 7 Reilly, E. B. *et al.* Characterization of ABT-806, a Humanized Tumor-Specific Anti-EGFR Monoclonal Antibody. *Mol Cancer Ther* **14**, 1141-1151, doi:10.1158/1535-7163.MCT-14-0820 (2015).
- 8 Doody, J. F. *et al.* Inhibitory activity of cetuximab on epidermal growth factor receptor mutations in non small cell lung cancers. *Mol Cancer Ther* **6**, 2642-2651, doi:10.1158/1535-7163.MCT-06-0506 (2007).
- 9 Perez-Torres, M., Guix, M., Gonzalez, A. & Arteaga, C. L. Epidermal growth factor receptor (EGFR) antibody down-regulates mutant receptors and inhibits tumors expressing EGFR mutations. *J Biol Chem* **281**, 40183-40192, doi:10.1074/jbc.M607958200 (2006).
- 10 Cho, J. *et al.* Cetuximab response of lung cancer-derived EGF receptor mutants is associated with asymmetric dimerization. *Cancer Res* **73**, 6770-6779, doi:10.1158/0008-5472.CAN-13-1145 (2013).
- 11 Ramalingam, S. *et al.* Dual inhibition of the epidermal growth factor receptor with cetuximab, an IgG1 monoclonal antibody, and gefitinib, a tyrosine kinase inhibitor, in patients with refractory non-small cell lung cancer (NSCLC): a phase I study. *J Thorac Oncol* **3**, 258-264, doi:10.1097/JTO.0b013e3181653d1b (2008).
- 12 Janjigian, Y. Y. *et al.* Dual inhibition of EGFR with afatinib and cetuximab in kinase inhibitor-resistant EGFR-mutant lung cancer with and without T790M mutations. *Cancer Discov* **4**, 1036-1045, doi:10.1158/2159-8290.CD-14-0326 (2014).
- 13 Horn, L. *et al.* Continued use of afatinib with the addition of cetuximab after progression on afatinib in patients with EGFR mutation-positive non-small-cell lung cancer and acquired resistance to gefitinib or erlotinib. *Lung Cancer* **113**, 51-58, doi:10.1016/j.lungcan.2017.08.014 (2017).
- 14 Soria, J. C. *et al.* Osimertinib in Untreated EGFR-Mutated Advanced Non-Small-Cell Lung Cancer. *N Engl J Med* **378**, 113-125, doi:10.1056/NEJMoa1713137 (2018).
- 15 Ramalingam, S. S. *et al.* Overall Survival with Osimertinib in Untreated, EGFR-Mutated Advanced NSCLC. *N Engl J Med* **382**, 41-50, doi:10.1056/NEJMoa1913662 (2020).

Summary of Repeat Experiments (#NCOMMS-20-11085A)

- This table annotates the number of biological replicates for experiments shown in the Main Figures and Supplemental Figures.
- However, we have been working on this project for 5 years (since our *Cancer Discovery* paper was published in 2015). As a result, there are numerous experiments that led up to the “final” experiments shown in the figures for the present manuscript.
- We would be happy to share the 5 years of experiments that provided the foundation for the “final figures” annotated below.

Figure	# biological replicates	Date of final read out of the experiment	Description of the experiment	Experimental Notes
Fig. 1c	3	 ▪ Replicate 1: 03/17/2017 ▪ Replicate 2: 03/29/2017 ▪ Replicate 3: 7/27/2020 	Western blot of EGFR-KDD intra-molecular dimer mutants in YAMC cells	
Fig. 1d	3	 ▪ Replicate 1: 05/24/2017 ▪ Replicate 2: 06/06/2017 ▪ Replicate 3: 06/13/2017 	Soft agar assay of EGFR-KDD intra-molecular dimer mutants in YAMC cells	Each independent experiment included 3 technical replicates for each sample tested.
Fig. 2e	3	 ▪ Replicate 1: 08/01/2019 ▪ Replicate 2: 02/13/2020 ▪ Replicate 3: 02/13/2020 ▪ For replicates #2 and #3, the cells were transfected on different days. 	Transient transfection of GGSx mutants into HEK293 cells	
Fig. 2g	3	 ▪ Replicate 1: 09/02/2019 ▪ Replicate 2: 09/11/2019 ▪ Replicate 3: 05/25/2019, 09/11/2019 	Transient transfection of linker mutations into HEK293 cells	
Fig. 3a	3	 ▪ Replicate 1: 06/15/2017 ▪ Replicate 2: 05/24/2017 ▪ Replicate 3: 02/21/2020 	Western blot of EGFR-KDD intra-molecular dimer mutants in YAMC cells with EGF treatment	
Fig. 3b	3	 ▪ Replicate 1: 02/19/2020 ▪ Replicate 2: 07/28/2020 ▪ Replicate 3: 08/18/2020 	Western blot of EGFR-KDD intra-molecular dimer mutants in YAMC cells with EGF and cetuximab treatment	
Fig. 4a	2	 ▪ Replicate 1: 10/02/2018 ▪ Replicate 2: 10/12/2018 	Co-Immunoprecipitation of EGFR-WT and EGFR-KDD by V5 antibody pulling-down	For each independent experiment, we performed the IPs with two different epitope tagged-antibodies and did two replicates for each tag.
Fig. 4c	2	 ▪ Replicate 1: 09/11/2018 ▪ Replicate 2: 07/13/2018 	Co-Immunoprecipitation of EGFR-KDD with EGFR-WT, HER2-WT and HER3-WT by V5 antibody pulling-down	For each independent experiment, we performed the IPs with two different epitope tagged-antibodies and did two replicates for each tag.

Fig. 5a	3	 ▪ Replicate 1: 05/08/2020 ▪ Replicate 2: 05/12/2020 ▪ Replicate 3: 07/11/2020 	YAMC EGFR-Ex19Del, L858R and EGFR-KDD cells with treatment of EGF, cetuximab and afatinib	Two technical replicates for biological replicate 1 and 2.
Fig. 5b	3	 ▪ Replicate 1: 08/24/2020 ▪ Replicate 2: 08/25/2020 ▪ Replicate 3: 08/26/2020 	Cell viability assay on BaF3 EGFR-KDD, Ex19Del and L858R cells supplemented with 0.5% FBS and treated with cetuximab and afatinib	Each independent experiment included 3 technical replicates for each sample tested.
Fig. 5c	3	 ▪ Replicate 1: 08/17/2020 for EGFR-KDD, 08/03/2020 for Ex19Del, 08/08/2020 for L858R ▪ Replicate 2: 08/18/2020 for EGFR-KDD, 07/20/2020 for Ex19Del and L858R ▪ Replicate 3: 09/04/2020 for EGFR-KDD, 07/28/2020 for Ex19Del and L858R 	Cell viability assay on BaF3 EGFR-KDD, Ex19Del and L858R cells supplemented with 10% FBS and treated with cetuximab and afatinib	Each independent experiment included 3 technical replicates for each sample tested.
Suppl Fig. 1a	2* (see comment in the "experimental notes" column)	 ▪ Replicate 1: 07/10/2017 ▪ Replicate 2: 06/26/2017 	Western blot of EGFR-KDD intra-molecular dimer mutants in NR6 cells	 ▪ Supplemental Figure 4a contain repeats of this exact experiment, with inclusion of new EGFR-KDD variants. ▪ Therefore, the experiment shown in Figure 1c has actually been repeated 5 independent times.
Suppl Fig. 1b	3	 ▪ Replicate 1: 02/20/2017 ▪ Replicate 2: 04/27/2017 ▪ Replicate 3: 05/18/2017 	Soft agar assay of EGFR-KDD intra-molecular dimer mutants in NR6 cells	Each independent experiment included 3 technical replicates for each sample tested.
Suppl Fig. 4a	3	 ▪ Replicate 1: 09/08/2017 ▪ Replicate 2: 07/18/2017 ▪ Replicate 3: 07/27/2017 	Western blot of EGFR-KDD intra-molecular dimer mutants in NR6 cells with EGF treatment	
Suppl Fig. 4b	3	 ▪ Replicate 1: 05/18/2018 ▪ Replicate 2: 06/26/2019 ▪ Replicate 3: 07/11/2020 	Western blot of EGFR-WT, L858R and EGFR-KDD in NR6 cells with EGF and cetuximab treatment	
Suppl Fig. 4c	3	 ▪ Replicate 1: 09/02/2020 ▪ Replicate 2: 09/15/2020 ▪ Replicate 3: 08/26/2020 	Western blot of EGFR-WT and EGFR-KDD in YAMC cells with EGF and mAb806 antibody treatment	
Suppl Fig. 4d	3	 ▪ Replicate 1: 09/16/2020 ▪ Replicate 2: 09/06/2020 ▪ Replicate 3: 09/09/2020 	Western blot of EGFR-KDD in YAMC cells with EGF, cetuximab and mAb806 antibody treatment	
Suppl Fig. 6a	2	 ▪ Replicate 1: 09/05/2018 ▪ Replicate 2: 10/09/2018 	Co-Immunoprecipitation of EGFR-WT and EGFR-KDD by Myc	For each independent experiment, we performed the IPs with two different

			antibody pulling-down	epitope tagged-antibodies and did two replicates for each tag.
Suppl Fig. 6c	2	 ▪ Replicate 1: 10/02/2018 ▪ Replicate 2: 07/18/2018 	Co-Immunoprecipitation of HER2-WT with EGFR-WT and EGFR-KDD by Myc antibody pulling-down	For each independent experiment, we performed the IPs with two different epitope tagged-antibodies and did two replicates for each tag.
Suppl Fig. 6d	2	 ▪ Replicate 1: 09/11/2018 ▪ Replicate 2: 07/18/2018 	Co-Immunoprecipitation of HER3-WT with EGFR-WT and EGFR-KDD by Myc antibody pulling-down	For each independent experiment, we performed the IPs with two different epitope tagged-antibodies and did two replicates for each tag.
Suppl Fig. 7b	3	 ▪ Replicate 1: 08/20/2020 ▪ Replicate 2: 08/21/2020 ▪ Replicate 3: 08/22/2020 	Ba/F3 EGFR-KDD, Ex19Del and L858R cell growth with different concentration of FBS	Each independent experiment included 3 technical replicates for each sample tested.
Suppl Fig. 7c	3	 ▪ Replicate 1: 07/20/2020 for EGFR-KDD, 07/15/2020 for Ex19Del and L858R ▪ Replicate 2: 07/25/2020 for EGFR-KDD, 07/17/2020 for Ex19Del and L858R ▪ Replicate 3: 08/03/2020 for EGFR-KDD, 07/13/2020 for Ex19Del and L858R 	Cell viability assay on BaF3 EGFR-KDD, Ex19Del and L858R cells supplemented with 10% FBS and treated with cetuximab and afatinib	Each independent experiment included 3 technical replicates for each sample tested.
Suppl Fig. 7d	3	 ▪ Replicate 1: 08/05/2020 for EGFR-KDD and Ex19Del, 08/22/2020 for L858R ▪ Replicate 2: 08/06/2020 ▪ Replicate 3: 08/07/2020 	Cell viability assay on BaF3 EGFR-KDD, Ex19Del and L858R cells supplemented with 10% FBS and treated with 5ng/mL EGF, cetuximab and afatinib	Each independent experiment included 3 technical replicates for each sample tested.
Suppl Fig. 7e	3	 ▪ Replicate 1: 07/20/2020 for EGFR-KDD, 07/15/2020 for Ex19Del and L858R ▪ Replicate 2: 07/25/2020 for EGFR-KDD, 07/17/2020 for Ex19Del and L858R ▪ Replicate 3: 08/03/2020 for EGFR-KDD, 07/13/2020 for Ex19Del and L858R 	Cell viability assay on BaF3 EGFR-KDD, Ex19Del and L858R cells supplemented with 10% FBS and treated with 50ng/mL EGF, cetuximab and afatinib	Each independent experiment included 3 technical replicates for each sample tested.

REVIEWERS' COMMENTS

Reviewer #1 (Remarks to the Author):

Du et al. have responded more than thoroughly to the previous (extensive) reviews, and several aspects of the study and its presentation are substantially improved. Despite the differences in opinion on some issues (notably the reasoning behind the model in Fig 3d), I think the manuscript should be published. The current presentation also gives a much better sense of the extent to which experiments were reproduced – with useful addition of quantitation for that purpose.

A couple of minor typos noticed:

On line 220, the authors refer to Fig. 2a when it should be Fig. 2b.

Line 245, JBM should be JMB.

Reviewer #4 (Remarks to the Author):

Dr. Lovly and Co-authors have presented the novel findings of kinase domain duplications in ERBB2, ERBB3, and ERBB4 beyond the initial finding of EGFR KDD. This in itself represents the identification addition druggable oncogenes. Additional mechanistic insight for the activation and multimerization is provide here. The authors also provide in vitro data supporting the trial of combination Ab and TKI in the treatment of these oncogenes. In my opinion the authors have addressed extensively the concerns raised by reviewer #1.

Response to Reviewer Queries

Reviewer #1 (Remarks to the Author)

Du et al. have responded more than thoroughly to the previous (extensive) reviews, and several aspects of the study and its presentation are substantially improved. Despite the differences in opinion on some issues (notably the reasoning behind the model in Fig 3d), I think the manuscript should be published. The current presentation also gives a much better sense of the extent to which experiments were reproduced –with useful addition of quantitation for that purpose.

A couple of minor typos noticed:

On line 220, the authors refer to Fig. 2a when it should be Fig. 2b.

Line 245, JBM should be JMB.

Authors' Response: We appreciate the Reviewer's positive comments. We have amended the text as suggested.

Reviewer #4 (Remarks to the Author)

Dr. Lovly and Co-authors have presented the novel findings of kinase domain duplications in ERBB2, ERBB3, and ERBB4 beyond the initial finding of EGFR KDD. This in itself represents the identification addition druggable oncogenes. Additional mechanistic insight for the activation and multimerization is provide here. The authors also provide in vitro data supporting the trial of combination Ab and TKI in the treatment of these oncogenes. In my opinion the authors have addressed extensively the concerns raised by reviewer #1.

Authors' Response: We appreciate the Reviewer's positive comments.